# A dual diffusion model enables 3D molecule generation and lead optimization based on target pockets

Lei Huang [1,2], Tingyang Xu[2], Yang Yu [2], Peilin Zhao[2], Xingjian Chen[3], Jing Han[4], Zhi Xie[4], Hailong Li [4] ✉, Wenge Zhong[4], Ka-Chun Wong [1] ✉ & Hengtong Zhang [2] ✉

Structure-based generative chemistry is essential in computer-aided drug discovery by exploring a vast chemical space to design ligands with high binding affinity for targets. However, traditional in silico methods are limited by computational inefficiency, while machine learning approaches face bottlenecks due to auto-regressive sampling. To address these concerns, we have developed a conditional deep generative model, PMDM, for 3D molecule generation fitting specified targets. PMDM consists of a conditional equivariant diffusion model with both local and global molecular dynamics, enabling PMDM to consider the conditioned protein information to generate molecules efficiently. The comprehensive experiments indicate that PMDM outperforms baseline models across multiple evaluation metrics. To evaluate the applications of PMDM under real drug design scenarios, we conduct lead compound optimization for SARS-CoV-2 main protease (M$^{pro}$) and Cyclin-dependent Kinase 2 (CDK2), respectively. The selected lead optimization molecules are synthesized and evaluated for their in-vitro activities against CDK2, displaying improved CDK2 activity.

Structure-based drug discovery (SBDD) plays a crucial role in modern drug development[1,2] and catalysis[3]. Given a specific target protein, it aims to identify suitable drug molecules that effectively bind to a specific target protein. Traditional in silico methods such as virtual screening discover molecules by iteratively (1) placing molecules from existing databases into the protein pocket cavity and (2) filtering the molecules based on criteria such as energy estimation[4] and toxicity by experimental essays. Despite their widespread applications, these approaches suffer from two significant limitations[5,6]. Firstly, naive exhaustive searches in the massive chemical space (range from $10^{60}$ to $100^{100}$ depending on the size of desired molecules)[7] are prohibitively costly. Secondly, this workflow is constrained by historical knowledge, thus infeasible to explore and generate molecular structures which are not already recorded in the existing databases.

Fortunately, the emergence of deep learning methods has paved the way for efficient and accurate drug molecular structure learning and has greatly facilitated the exploration of chemical spaces in structured biological data distributions in recent years. A plethora of studies consider generating molecules via advanced generative methods, including variational autoencoder (VAE)-based models[8], generative adversarial network (GAN)[9], normalizing flows[10–13] and diffusion models[14,15]. By adopting generative models, current machine learning methods[10,11,16–19] start from learning the underlying distribution of molecules and yield candidate molecules from perturbed hidden information. Nonetheless, these methods typically represent the molecules as SMILES strings (1D) or graphs (2D), neglecting the crucial 3D-spatial information that is crucial to determine the properties of molecules. For example, a molecular graph is capable of forming

[1]City University of Hong Kong, Hong Kong, SAR, China. [2]Tencent AI Lab, Shenzhen, China. [3]Harvard Medical School, Boston, USA. [4]Regor Therapeutics Group, Shanghai, China. ✉e-mail: hailong.li@qlregor.com; kc.w@cityu.edu.hk; htzhang.work@gmail.com

various conformations with different properties in 3D space due to intramolecular interactions or the orientation of structural motifs[20]. Some methods[12-15,21] have considered considering 3D-spatial information to generate 3D molecules. However, these methods do not involve the pocket information, limiting their ability to generate molecules with high binding affinity to specific protein pockets, which is crucial for wet experiments. This gives rise to the idea of structure-based generative chemistry, where the molecules with high binding affinity in the protein pocket are distilled. Here, the models perceive the 3D structure of the target pocket as conditional information and capture the interaction between molecules and proteins to learn the conditioned density of desired molecular data. Early studies focus on 1D or 2D molecule generation based on the pocket structure. Skalic et al.[22] propose a framework which is based on a variant GAN to generate SMILES strings of ligands after encoding the molecule strings in the shared space with the pocket protein. Xu et al.[23] employ the conditional RNN to train two descriptors which contain the 3D information of pocket to generate compounds. However, these methods also only generate molecules in SMILES sequence format (1D) or graph (2D) which cannot verify the fitness of the target pocket although they considered the 3D information of the pocket.

Most recently, some generative models have been proposed to enable 3D sampling of molecules within the pocket cavity[24-27]. Early attempts[25] employed conditioned VAE to handle the voxelized atomic density images and obtained molecules from the images by post-processing algorithm. They use convolutional neural networks to encode the density girds into separate ligand and protein latent spaces. Compared with previous work which can only generate small molecules, this method can generate more drug-like 3D molecules. However, this method compressed the pocket structure information and failed to generate accurate molecules with fine-grained positions. Besides, it does not consider the equivariance of molecular geometry and is hard to scale to large proteins due to the voxel design. To tackle this issue, Luo et al.[26] model the atom probability by graph neural networks and equip the mask-fill schema to estimate the landscape of the pocket. Liu et al.[27] further incorporate distance and angle embeddings to place the atoms one by one. The existing generative models typically adopt the auto-regressive strategy to sample the atom sequentially, which enables the current atom to learn the historically placed atom information. Nonetheless, these methods have inherent limitations: (1) the models may suffer from deviation accumulations especially when invalid structures are generated in the early steps; (2) the sequential sampling algorithm that relies on MCMC does not consider the global context information; (3) auto-regressive models place one atom at a time, thus the number of the sequential sampling is the same as the length of the ligand, making it time-consuming to generate large-scale molecules. Consequently, the challenge of achieving 3D sampling of molecules within pocket cavities persists, as existing methods face limitations in accurately capturing fine-grained positions, efficiently exploring the chemical space, and maintaining global context information.

Recently, diffusion models[28] have garnered a huge amount of attention in computer vision tasks[29-31], especially in point cloud generation[32-34] which shares similarities with 3D molecule generation. These methods excel at inpainting 3D objects by learning the joint distribution. Although there is a diffusion model[35] developed for structure-based molecule generation, however, it requires training the user-defined parameters, leading to inefficient sampling. Besides, it only utilizes the fully connected adjacent matrix thus ignoring the intrinsic topology of the molecular graph. Inspired by the success of diffusion models in computer vision tasks, we propose a one-shot generation framework named Pocket based Molecular Diffusion Model (PMDM) to tackle these issues. Fig. 1 outlines the overview of PMDM. Specifically, molecular atoms with fixed pocket information are regarded as 3D point clouds and diffused in the forward process

which is similar to the phenomena in nonequilibrium thermodynamics. The goal of PMDM is to learn how to reverse such process to model a conditioned data distribution. This allows us to efficiently generate accurate molecules with high binding affinity once the pocket information is fixed. However, regular methods for 3D point clouds cannot involve edge information like chemical bond information if we represent 3D molecular geometries as 3D point clouds. Thus, we define a dual diffusion strategy to build two kinds of virtual edges. In detail, pairs of atoms with interatomic distances below a certain threshold are bonded via covalent localized edges because chemical bonds can dominate interatomic forces when two atoms are close enough to each other while global edges are linked to the remaining pairs of atoms to simulate the van der Waals force. Besides, we design an equivariant dynamic kernel that obeys the translation, rotation, reflection, and permutation equivariance of molecular geometry systems. The experiments on synthetic CrossDocked dataset[36] demonstrate that PMDM can generate drug-like, synthesis-accessible, diverse molecules with high binding affinity against specific proteins and outperform the state-of-the-art (SOTA) models on multiple evaluation metrics. By proposing the sampling algorithm for scaffold hopping and linker generation, PMDM exhibits the ability to generate a large number of bioactive molecules with high binding affinity for target proteins without retraining on the specific datasets. The in-vitro experiments suggest that the selected molecules display improved CDK2 activity and comparable or even better CDK1 selectivity than the reference compound.

## Results
### Overview of the PMDM model
Figure 1 outlines an overview of the conditional generative model PMDM, elucidating its structural components and the processes involved in training and sampling. PMDM gradually introduces Gaussian noise in the forward process while employing a parameterized reverse process to iteratively eliminate the noise (Fig. 1a). The model comprises two invariant graph neural networks Schnet[37] to obtain the molecule embeddings $z_L$ and pocket embeddings $h_P$ (Fig. 1b). To facilitate conditional generation, we have designed two context mechanisms to incorporate both the semantic and geometric information of the protein pocket. Specifically, cross-attention layers are utilized to calculate the attention scores of the molecule and protein, protein pocket. Additionally, a dual diffusion strategy is employed to enable the model to discern atom-wise forces. This strategy involves constructing two types of virtual edges. Firstly, pairs of atoms with interatomic distances below the local threshold $\tau_l$ are bonded via covalent localized edges because chemical bonds tend to dominate interatomic forces when atoms are in close proximity. Secondly, we build the global edges which are linked to the remaining pairs of atoms to simulate the van der Waals force for the atoms whose distances are greater than the local threshold $\tau_l$ but less than the global threshold $\tau_g$ (Fig. 1d). Furthermore, we have designed an equivariant dynamic kernel that adheres to the translation, rotation, reflection, and permutation equivariance of molecular geometry systems. To ensure the generated molecule is adapted to the structure pocket, we keep the pocket position fixed during the update of the hidden states in the dual equivariant encoders.

In the training stage, both molecules and their corresponding binding protein pockets are regarded as 3D point clouds. In the forward process of PMDM, the molecule input undergoes diffusion, resembling phenomena observed in nonequilibrium thermodynamics, with the sampled time step drawn from the union distribution. Meanwhile, the protein pocket input remains fixed as it serves as the conditional information (Fig. 1c). The primary objective of PMDM is to learn how to reverse this process to model a conditioned data distribution. This enables the efficient generation of accurate molecules with high binding affinity when the pocket information is fixed. At each

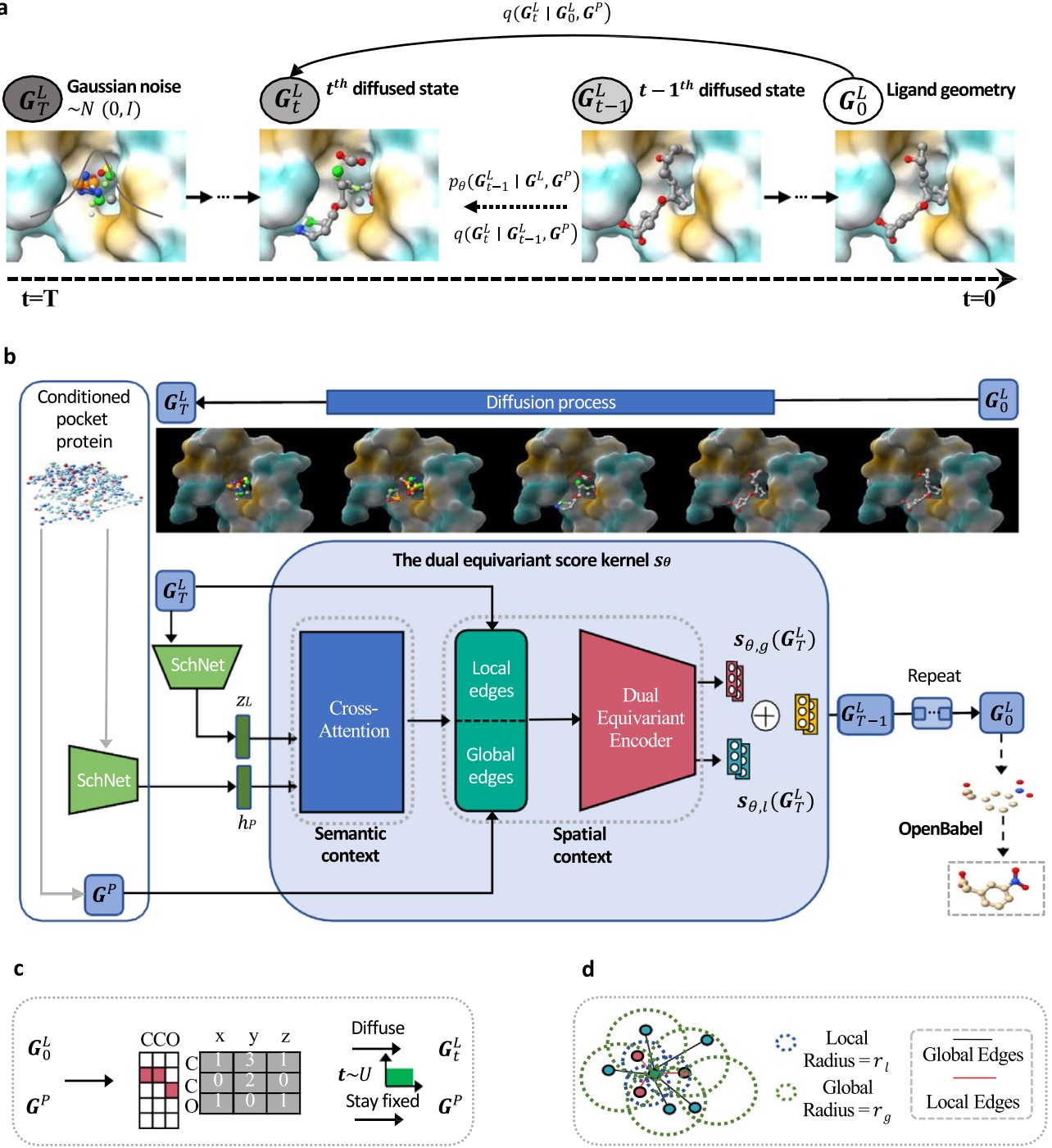

**Fig. 1 | Overview of the PMDM framework. a** The diagram of diffusion process in PMDM. PMDM is based on the diffusion model, which defines two Markov processes: diffusion process and reverse process. The diffusion process iteratively adds Gaussian noises to the ligand data $\mathbf{G}_0^L$ according to a variance preserve schedule while the reverse process generates a realistic ligand from the corruption state $\mathbf{G}_T^L$ through eliminating the noise. In the training phase, any immediate state $\mathbf{G}_t^L$ can be calculated by $q(\mathbf{G}_t^L|\mathbf{G}_0^L,\mathbf{G}^P)$, we will elaborate this desired property in section Methods. Since the diffusion process is fixed, PMDM is trained to learn the reverse probability transition distribution $p_\theta(\mathbf{G}_t^L|\mathbf{G}_{t-1}^L,\mathbf{G}^P)$. **b** The structure of PMDM. PMDM is designed to generate the ligand given the target pocket protein. PMDM could encode the protein semantic context information and spatial context

information. The protein point cloud data is fed into an invariant encoder SchNet[38] to obtain the semantic representation $h_p$. Then the semantic information is fused with the ligand data by the cross-attention layers. We define local and global edges for ligand point cloud data. Then the ligand data with two kinds of edges and pocket protein data go through the dual equivariant encoder which handles different edges and keeps the protein spatial information fixed to obtain the score $s_\theta$. This process will repeat $T$ times until we obtain the realistic ligand geometry $\mathbf{G}_0^L$, and we use OpenBabel to construct the bonds. **c** The ligand and protein are represented by one-hot encoded atom types and 3D coordinates. The ligand data will be diffused to $\mathbf{G}_t^L$ at an arbitrary time step while the protein will stay fixed during training. **d** The construction of local and global edges.

time step, the model outputs the (Stein) score, which represents the logarithmic density of the data point. The ELBO objective is derived from these scores and serves as the loss function (See Method).

In the sampling stage, we initialize the data state by sampling from $\mathcal{N}(0, I)$ and obtain the transition probability by the dual equivariant encoder of PMDM, given the target pocket protein. The next less chaotic states are iteratively generated by $p_\theta(\mathbf{G}_{t-1}^L | \mathbf{G}^L, \mathbf{G}^P)$. The final molecule $\mathbf{G}_0$ is generated by progressively sample $\mathbf{G}_{t-1}$ for $T$ times. Finally, the atom types of the molecule are identified by adopting the argmax function to choose the atom type that has the largest value while we directly adopt $r_0^L$ outputted by the model.

## Metrics
We adopt widely-used metrics[26,38] to evaluate the quality of molecules generated by PMDM: (1) **Vina Score** estimates the binding affinity between the ligand and the target pocket which is the most important measurement to evaluate how the generated molecule fits into the protein pocket of interest; (2) **High Affinity** is the percentage of the molecules whose **Vina Score** is higher than the ground truth molecule in the test set; (3) **QED** estimates the drug-likeness of the molecule via combining several desirable molecular properties; (4) **SA** (synthetic accessibility) measures the molecule synthetic accessibility; (5) **Lipinski** measures how many rules the drug follows five Lipinski's rules[39]; (6) **LogP** indicates the octanol-water partition coefficient, which should be between -0.4 and 5.6 if the molecule is a good drug candidate[40]; (7) **Diversity** represents the average pairwise Tanimoto dissimilarity of the generated molecules targeting for each pocket. (8) **Time** is the average time to generate 100 samples for each pocket across all the targets.

## Baseline models
We compared PMDM with SOTA models for the SBDD task including CVAE[25], AR-SBDD[26], and DiffSBDD[35]. CVAE and AR-SBDD adopt an auto-regressive strategy to generate samples. DiffSBDD is based on the diffusion model. Besides, we also report the calculation results of molecules in the test set for a more comprehensive comparison.

## Evaluation of PMDM on the general metrics
We generate 100 molecules for each target protein in the test set (10000 molecules in total). Here, the size of the generated molecules is sampled from the size distribution of the training set. The overall results of PMDM and the baseline models are presented in Table 1. We observe that PMDM outperforms all the baseline models on almost every metric except SA and Diversity. According to the Vina score, PMDM is able to generate the molecules with high affinity to the pocket ($-7.472 \pm 2.90$) which is 20.2% better than the best auto-regressive baseline, AR-SBDD and 15.0% than another diffusion model DiffSBDD. Besides, PMDM surpasses AR-SBDD and DiffSBDD on QED ($0.594 \pm 0.12$) by 18.3% and 20.0%, and Lipinski ($4.975 \pm 0.16$) by 3.9% and 3.7%. The logP value of PMDM within the compliance range ($-0.4$ - $5.6$) implies that the molecules generated by PMDM hold greater promise as drug candidates, which is crucial for clinical trials. For the SA, PMDM performs much better than the diffusion model DiffSBDD and CVAE, and archives competitive results compared to AR-SBDD. On the other hand, the diversity of generated molecules should fall within a reasonable range so that the ability to explore the molecular space confined by protein pockets is high enough to discover potential molecules. As in Table 1, the diversity of PMDM is a little bit lower than that of AR-SBDD and DiffSBDD but higher than that of CVAE, implying that our model satisfies this desired property.

Notably, the molecules generated by PMDM perform even better than those in the test set on Vina Score, QED, and Lipinski, suggesting that PMDM has great potential to generate more drug-like molecules with higher affinity outside the distribution of the dataset. The one-shot nature of PMDM ensures that the model effectively considers the

global information of the molecule rather than sampling the local optimum atom like auto-regressive methods, which is time-consuming. Besides, although DiffSBDD also generates molecules in a one-shot manner, it incorporates neural networks to learn the user-defined parameters which also requires additional computations. Thus, as a one-shot method with fewer back forward parameters, PMDM is able to sample molecules up to twenty times faster than auto-regressive models and two times than DiffSBDD while achieving better or competitive performance.

## Analysis of PMDM on local geometries
Although conventional metrics can reflect the quality of generated molecules to a certain extent, the quality of the sub-structures of generated molecules also needs to be considered when evaluating model performance. We select several pocket proteins to visualize as representative samples for sub-structure analysis. As depicted in Fig. 2, we choose 14GS, 2RMA, and 3AF2 as the targeted pocket protein. We observe that the AR-SBDD DiffSBDD tends to generate the three-atom rings while our proposed model PMDM avoids generating such unstable rings. Although the dataset contains only 3% three-atom rings, AR methods generate more of these unstable structures, which means that these methods get stuck in local optima and fail to learn the data distribution well. Instead, PMDM can consider the shape of the pocket hole and generate larger and more complicated rings which are shown in the 3AF2 pocket samples.

To obtain a global overlook of the structure distributions of generated molecules, we present the ring number distribution of molecules generated by PMDM and the molecules in the test set and the train set (Fig. 3a). The distribution of the PMDM is close to the test set and the train set. The molecules generated by PMDM contain around 2.990 rings while the molecules in the test set and the train set contain 2.470 rings and 2.737 rings on average. Overall, the results suggest that PMDM is able to learn the ring sub-structure size distribution from a local perspective and the distribution of ring numbers from a global perspective.

To further quantify the ring sub-structure of the molecules generated by those methods, we report the proportion of molecules containing rings of different sizes in the training set, the test set, and the generated sets from the methods. In the case of molecules that contain multiple rings, the counting process takes into consideration each individual ring present, resulting in repeated counts proportional to the number of rings. As presented in Fig. 3b, the molecules generated by PMDM contain few unstable rings, including the three-atom ring and the four-atom ring. Auto-regressive methods have a tendency to limit themselves to the local topological structure by considering only the previously generated part, which often results in the generation of small rings. DiffSBDD constructs the fully connected edges for all the atoms, which may lead to a higher likelihood of forming small rings due to the lessening of interatomic distances. Specifically, PMDM yields a mere 2% of molecules, while CVAE generates 36.1%, AR-SBDD generates 48.4%, and DiffSBDD generates 44.4% of molecules containing three-atom rings. Regarding macro rings, PMDM generates as many 8-atom and 9-atom rings as other methods, except for DiffSBDD which generates 2.5% of molecules containing 9-atom rings, whereas other methods only generate 0.6%. It is evident that both the training set and the molecules generated by PMDM exhibit a Gaussian-like distribution in the number of rings. We also notice that PMDM generates relatively more molecules with 7-atom rings. This is due to the fact that distinguishing between 7-atom rings and 6-atom rings at the geometric level is challenging, given their similar structural appearances. On the other hand, PMDM generates fewer molecules which contain 8-atom and 9-atom rings compared to DiffSBDD since PMDM constructs the local edges to consider close atomic forces while DiffSBDD only constructs the fully connected edges to incorporate many distant atomic interactions. In contrast,

**Table 1 | The comparison of 10000 generated molecules of PMDM and baseline models on the CrossDocked dataset**

| Methods | Vina Score (kcal/mol) ↓ | High Affinity ↑ | QED ↑ | SA ↑ | Lipinski ↑ | LogP | Diversity | Time (seconds) ↓ |
|---|---|---|---|---|---|---|---|---|
| CVAE | −6.144 ± 1.57 | 0.238 | 0.369 ± 0.22 | 0.590 ± 0.15 | 4.367 ± 1.14 | −0.140 ± 2.73 | 0.654 ± 0.12 | - |
| AR-SBDD | −6.215 ± 1.54 | 0.267 | 0.502 ± 0.17 | **0.675** ± 0.14 | 4.787 ± 0.50 | 0.257 ± 2.01 | 0.742 ± 0.09 | 19659 ± 14704 |
| DiffSBDD | −6.584 ± 2.06 | - | 0.495 ± 0.15 | 0.336 ± 0.09 | 4.795 ± 0.49 | - | 0.730 ± 0.11 | 1634 ± 769 |
| PMDM | **−7.572** ± 2.50 | **0.628** | **0.594** ± 0.12 | 0.611 ± 0.16 | **4.975** ± 0.16 | 0.301 ± 1.01 | 0.709 ± 0.10 | **906** ± 110 |
| Test set | −7.024 | - | 0.466 | 0.725 | 1.413 | 0.929 | - | - |

We generate 100 samples for each target pocket.
The bold values indicate the best performance results. These values may not necessarily be the largest or smallest, depending on the different metrics.

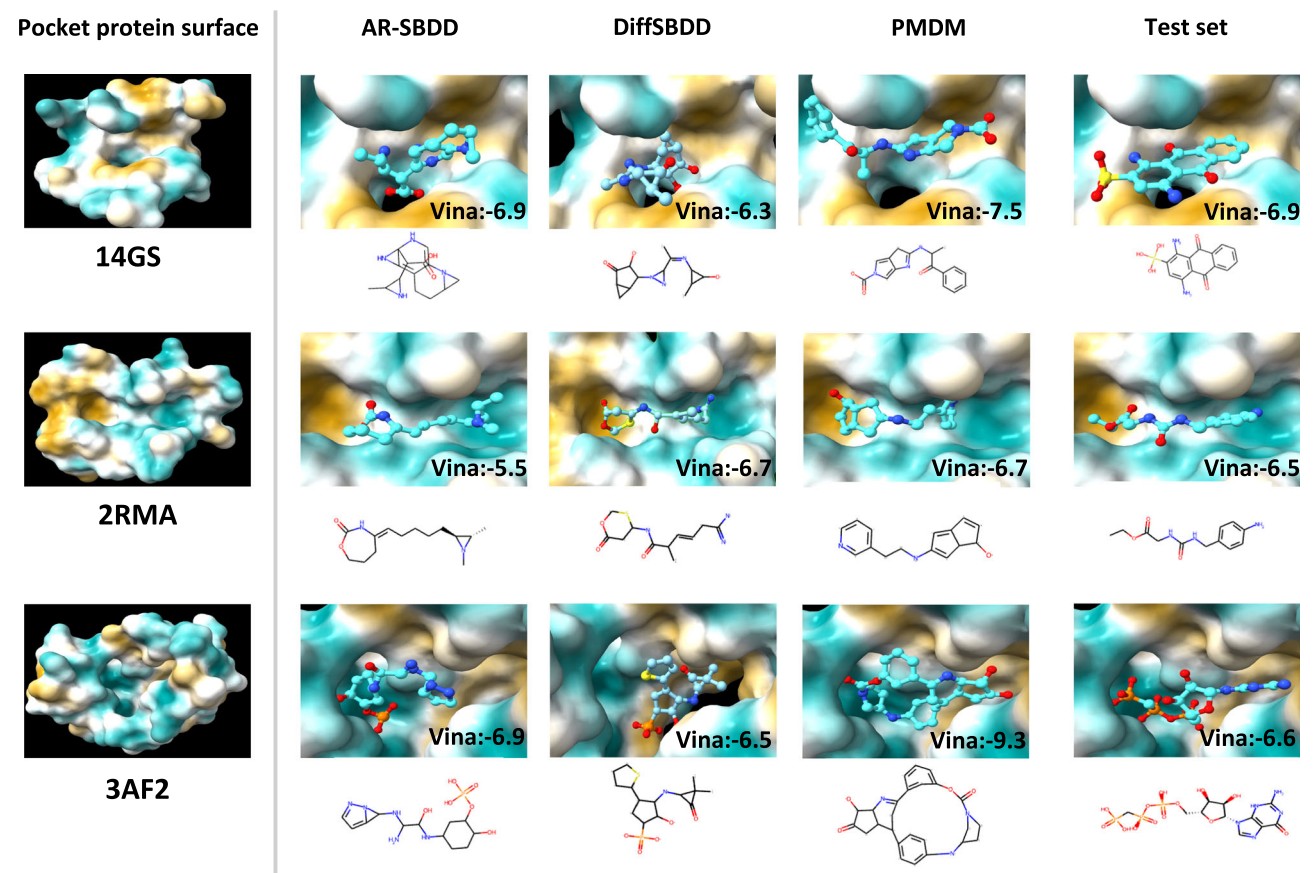

**Fig. 2 | The comparison of the example molecules which are generated by AR-SBDD, DiffSBDD and PMDM, and from the test set, respectively.** The molecules are targeting GLUTATHIONE S-TRANSFERASE (PDB id: 14GS), a complexed Crystal Structure of Cyclophilin (PDB id: 2RMA), and Pantothenate kinase (PDB id: 3AF2), respectively.

both auto-regressive methods and the diffusion model DiffSBDD are prone to generate a relatively large amount of unstable molecules with three-atom rings and four-atom rings. Besides, PMDM is inclined to generate more molecules with five-atom rings and six-atom rings, where hydrogen bonds occur most frequently. Such sub-structures are actively used in drug design. Another meaningful phenomenon is that PMDM generates molecules in a similar proportion to the test set, which indicates that PMDM can learn the data distribution without bias.

We also screen out the common bond pairs and triples according to previous work[41] and then adopt RDKit to calculate the bond angles and the dihedral angles in radians. We measure the distribution of the bond angles and dihedral angles of the generated molecules and reference molecules and then assess the distribution deviation by utilizing the Kullback-Leibler (KL) divergence. As reported in 3c and 3d, the molecules generated by PMDM demonstrate the lowest KL divergence in all the bond pair patterns and bond triple patterns among all

the models. The results indicate that the PMDM is capable of capturing the local atom geometry of the data.

**Analysis of PMDM on chemical space distribution**

Having analyzed the local geometry of molecules generated by PMDM, we then evaluate the generated molecular chemical space distribution from a global perspective. Since the three-dimensionality of the chemical structures is the essence of molecular design in medicinal chemistry, we also place our focus on the shape of chemical structures. Herein, we adopt 2D and 3D molecular fingerprints including Morgan[42], RDKit, and USRCAT (Ultrafast Shape Recognition with CREDO Atom Types)[43] fingerprints to represent the chemical space of generated molecules and test set molecules. Specifically, we utilize the Extended-Connectivity Fingerprints(ECFP) which are based on the Morgan algorithm to assign the unique identifiers after preset iterations. This kind of fingerprint takes atom types including connectivity and chemical features such as Donor and Acceptor and the

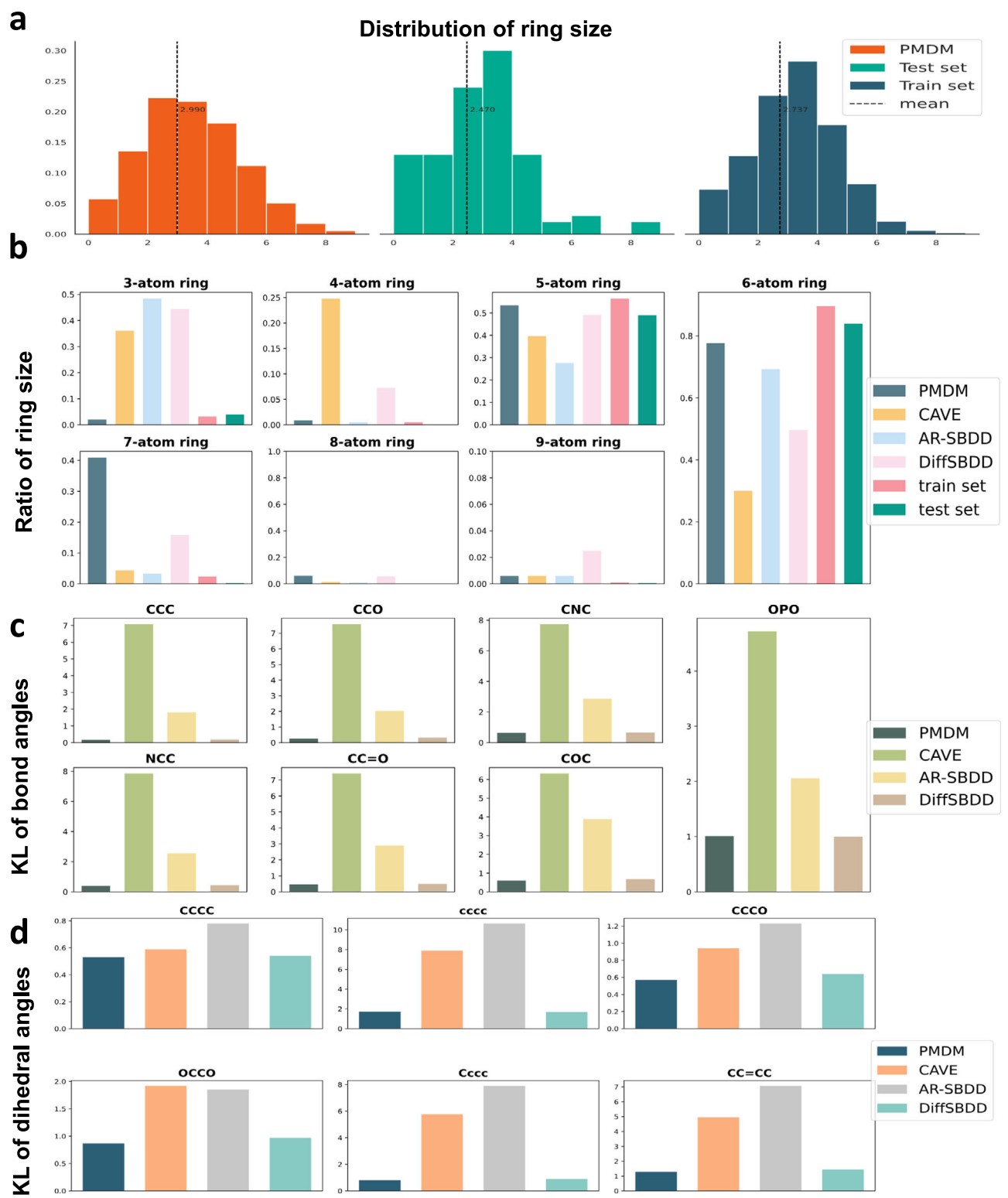

**Fig. 3 | Local geometry analysis. a** The distribution of the number of rings of molecules generated by PMDM. **b** The ratio of the molecules which contain rings of different sizes. **c** The KL divergence of the bond angles of generated molecules from models with the test set. **d** The KL divergence of the dihedral angles of generated molecules from models with the test set. Source data are provided with this paper.

neighborhood of each atom into account. RDKit fingerprint is designed to measure the molecular 2D substructure by considering the atom types and bond types, which is inspired by Daylight finger-print. In contrast, USRCAT improved USR (Ultrafast Shape Recognition) algorithm by incorporating pharmacophoric information to measure the molecular 3D shape. The visualization of the chemical space distribution using t-SNE is presented in Fig. 4. The chemical space of molecules generated by PMDM can cover the molecules from the test set in the 2D substructure space, indicating that PMDM can correctly model the 2D chemical space of the test set (Fig. 4a, b). As shown in Fig. 4c, the 3D chemical space of the generated molecules can basically capture the space of the test molecules due to the complexity

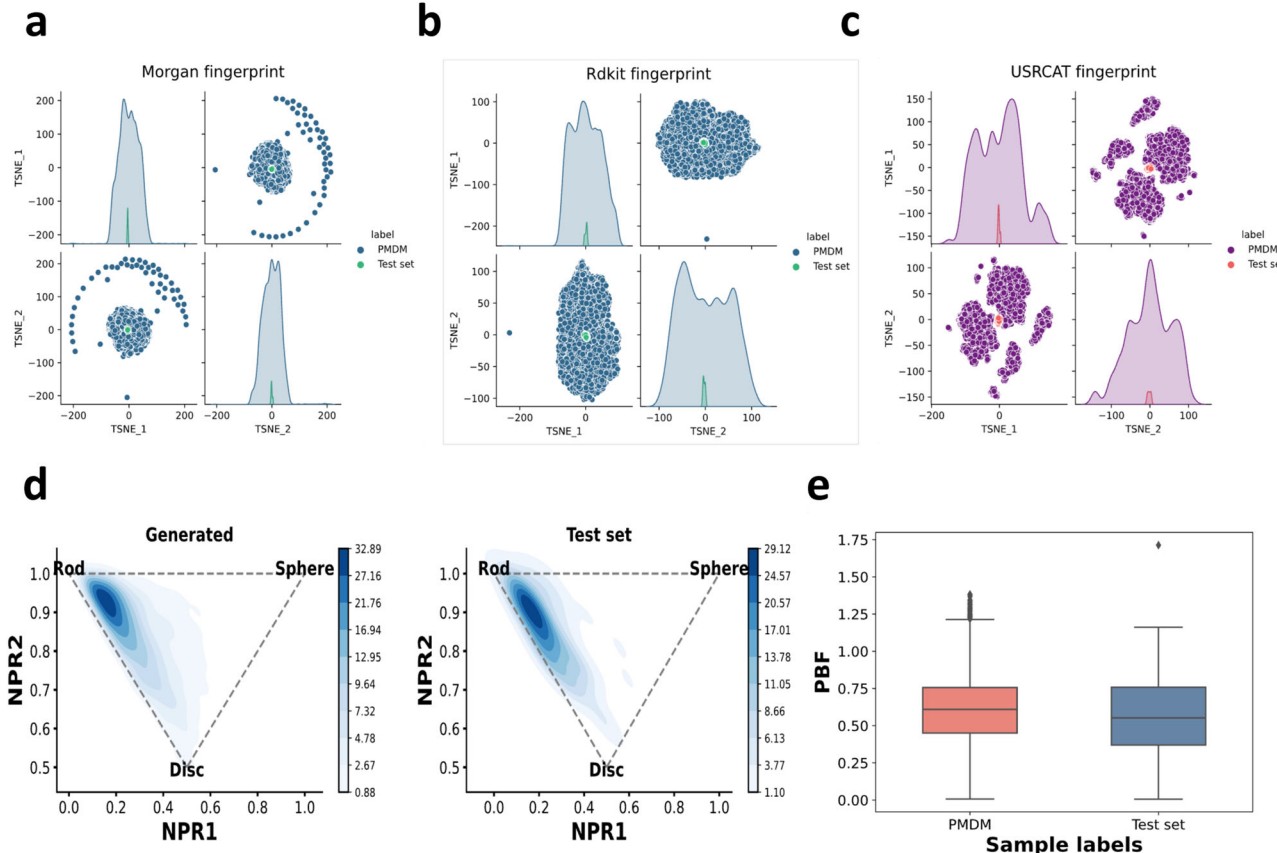

**Fig. 4 | The chemical space distribution visualization. a** Morgan. **b** RDKit.
**c** USRCAT fingerprints using t-SNE in two-dimensional space. 3D chemical structure measured by chemical descriptors. **d** Shape distribution of generated (left) and test set (right) molecules, which is visualized using the Normalized Principal Moment of Inertia ratios(NPR) descriptors. **e** The Plane of Best Fit (PBF) descriptor values ($n = 10,000$ for PMDM, $n = 100$ for test set; center line, median; box limits, upper and lower quartiles; upper line, maxima; whiskers, lower line, minima; $1.5 \times$ inter-quartile range;). Source data are provided with this paper.

of the conformations. Despite the incomplete coverage of the reference chemical space, there are no significant distribution mismatches between the generated and test set molecules. Furthermore, the wider three feature distributions of the molecules generated by PMDM highlight the capacity of PMDM to generate molecules across a broader chemical space. We have conducted an analysis of the chemical space distribution of molecules generated by other baseline models, as depicted in Supplementary Fig. 2. Our findings indicate that these models are unable to fully encompass the Morgan chemical space of the test set. Additionally, we observed that both CVAE and PMDM exhibited limited diversities in the 3D space.

Since the shape of 3D chemical structures is crucial for evoking molecular recognition activities with biological targets[44], we consider leveraging molecular descriptors to characterize the three-dimensionality of molecular structures beyond the aforementioned fingerprints. Here, we adopt two widely adopted molecular descriptors: Principal Moments of Inertia (PMI)[45] and Plane of Best Fit (PBF)[46], to investigate the specific 3D shapes from two perspectives. Specifically, the PMI descriptors can reflect the extent to which a given molecular geometry is rod-shaped, disc-shaped, or sphere-shaped while the PBF descriptors introduce the plane of best fit across all the heavy atoms of a molecule with a given conformation and calculate the distance of heavy atoms from the plane. Fig. 4d depicts the Normalized Principal Moment of Inertia ratios(NPR) on a ternary plot. The closer a point is to the three corners, the more its morphology exhibits these primitive shape classes. We can observe that the generated molecules exhibit a similar gather tend to the molecules from the test set. Both the generated and test set molecules are prone to gather around the

rod corner of the triangle. Furthermore, the generated molecules even touch the disc corner and sphere corner which are not covered by the original test data distribution, indicating that PMDM can not only learn the molecule 3D shape distribution of the dataset but also can explore shapes beyond the dataset by importing the random information, which can alleviate the out of distribution (OOD) problems in machine learning. In other words, PMDM has the potential to generate more diverse molecules even when facing proteins which do not obey the distributions of the proteins in the training set. We further calculate SA of these out-of-distribution molecules and the mean SA value is $0.628 \pm 0.29$, which is higher than the average SA value of the whole generated molecules, indicating that these molecules are computationally synthesizable. In addition, we have generated the NPR (Normalized Property Ratio) distributions of molecules produced by alternative baseline models, as shown in Supplementary Fig. 3. Our observations indicate that the molecules generated by PMDM exhibit the closest resemblance to the test set distribution. Conversely, the molecules generated by CVAE tend to cluster in the central region, while those from AR-SBDD extend towards the disc corner. Furthermore, DiffSBDD displays limited diversity in its generated molecules.

Besides, the molecules also achieve reasonable values on other chemical properties. As shown in Fig. 4g, we observe that the PBF values of the generated molecules align well with those of the test set molecules, indicating a similar degree of distance from the 2D shape. In contrast, CVAE exhibits a substantial gap compared to the test set, suggesting that the heavy atoms are significantly distant from the predefined plane (Supplementary Fig. 4). To summarize, PMDM can correctly model the distribution of important 3D and 2D molecular

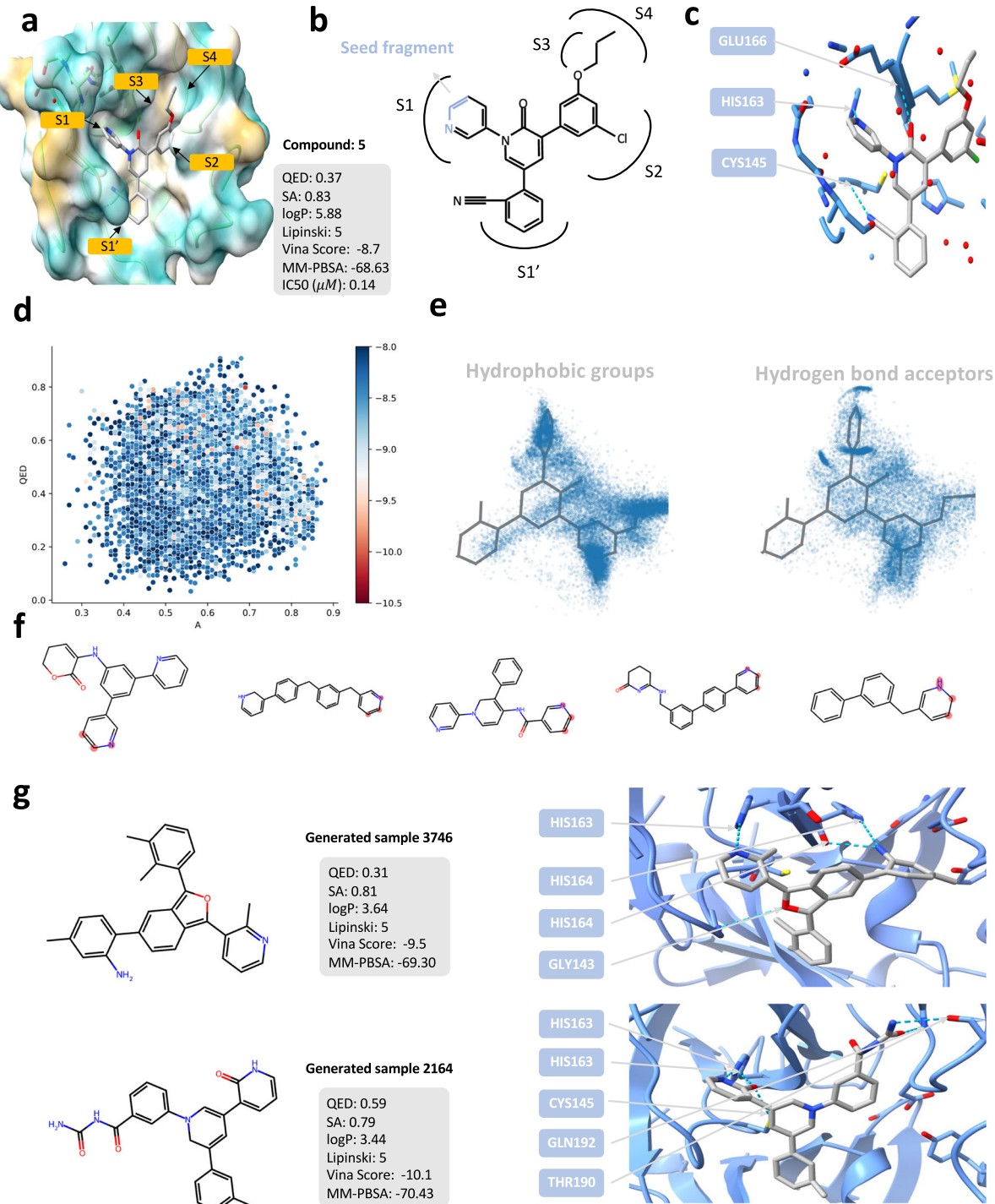

**Fig. 5 | Lead generation case of SARS-CoV-2 main protease (M^pro).** **a** The complex structure of noncovalent and nonpeptidic inhibitor Compound 5 targeting M^pro with the pharmacochemucal properties. **b** The structure of the compound 5. The blue part is the seed fragment which we utilize to generate the molecules. **c** The hydrogen bonds between compound 5 and M^pro. **d** The Vina score, QED, and SA distribution of the generated molecules with high affinities. **e** The spatial distribution of the key pharmacophore groups of generated molecules with high affinities. **f** Examples of the scaffolds of generated molecules with high affinities. **g** Two examples of generated molecules with high affinities and lower free energies. Source data are provided with this paper.

structures and has the potential to guide a more comprehensive exploration to develop drug-like structures.

## Lead generation and optimization
**PMDM enables bioactive molecule generation towards specific targets.** To further investigate the practical implications of PMDM, we apply the trained model to generate the molecules targeted for SARS-CoV-2 related proteins with high affinities. Herein, we select SARS-CoV-

2 main protease (M^pro) as a test case to perform noncovalent inhibitors design following the previous work[47]. M^pro in SARS-CoV-2 is the main protease which can cleave the polyproteins at multiple positions to cleave the polyproteins, enabling it to be treated as a viable drug target. Recently, Zhang et al.[48] redesigned the weak hit perampanel to develop a series of potent noncovalent and nonpeptidic inhibitors targeting M^pro. In contrast to the peptide-like molecules that covalently bind to the residue (Cys145), the designed inhibitors avoid the issues of

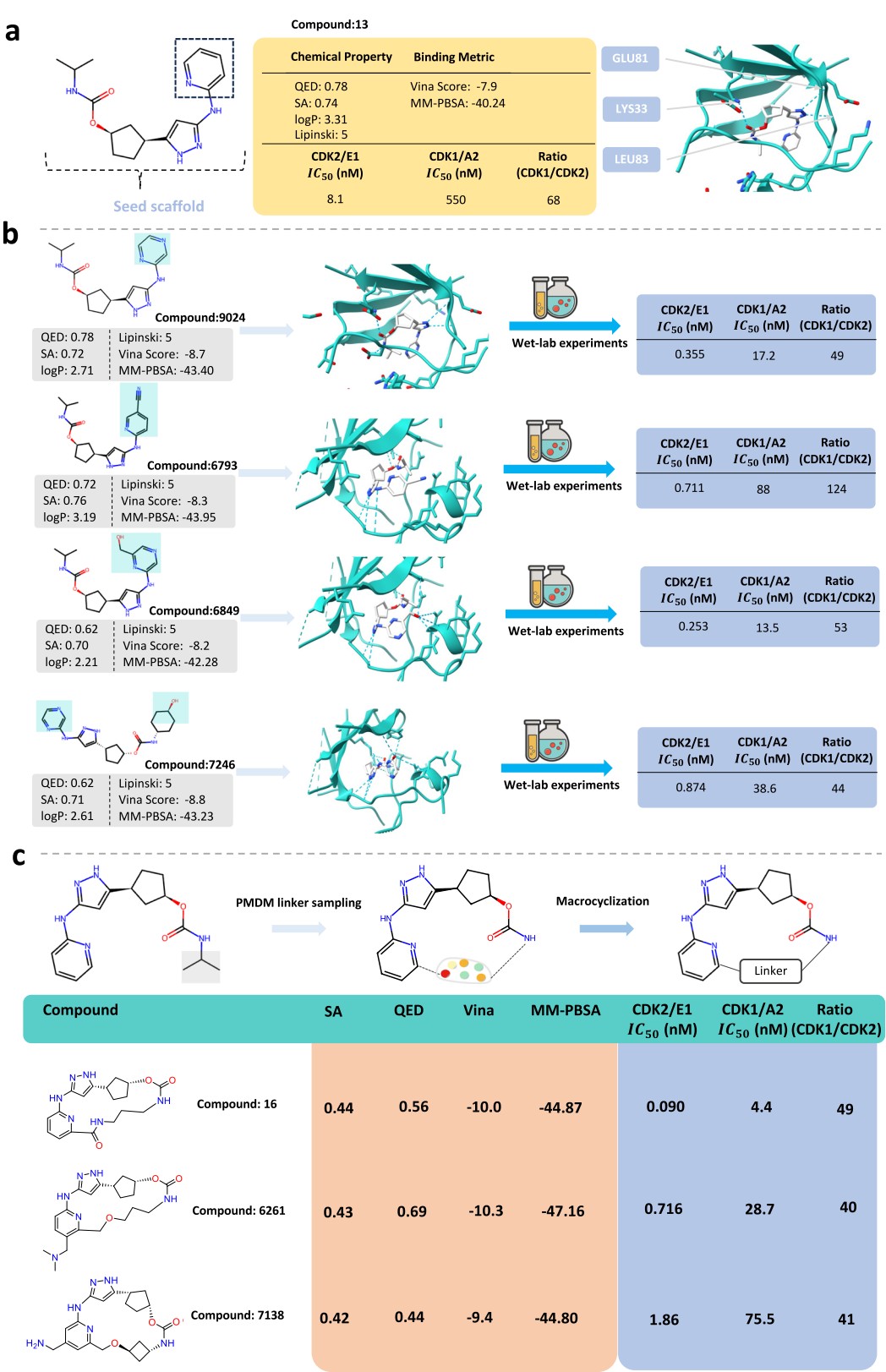

**Fig. 6 | Scaffold hopping case of Cyclin-dependent Kinase 2 (CDK2). a** The complex structure of the inhibitor Compound 13 targeting CDK2 with pharmacochemical properties. **b** The generated molecules with desired properties by the scaffold hopping strategy. We selected four molecules for wet-lab experiments and the inhibitory activities for CDK2/E1 and CDK1/A2 were evaluated by LANCE Ultra time-resolved fluorescence energy transfer (TR-FRET) assays **c** The generated macrocyclic CDK2 inhibitor by the linker generation method. We selected two molecules for wet-lab experiments and the inhibitory activities for CDK2/E1 and CDK1/A2 were evaluated by LANCE Ultra time-resolved fluorescence energy transfer (TR-FRET) assays. The wet-lab results of Molecule 16 are reported from previous work.

proteolytic degradation, limited antiviral activity, and molecular promiscuity toxicities. Fig. 5a shows the crystal structure of one of the inhibitors complexed with M^pro with high bioactivity which is included in the Protein Data Bank (PDB ID: 7L11). There are several features which contribute to the high binding affinity of the molecule with M^pro, (1) the four rings of the molecule are being placed in the four sites (S1′, S2, and S3) of the pocket; (2) the carbonyl group in the central pyridinone ring forms a hydrogen bond with the backbone NH of residue Glu166; (3) the Nitrogen in the pyridinone ring connected by the central pyridinone ring forms a hydrogen bond with the residue His 163.

We aim to generate molecules with more diverse scaffolds, which is called lead generation. Toward this end, we utilize three atoms as the seed fragment which is the blue part of Fig. 5b. We adopt the inpainting method to diffuse the data of the seed fragment according to the time step and assemble with the generation part which remains denoising. The manually diffused fragment is finally denoised together with the part denoised in the previous step (See section Sampling given specific fragments). We generate 40000 molecules and filter out those whose Vina scores are smaller than -8.0 kcal/mol. Finally, we obtain 10627 molecules with high affinities. We checked all the filtered molecules, and none of them is presented in the training set. It indicates that PMDM can still generate molecules binding well to the targeting proteins despite the high affinity of the reference molecule. As demonstrated in Fig. 5, we plot the distribution of three key properties (QED, SA, and Vina score) of the filter molecules. As we can observe, PMDM is capable of generating molecules with good affinities while containing nice properties. Statistically, the average QED value of the molecules is 0.57, which is higher than the reference compound 5, and the maximum QED value is 0.75. For Vina score, the average value is −8.6 and the minimum value is -12.3 despite sacrificing performance in terms of synthetic accessibility which the average SA value is 0.30 and the maximum SA value is 1.0. The results demonstrate that PMDM can learn the distribution of the training data. Thus it could generate the molecules that adapt to the pocket structure and satisfy the requirement for high drug-likeness and good synthetic accessibility without inputting the desired properties as conditional information.

As we mentioned before, compound 5 contains several features contributing to its high affinity with the M^pro. In order to investigate whether the generated molecules contain the same features, we first calculate the pharmacophore models using the software Align-It. We selected the hydrophobic groups including aromatic ring(AROM), lipophilic region(LIPO), and aromatic and lipophilic(HYBL), to visualize the spatial distribution. As shown in Fig. 5e, the hydrophobic groups are clustered in the S1′, S1, S2, S3, and S4, which is in accordance with the compound 5, revealing the reducing capacity of PMDM. The visual inspection of the hydrogen bond acceptors demonstrates that the interactions of HIS 163 and GLU166 are covered by the generated molecules and the position of the hydrogen bond donors aligns well with those of compound 5. Besides, there are other cluster regions which suggest that the molecules also form hydrogen bonds with the residues of the pocket.

Since we only incorporate a small fragment which only contains three atoms as the seed fragment, PMDM manages to generate molecules with more rational scaffolds. Finally, we extracted 8950 Bemis-Murcko scaffolds by RDKit from the 9209 filter molecules. Fig. 5f shows examples of the scaffolds. The scaffolds reflect a shared commonality that all the scaffolds contain multiple rings, especially aromatic rings. The rings occupy the key binding sites (S1, S2, S3, and S4), which is key binding sites of M^pro. Besides, we found that there are scaffolds similar to that of the reference molecules. Specifically, the first and third example scaffolds shown in Fig. 5f consist of the aromatic ring connected to three rings.

The results imply that PMDM can discover the significant structure patterns which are verified by the reference molecule. To further investigate the quality of generated molecules, we selected two compounds with improved Vina scores and MM-PBSA values. We searched PubChem, ChEMBL, and DrugBank and found the two compounds are not recorded in all the datasets. Both compounds form similar interaction patterns with multiple residues of M^pro. In addition to the hydrogen bond with residue HIS163, the compounds form hydrogen bonds with more residues to achieve higher binding affinities. Specifically, the hydroxyl group besides the seed fragment in the generated sample 1 forms three hydrogen bonds with three residues: SER144, LEU141, and CYS145. For the generated sample 2164, the hydroxyl groups form five hydrogen bonds with four residues: HIS163, CYS145, GLN192, and THR190. Furthermore, both the generated molecule contains an aromatic ring connecting with three aromatic rings, which occupy the desired binding sites. These results spotlight that PMDM can generate molecules highly binding to the targeted proteins.

**PMDM enables scaffold hopping and linker generation for real synthetic bioactive molecule design.** Scaffold hopping is very important with appropriate hit compounds in lead optimization since it could not only generate known active scaffolds and improve binding affinity but also identify core structures that confer improved properties to overcome challenges in in-vitro profiles[49]. The development of advanced methods for making, analyzing, and purifying molecules of drug-like size has made it possible to synthesize analogs based on a common scaffold, along with the higher and more widespread access to commercial building blocks[50]. In order to validate whether our model could be applied in scaffold hopping to improve the binding affinities of the given basic bioactive molecule, we select Cyclin-dependent Kinase 2 (CDK2) as the target protein to generate desired molecules with core structures. The transition from G1 to S phase is driven by CDK2 in complex with its canonical partner cyclin E1 (CCNE1), which is often amplified in various cancers and is associated with worse survival outcomes in patients with breast, ovarian, and other malignancies[51–54]. Therefore, CDK2 is a potential cancer therapy target with abnormal levels or activity of many tumors. However, there is only a limited number of selective CDK2 inhibitors which are active in clinical trials.

We utilize PMDM to perform scaffold hopping on compound 13 complexed with CDK2 (PDB ID: 8H6T) to develop potential inhibitors[54]. The reference compound 13 is illustrated in Fig. 6a. The aminopyrazole moiety of the reference compound forms two hydrogen bonds with residues LEU83 and GLU81 and the carboxyl of the compound forms one hydrogen bond with the residue LYS 33. Besides, the gatekeeper residue's phenyl side chain has van der Waals interactions with the cyclopentyl ring of the compound. The pyridine moiety of the compound is oriented towards the solvent-accessible region and does not exhibit any significant polar or nonpolar contacts with CDK2. After reviewed by chemical experts, we remove the pyridine ring (dashed box in Fig. 6a) and reserve the remaining fragment as the seed scaffold (Fig. 6a) which is the key scaffold of the existing CDK2 inhibitors[54,55]. Finally, we leverage PMDM to generate a library of 10000 molecules for relacing the essential fragments. Then the potential inhibitors were filtered through Vina docking and MM-PBSA values with visual selection. We selected four compounds for further visual inspection, synthesizing, and testing. As illustrated in Fig. 6b, all the potential inhibitors exhibit higher Vina scores and MM-PBSA values with suitable SA scores. In-vitro experiments were conducted to assay their CDK1/2 inhibitory activities. As reported in Fig. 6b, all the molecules displayed improved CDK2 activity in enzyme assay, with significant CDK1 selectivity of at least ~44-fold. Compound 6793, which reintroduced a cyano group on the pyridine, displayed the best CDK1 selectivity (124-fold). Notably, compound 6849, containing the pyrazine ring with a hydroxymethyl, exhibited the highest CDK2 activity with substantial CDK1 selectivity. Additionally, compound 6849 turned out to be an advanced lead molecule during the lead

optimization campaign and exhibited good selectivity against other closely related kinases, including CDK9 (CDK9/T1 inhibition $IC_{50}$ = 32.3 nM, CDK9/CDK2 = 127) and GSK3$\beta$ (GSK3$\beta$ inhibition $IC_{50}$ = 703 nM, GSK3$\beta$/CDK2 = 2780).

The reference compound 13 exhibits a U-shaped conformation with the 6-position carbon of the pyridine ring and the nitrogen atom of the carbamate moiety oriented towards each other. The interatomic distance between the two atoms is determined to be 5.2 Å, offering logical connection points for macrocyclization. Given that our model facilitates ring formation via global edge construction, we investigated the potential of our model to generate linker molecules for macrocyclization. Unlike the specific fragments strategy, which utilizes the sampling given specific fragments strategy, we fix the seed fragment to enable the model to be aware of the fragment geometries (See section Sampling for linker generation). This could help the model generate linkers which connect the fragments coherently (Fig. 6c). The effect of linker length on the pharmacological properties of the reference compound is examined by medicinal chemists employing structure-based drug design approaches. Therefore, we explore linkers ranging from 4 – 6 atoms in length[54]. We fix the connecting points of the linker at the pyridine ring and the nitrogen atom of the carbamate motif. Finally, we utilize PMDM to generate 5000 macrocycles for the reference compound with the preset attachment points. We selected five potential macrocylized inhibitors for visual inspection after filtering through Vina docking and MM-PBSA values with visual selection. As illustrated in Fig. 6c, PMDM successfully generated the linkers which connect the preferred attachment points although we do not train PMDM on the specific linker datasets. The generated linkers improve the Vina score of the reference compound while retaining similar biological activity. Similar to the linear molecules, we selected three potential inhibitors which have better MM-PBSA values to investigate their in-vitro results. Specifically, macrocyclic compound 16 is generated again by PMDM which has been discovered by the previous work[54]. We performed in-vitro experiments on two additional compounds, namely compound 7138 and compound 6261. The two macrocyclic compounds are featured with different linker types and also display improved CDK2 inhibition activity and comparable CDK1 selectivity than the reference compound 13.

## Discussion

In this paper, we proposed a conditional diffusion model, PMDM which enables 3D small-molecule ligand generation conditioned on specific target proteins in a one-shot manner by incorporating the diffusion framework. PMDM utilizes a dual equivariant encoder to handle different (global & local) molecular dynamics. To achieve protein-conditioned generation, PMDM employs the cross-attention mechanism to consider the protein semantic information by fusing the protein representation and the ligand representation in a shared high-dimension space and incorporates the whole pocket as the input of the equivariant kernel in which the protein spatial information is fixed across the neural net layers, to consider the protein structure information.

With much less complexity and sampling time, PMDM achieves substantially better or competitive performance against the SOTA methods. The chemical space analysis for generated molecules demonstrates the rationality of the generated molecule structures compared to reference molecules in both 2D and 3D spaces. Furthermore, PMDM exhibits the ability to generate a large number of bioactive molecules with high binding affinity for target proteins that are not included in the training set. This inspires us to leverage PMDM to conduct lead compound generation and optimization for SARS-CoV-2 main protease ($M^{pro}$) and Cyclin-dependent Kinase 2 (CDK2), respectively. The lead generation results demonstrate that PMDM can generate molecules containing structure patterns verified by the reference molecule. By proposing the sampling algorithm given

specific fragments and sampling algorithm for linker generation, our model could be applied in lead optimization scenarios including scaffold hopping and generation without retraining it on the specific datasets. The selected lead optimization molecules are synthesized and evaluated for their in-vitro activities against CDK1 and CDK2. The in-vitro results indicate that all the molecules displayed improved CDK2 activity with suitable CDK1 selectivities. We anticipate that PMDM can advance the de novo drug optimization targeting the specific protein and accelerate future research in drug development.

## Methods
### Data processing
We conduct experiments to evaluate the generative performance of PMDM on the CrossDocked dataset[36]. This dataset contains 22.5 million docked protein-ligand pairs and each pair has different poses to multiple pockets across the Protein Data Bank. The ligands that were associated with a specific pocket were subsequently subjected to docking with each receptor assigned to that particular pocket by utilizing smina through Pocketome. The binding data (pK) for the CrossDocked2020 set was obtained from PDBbind v2017, and it was observed that 41.9% of the complexes have available binding affinity data. For a fair comparison, we follow previous work[26] to only choose the binding pose data whose root-mean-squared deviations (RMSD) is <1Å. The dataset is then refined through clustering at 30% sequence identity using MMseqs2[56], finally we obtain 100,000 pairs for training and 100 pairs for evaluation. Figs. 5a and 6b are generated by ChimeraX software[57].

### Preliminary
Let $G = (x, r)$ denote the 3D molecular geometry where $x = (x_1, x_2, \cdots, x_n) \in \{0, 1\}^{n \times f}$ denotes the discrete one-hot encoded atom types (a.k.a. chemical elements), and $r = (r_1, r_2, \cdots, r_n) \in \mathbb{R}^{n \times 3}$ denotes the continuous atom coordinates as depicted in Fig. 1c. Specifically, we denote 3D ligand geometry as $G^L = (x^L, r^L)$ and 3D protein pocket geometry as $G^P = (x^P, r^P)$. We denote $G_t$ for $t = 1, ..., T$ as a sequence of latent geometries where $t$ indicates the index of diffusion steps.

### Background
The diffusion model[28] is formulated as two Markov chains: *diffusion process* and *reverse process* (a.k.a denoising process). The diffusion process iteratively adds Gaussian noises to the data according to a variance preserve schedule while the reverse process gradually refines the data until it recovers the real data by eliminating the noise. The refined goal of the diffusion model is to learn the reverse process via a parameterized neural network.

The diffusion process gradually diffuses the real data distribution into a predefined noise distribution with the time setting 1...$T$. The transformation in every time step is set as a Gaussian distribution. This whole process is then formulated as a fixed Markov chain that gradually adds Gaussian noise to the data with a variance schedule $\beta_1...\beta_T$ ($\beta_t \in (0, 1)$):

$$q(G_t \mid G_{t-1}) = \mathcal{N}\left(G_t; \sqrt{1-\beta_t}G_{t-1}, \beta_t I\right), q(G_{1:T}|G_0) = \prod_{t=1}^{T} q(G_t|G_{t-1}),$$
(1)

where $G_{t-1}$ is mixed with the Gaussian noise to obtain $G_t$ and $\beta_t$ controls the extent of the mixture. By setting $\bar{\alpha}_t = \prod_{s=1}^{t} 1 - \beta_s$, a delightful property of the diffusion process is achieved that any arbitrary time step, $t$, sampling of the data has a closed-form formulation via a reparameterization trick as:

$$q(G_t \mid G_0) = \mathcal{N}\left(G_t; \sqrt{\bar{\alpha}_t}G_0, (1 - \bar{\alpha}_t)I\right).$$
(2)

We can observe that the final distribution will be closer to a standard Gaussian distribution if the time step is large enough.

The reverse process is designed to recover the real data $\mathbf{G}_0$ from the diffused data $\mathbf{G}_T \sim p(\mathbf{G}_T)$ which is achieved by the diffusion process. The reverse process is also a Markov chain with learnable parameters which can be formulated as follows:

$$
\begin{aligned}
p_\theta(\mathbf{G}_{t-1} \mid \mathbf{G}_t) &= \mathcal{N}\left(\mathbf{G}_{t-1}; \boldsymbol{\mu}_\theta(\mathbf{G}_t, t), \sigma_t^2 I\right), p_\theta(\mathbf{G}_{0:T-1} \mid \mathbf{G}_T) \\
&= \prod_{t-1}^{T} p_\theta(\mathbf{G}_{t-1} \mid \mathbf{G}_t),
\end{aligned} \tag{3}
$$

where $\boldsymbol{\mu}_\theta$ denotes the parameterized neural networks to approximate the mean, and $\sigma_t^2$ denotes user-defined variance. Specifically, we follow previous work[29] to paramterize $\boldsymbol{\mu}_\theta$ as:

$$
\boldsymbol{\mu}_\theta(\mathbf{G}_t, t) = \frac{1}{\sqrt{1-\beta_t}}\left(\mathbf{G}_t - \frac{\beta_t}{\sqrt{1-\bar{\alpha}_t}} \boldsymbol{\epsilon}_\theta(\mathbf{G}_t, t)\right), \tag{4}
$$

where $\boldsymbol{\epsilon}_\theta$ is a neural network w.r.t trainable parameters $\theta$. Having formulated the reverse process, we could maximize the likelihood of the training data as our object. Since directly calculating the likelihood is intractable, we adopt the variational lower bound (VLB)[29] to optimize.

$$
\begin{aligned}
\mathbb{E}\left[-\log p_\theta(\mathbf{G})\right] &\leq \mathbb{E}_{q(\mathbf{G}_0)}\left[-\log\left(\frac{p_\theta(\mathbf{G}_{0:T})}{q(\mathbf{G}_{1:T} \mid \mathbf{G}_0)}\right)\right] \\
&= \mathbb{E}_{q(\mathbf{G}_0)}[\underbrace{D_{\mathrm{KL}}(q(\mathbf{G}_T \mid \mathbf{G}_0) \| p(\mathbf{G}_T))}_{\mathcal{L}_T} \\
&\quad + \sum_{t=2}^{T} \underbrace{D_{\mathrm{KL}}(q(\mathbf{G}_{t-1} \mid \mathbf{G}_t, \mathbf{G}_0) \| p_\theta(\mathbf{G}_{t-1} \mid \mathbf{G}_t))}_{\mathcal{L}_t} \\
&\quad - \underbrace{\log p_\theta(\mathbf{G}_0 \mid \mathbf{G}_1)}_{\mathcal{L}_0}],
\end{aligned} \tag{5}
$$

where $q_\phi(\cdot)$ denotes a learnable variational noising encoder. The detailed derivation is left in the Appendix. $\mathcal{L}_T$ is a constant and $\mathcal{L}_0$ can be approximated by the product of the PDF of $\mathcal{N}(\mathbf{x}_0; \boldsymbol{\mu}_\theta(\mathbf{x}_1, 1), \sigma_1^2 I)$ and discrete bin width. Hence, we adopt the simplified training objective as follows:

$$
\mathcal{L}_t = \mathbb{E}_{\mathbf{G}_0}\left[\gamma \| \boldsymbol{\epsilon} - \boldsymbol{\epsilon}_\theta(\mathbf{G}_t, t)\|^2\right], \tag{6}
$$

where $\gamma = \frac{\beta_t^2}{2(1-\beta_t)(1-\bar{\alpha}_t)\sigma_t^2}$ refers to a weight term. We can observe that the terminal goal of the reverse process is to learn the noised added in the diffusion process. Actually, $\boldsymbol{\epsilon}_t$ can be represented as $\frac{\mathbf{G}_t - \sqrt{\bar{\alpha}_t}\mathbf{G}_0}{\sqrt{1-\bar{\alpha}_t}}$ from Eq. (2) via the reparameterize trick, where $\sqrt{\bar{\alpha}_t}\mathbf{G}_0$ is the mean $\mu$ and $1-\bar{\alpha}_t$ is the variance $\sigma^2$. Since the logarithmic gradient of $q(\mathbf{G}_t \mid \mathbf{G}_0)$ can be formulated as $\nabla_{\mathbf{G}_t} \log q_\sigma(\mathbf{G}_t \mid \mathbf{G}_0) = -\frac{\mathbf{G}_t - \sqrt{\bar{\alpha}_t}\mathbf{G}_0}{1-\bar{\alpha}_t}$, then we can obtain that $\boldsymbol{\epsilon} = -\nabla_{\mathbf{G}_t} \log q_\sigma(\mathbf{G}_t \mid \mathbf{G}_0)^* \sigma$. In other words, the purpose of the diffusion model is equivalent to moving the data distribution to the high-density region of the distribution led by the logarithmic gradient which initially starts from a low-density region. Therefore, the negative modified eliminated noise part $-\boldsymbol{\epsilon}_\theta \sigma$ is also regarded as the *(stein) score*[58], the logarithmic density of the data point at every time step. Now we can rewrite Eq. (4) as:

$$
\boldsymbol{\mu}_\theta(\mathbf{G}_t, t) = \frac{1}{\sqrt{1-\beta_t}}(\mathbf{G}_t + \beta_t \cdot \mathbf{s}_\theta(\mathbf{G}_t, t)). \tag{7}
$$

## PMDM: Pocket based Molecular Diffusion Model

In this section, we will elaborate on our proposed model PMDM: Pocket based Molecular Diffusion Model. Different from the pure diffusion model, PMDM is a conditional diffusion model instead, where the pocket protein guides the molecule generation. Thus, we attempt to model the $p_\theta\left(\mathbf{G}^L \mid \mathbf{G}^P\right)$ to obtain the distribution of the ligand binding to the pocket protein. The conditioned pocket protein semantic information is achieved by the cross-attention layer, which is effective for fusing various modalities. Specifically, we design a dual equivariant diffusion model for learning and generating the binding molecule geometry. Based on our previous model MDM[15], we devise two equivariant kernels to simulate the local chemical bonded graph and the global distant graph. In order to ensure the relative distance between the ligand and the protein, we employ an equivariant graph neural network EGNN to handle the whole pocket which can treat the pocket geometry as the condition information. Fig. 1b presents an overview of PMDM framework. We will elaborate on each component of PMDM in the following sections.

**Conditioned protein semantic information encoder.** Here, we adopt an invariant graph neural network SchNet[37] to encode the protein semantic information first. SchNet is a graph neural network modeling quantum interaction in molecules in 3D space. It consists of continuous-filter convolutional layers to model atomistic systems and maintain the invariant properties, achieving state-of-the-art performance for benchmarks of equilibrium molecules and molecular dynamics trajectories. Formally, the updates of protein node features are computed as follows:

$$
\mathbf{m}_{ij} = \phi_{\mathrm{w}}\left(d_{ij}\right)\phi_{\mathrm{s}}\left(\mathbf{h}_j^l\right), \mathbf{m}_i = \sum_{j \in N(i)} \mathbf{m}_{ij}, \mathbf{h}_i^{l+1} = \mathbf{h}_i^l + \phi_{\mathrm{m}}(\mathbf{m}_i), \tag{8}
$$

where $\phi_{\mathrm{w}}$ denotes a weight network, $\phi_{\mathrm{s}}$ and $\phi_{\mathrm{m}}$ are multilayer perceptrons (MLPs), $d_{ij}$ denotes the Euclidean distance between atom $i$ and atom $j$ of the pocket protein, and $N(i)$ is the radius neighborhood of atom $i$. We obtain the protein vector of the first hidden layer by a single leaner layer: $\mathbf{h}^0 = \text{Linear}(\mathbf{x}^P)$. We denote the final output of the protein encoder as $\mathbf{h}_P$ for a clear description. Similarly, we also employ another SchNet to project the ligand atom feature into an intermediate representation:

$$
\mathbf{z}_L = \text{SchNet}(\mathbf{x}^L, \mathbf{r}^L). \tag{9}
$$

We implement the cross-attention mechanism to fuse the protein semantic information and ligand hidden information:

$$
\begin{aligned}
\text{Attention}(Q, K, V) &= \text{softmax}\left(\frac{QK^T}{\sqrt{d}}\right) \cdot V, \text{with} Q = W_Q \cdot \mathbf{z}_L, K \\
&= W_K \cdot h_P, V = W_V \cdot h_P,
\end{aligned} \tag{10}
$$

where $\sqrt{d}$ turns the attention matrix into a standard normal distribution. Specifically, the protein information is considered as the query to compute the attention score. The output of the cross-attention layer incorporates the protein semantic information as the conditioned context.

**The dual equivariant score kernels.** As the molecular geometries are *invariant* to rotations and translations, we should take this property into account when devising the Markov kernels. In essence, Kohler et al.[13], proposed an equivariant invertible function to transform an *invariant* distribution into another *invariant* distribution. This theorem is also applied to the diffusion model[59]. If $p(\mathbf{G}_T)$ is

*invariant* and the neural network $q_\theta$ which learns to parameterize $p(\mathbf{G}_{t-1}|\mathbf{G}_t)$ is *equivariant*, then the distribution $p(\mathbf{G}_0)$ is also *invariant*. Therefore, we utilize an *equivariant* Markov kernel to achieve this desired property.

## Edge construction

As we mentioned before, molecular geometries in 3D generation are represented as point clouds. Thus, we need to construct edges manually for the point clouds to feed them into the subsequent equivariant kernels. Previous works[13],[14] consider the fully connected edges to feed into the equivariant graph neural network. However, the fully connected edges connect all the atoms and treat the interatomic effects equally but regret the effects of covalent bonds. Besides, the redundant edges contain meaningless information, leading to inefficiency. Therefore, we further define the edges whose lengths are shorter than the radius $\tau_l$ as local edges to simulate the covalent bonds and the edges whose lengths are between $\tau_l$ and $\tau_g$ as global edges to capture the long-distance information such as van der Waals force, which is shown in Fig. 1d.

Practically, we set the local radius $\tau_l$ as 3 Å which could include almost all the chemical bonds and the global radius $\tau_g$ as 6 Å. The one-hot encoded atom features and coordinates with the local edges and global edges are fed into the dual equivariant encoders, respectively. Specifically, the local equivariant encoder models the intramolecular force such as the real chemical bonds via local edges while the global equivariant encoder captures the interactive information among distant atoms such as van der Waals force via global edges.

## Conditioned protein spatial information

In addition to the conditioned protein semantic information, we also need to consider the conditioned protein spatial information to ensure the generated ligand can fit the pocket structure without the clash issue. Here, we combine the ligand and protein as the complete pocket as the input of the equivariant kernel. Thus, we construct the local edges and global edges for the input pocket. Specifically, we only construct the edges within the ligand and the edges within the protein to avoid cross-modal distance inference.

$$A^g_{pocket} = \begin{bmatrix} A^g_{ligand} & 0 \\ 0 & A^g_{pocket} \end{bmatrix}, A^l_{pocket} = \begin{bmatrix} A^l_{ligand} & 0 \\ 0 & A^l_{pocket} \end{bmatrix}, \quad (11)$$

where $a_{ij} = (0,1) \in A^g_{pocket}$ and $\tau_l < d_{i,j} \leq \tau_g$ if $a_{ij} = 1$, and $a_{mn} = (0,1) \in A^l_{pocket}$ and $d_{i,j} \leq \tau_l$ if $a_{mn} = 1$. It should be noted that we also remove the self-loop edges to eliminate replicated calculations. By constructing such separate edges, PMDM can perceive the shape of the pocket hole and ensure that the ligand can aggregate the neighborhood information independently via the message-passing process of graph neural networks. Since the pocket spatial information is treated as the condition, we keep the protein position fixed during the update of each layer of the equivariant kernel.

## Equivariant kernel

We employ E(n) Equivariant Graph Neural Networks (EGNN)[12] to achieve the equivariant property. Here, EGNN is equivariant w.r.t the SE(3) group: $\text{EGNN}(\mathbf{AG} + \mathbf{b}) = \mathbf{A}\text{EGNN}(\mathbf{G}) + \mathbf{b}$ where $\mathbf{A}$ is an orthogonal rotation matrix and $\mathbf{b}$ is a translation vector. Here, we concatenate the ligand atom embeddings which already contain the protein semantic information and pocket atom features as $x^0 = [z_L, h_P]$, and the ligand atom coordinates and the protein coordinates as $r_0 = [r_L, r_P]$. Specifically, the equivariant convolution layer takes the node embeddings $\mathbf{x}^l \in \mathbb{R}^{n \times d}$, corresponding coordinate embeddings $\mathbf{r}^l \in \mathbb{R}^{n \times 3}$ and edge information $e_{ij}$ as inputs at layer $l$ and outputs $\mathbf{x}^{l+1}$ and $\mathbf{r}^{l+1}$.

Formally, the updates of node feature and coordinate embeddings of each layer are computed as follows:

$$\mathbf{m}_{ij} = \phi_e\left(\mathbf{h}^l_i, \mathbf{h}^l_j, \mathbf{d}_{ij}, a_{ij}\right), \mathbf{m}_{ij} = e_{att} \cdot \mathbf{m}_{ij}, \hat{\mathbf{m}}_{ij} = |\mathbf{r}_i - \mathbf{r}_j|\phi_m\left(\mathbf{m}_{ij}\right),$$

$$\mathbf{m}_i = \sum_{j \in N(i)} \mathbf{m}_{ij}, \hat{\mathbf{m}}_i = \sum_{j \in N(i)} \hat{\mathbf{m}}_{ij},$$

$$\mathbf{x}^{l+1}_i = \mathbf{x}^l + \phi_x(\mathbf{m}_i),$$

$$\mathbf{r}^{l+1}_i = \begin{cases} \mathbf{r}^l_i + \hat{\mathbf{m}}_i, & \text{if atom } i \in \text{ligand} \\ \mathbf{r}^l_i, & \text{if atom } i \notin \text{ligand} \end{cases}, \quad (12)$$

where $\phi_e$, $\phi_m$, and $\phi_x$ are MLPs, and $a_{ij} = \text{MLP}(\mathbf{d}_{ij})$ is the edge length embedding. $e_{att} = \phi_{inf}(\mathbf{m}_{ij})$ where $\phi_{inf} : \mathbb{R}^{n \times d} \to [0,1]^1$ is to estimate the edge value by an attention mechanism. $m_{ij}$ is the message vector aggregated for atom nodes while $\hat{m}_{ij}$ is the message vector aggregated for edges. Here, we only update the coordinates of ligands to maintain the protein spatial context fixed at each layer of EGNN.

Then only the node embeddings and coordinate embeddings of the ligand part of the final layer are reserved. Finally, we add the outputs of the local equivariant kernel and the global equivariant kernel to obtain the corresponding $s_\theta$:

$$\mathbf{x}'_{local}, \mathbf{r}'_{local} = \text{EGNN}_{local}(\mathbf{G}_{pocket}), \mathbf{x}'_{global}, \mathbf{r}'_{global} = \text{EGNN}_{global}(\mathbf{G}_{pocket}),$$

$$s_\theta(\mathbf{x}) = \mathbf{x}'_{local} + \mathbf{x}'_{global}, s_\theta(\mathbf{r}) = \mathbf{r}'_{local} + \mathbf{r}'_{global},$$

$$s_\theta = [s_\theta(\mathbf{x}), s_\theta(\mathbf{r})] \quad (13)$$

**Training.** The goal of the diffusion model is to learn to reverse the diffusion process. Recall Eq. (6), we also adopt the ELBO objective of the loss function. The differences here are that we have considered the protein context information and converted the $\epsilon_\theta$ to $s_\theta$, thus the loss function becomes:

$$\mathscr{L}_t = \mathbb{E}_{\mathbf{G}^L_0}\left[\gamma \parallel s_\theta\left(\mathbf{G}^L_t, \mathbf{G}^P, t\right) - \nabla_{\mathbf{G}^L_t} \log q_\sigma(\mathbf{G}^L_t|\mathbf{G}^L_0, \mathbf{G}^P)\parallel^2\right]. \quad (14)$$

As shown in Fig. 1c, PMDM sample $t$ from the Uniform distribution for training every iteration. From another perspective, it ensembles $t$ small models to learn the reverse process. Have achieved the equivariance of $s_\theta$, we also need to take this property of the coordinates of $\nabla_{\mathbf{G}^L_t} \log q_\sigma(\mathbf{G}^L_t|\mathbf{G}^L_0)$ into account. Hence, we calculate $\nabla_{d^l_t} \log q_\sigma(d^l_t|d^l_0)$ instead of $\nabla_{r^l_t} \log q_\sigma(r^l_t|r^l_0)$ via the chain rule[60]:

$$\nabla_{\tilde{\mathbf{r}}_i} \log q_\sigma(\tilde{\mathbf{r}}_i \mid \mathbf{r}_i) = \sum_{j \in N(i)} \frac{\nabla_{\tilde{\mathbf{d}}_{ij}} \log q_\sigma(\tilde{\mathbf{d}}_{ij}|\mathbf{d}_{ij}) \cdot \left(\mathbf{r}_i - \mathbf{r}_j\right)}{\mathbf{d}_{ij}}, \quad (15)$$

where $\tilde{\mathbf{r}}$ denotes the diffused atom coordinate $r^L_t$ and $\tilde{\mathbf{d}}$ denotes the corresponding diffused distance. We approximately calculate $\nabla_{\tilde{\mathbf{d}}} \log q_\sigma(\tilde{\mathbf{d}} \mid \mathbf{d})$ as $\frac{-\sqrt{\bar{\alpha}_t}(\tilde{\mathbf{d}} - \mathbf{d})}{1 - \bar{\alpha}_t}$.

Empirically, if $\gamma$ in Eq. (14) is ignored and set as 1 during the training phase, the model performs better than instead with the simplified objective whose $\gamma = \frac{\beta^2_t}{2(1 - \beta_t)(1 - \bar{\alpha}_t)\sigma^2_t}$, which is verified by previous work[29]. Such a simplified objective is equivalent to learning the $s_\theta$ in terms of the gradient of log density of data distribution by sampling the diffused molecule $\mathbf{G}_t$ at a stochastic time step $t$.

**Sampling from scratch.** Since we have formulated the model of $s_\theta$, now we can calculate the $\mu_\theta$ by Eq. (4). As presented in Fig. 1a, the chaotic state $\mathbf{G}_T$ is sampled from $\mathcal{N}(0, I)$ and $\mu_\theta$ is obtained by the dual equivariant encoder, given the target pocket protein. The next less

chaotic state $\mathbf{G}_{T-1}$ is generated by $\mathcal{N}(\mathbf{G}_T; \boldsymbol{\mu}_\theta, \sigma_T^2 I)$. The final molecule $\mathbf{G}_0$ is generated by progressively sample $\mathbf{G}_{t-1}$ for $T$ times. Finally, the atom types of the molecule are identified by adopting the argmax function to choose the atom type which has the largest value while we directly adopt $r_0^t$ outputted by the model. We adopt OpenBabel[61] to build the chemical bonds according to the atom pairwise distances (Fig. 1b). For the generic structure-based molecule generation, we adopt this sample strategy.

**Sampling given specific fragments.** Different from the sampling strategy from scratch which samples the molecule noise from the standard Gaussian distribution, the given fragment information $\mathbf{G}_f$ should be fixed as a seed start point. Here, we adopt a masked strategy to simulate the sampling process from scratch. During each iteration, the seed fragment is masked by the diffusion process according to the corresponding time step,

$$q\left(\mathbf{G}_t^f | \mathbf{G}_0^f\right) = \mathcal{N}\left(\mathbf{G}_t^f; \sqrt{\bar{\alpha}_t}\mathbf{G}_0^f, (1-\bar{\alpha}_t)I\right). \tag{16}$$

The manually diffused fragment is denoised together with the part denoised in the previous step,

$$p_\theta\left(\mathbf{G}_{t-1}^l, \mathbf{G}_{t-1}^f | \mathbf{G}_t^l, \mathbf{G}_t^f\right) = \mathcal{N}\left(\mathbf{G}_{t-1}^l, \mathbf{G}_{t-1}^f; \boldsymbol{\mu}_\theta\left(\mathbf{G}_t^l, \mathbf{G}_t^f, t\right), \sigma_t^2 I\right) \tag{17}$$

We drop the denoised fragment data $\mathbf{G}_{t-1}^f$ and only retain the rest of the denoised part $\mathbf{G}_{t-1}^l$ for the next iteration. The identification of atom types and coordinates is the same as the sampling process from scratch. Finally, we combine the fragment data and the denoised part to obtain the complete molecule by OpenBabel. For lead optimization, we adopt this sample strategy.

**Sampling for linker generation.** In order to generate linkers given specific fragments, we keep the seed fragments fixed to enable the model to be aware of the geometries of fragments. The global edges will connect the distant atoms based on the known positions. In each iteration, the seed fragment is fixed and serves as the context that contains the protein information. The generation part is denoised by conditioning on the seed fragment data,

$$p_\theta\left(\mathbf{G}_{t-1}^l | \mathbf{G}_t^l, \mathbf{G}^f\right) = \mathcal{N}\left(\mathbf{G}_{t-1}^l; \boldsymbol{\mu}_\theta\left(\mathbf{G}_t^l, \mathbf{G}^f, t\right), \sigma_t^2 I\right) \tag{18}$$

The identification of atom types and coordinates is the same as the sampling process from scratch. Finally, we combine the fragment data and the denoised part to obtain the complete molecule by OpenBabel. For linker generation, we adopt this sample strategy.

### Reporting summary

Further information on research design is available in the Nature Portfolio Reporting Summary linked to this article.

## Data availability

The data we used in this study has been deposited in the CrossDocked dataset: https://bits.csb.pitt.edu/files/crossdock2020/. The PDB file of SARS-CoV-2 main protease (M^pro) and Cyclin-dependent Kinase 2 (CDK2) has been deposited in the PDB dataset https://www.rcsb.org/ under accession code https://www.rcsb.org/structure/7l117L11 and https://www.rcsb.org/structure/8H6T8H6T. The data generated in this study and processed training and test data have been publicly deposited to Zenodo under https://zenodo.org/records/10630921. Source data are provided with this paper.

## Code availability

The code of PMDM is freely available at https://github.com/Layne-Huang/PMDM/tree/main. The code is also available at Zenodo (https://zenodo.org/records/10631358).

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

## Acknowledgements

This research was substantially sponsored by the research project (Grant No. 32170654 KC.W.) supported by the National Natural Science Foundation of China and was substantially supported by the Shenzhen Research Institute, City University of Hong Kong. The work described in this paper was substantially supported by the grant from the Research Grants Council of the Hong Kong Special Administrative Region [CityU 11203723 K-C.W.]. The work described in this paper was partially supported by the grants from City University of Hong Kong (2021SIRG036, CityU 9667265, CityU 11203221 K-C.W.) and Innovation and Technology Commission (ITB/FBL/9037/22/S KC.W.). This work is done when Lei Huang works as an intern in Tencent AI Lab.

## Author contributions

Lei Huang developed and implemented the algorithms and wrote the manuscript. Tingyang Xu provided advice on the model development and case studies. Yang Yu provided advice on case studies. Jing Han

designed, supervised the in vitro biochemical assays Hailong Li designed, supervised syntheses of compounds. Tingyang Xu, Yang Yu, Peilin Zhao, Jing Han, Xingjian Chen, Hailong Li, Wenge Zhong and Hengtong Zhang revised the manuscript. Ka-Chun Wong and Hengtong Zhang conceived and supervised the project.

## Competing interests

The authors declare no competing interests.
