## [Peer Review File · Nature Communications]

A dual diffusion model enables 3D molecule generation and lead optimization based on target pocketsReviewer #1 (Remarks to the Author):

The main contribution of this paper is the NN architecture of having separate EGNN parameters for local and distant interactions. The method is interesting and seems to perform better than several recent similar methods. However, the main ideas in the paper are fairly common in the field at this point and the main advance is an incremental gain in metrics. Given this and the lack of wet-lab validation, I think the paper is a better fit for a ML conference than Nature comm. If the lead compounds for SARS-COV2 protease (or some other example application) are synthesized and tested, then I think it is potentially suitable for this journal.

General feedback:

- The description of the method is lacking some basic details, such as how the atom identities in the small molecule are chosen. Are they output by the model, or are these determined by the user and given as inputs?
- There is no experimental data in this paper and it is unclear based on the metrics shown whether the generated molecules could be synthesized and tested in the lab or whether these molecules would actually have high affinities for the target if tested.
- I know it is common in this subfield to evaluate using Vina energy and train on the CrossDocked set, but this seems like a bad practice to me because the CrossDocked dataset is generated by smina (fork of Vina). The model could just learn how to exploit the Vina force-field and not actually make high affinity compounds. Perhaps the authors could explain the rationale for this choice, or otherwise point this out as a limitation of their (and others') analysis. This makes it all the more important to have wet-lab validation (or an alternative in-silico metric or training set)

Specific points:

- It should be stated in the abstract that all the experiments are done in silico.
- In related work, there is another similar 3D equivariant SBDD network recently released (<https://arxiv.org/pdf/2210.13695.pdf>). The authors should mention this work and, ideally, compare to it.
- In table 1, there is lower synthetic accessibility for the molecules generated by the model compared to other models. How many of the molecules could actually be synthesized in the real world? It might be interesting to plot these values only for samples with some minimum synthetic accessibility score.
- Fig. 3b-d has a very difficult-to-see color scheme. Please make the bars different colors, not slightly varying shades of the same color.
- Fig. 3 shows that PMDM tends to generate many more 7-9 atom rings than other methods or the test set. Why is this? Do these rings affect the SA scores?
- In Figure 3d, it is unclear to me what the difference between the capital C and the lower case c dihedral angles is.
- In Figure 5d it almost looks like SA and Vina energy have an inverse correlation. Perhaps plot these 2 variables on their own scatterplot so one can see clearly if there are samples with good vina and SA scores.
- In Figure 5f, I think it would be helpful to highlight the motif in the new scaffolds that are generated by the model.
- In Figure 5g, the selected samples have very low synthetic accessibility and in the plot in figure 5d, it looks like most of the samples with low Vina energy also had low synthetic accessibility. What do samples with high synthetic accessibility look like?
- In Table 2 of the Supplement, the model trained without cross attention between the protein and small molecule seems to perform comparably on the Vina metric and generate more synthetically accessible compounds. It seems that this model is actually potentially worth analyzing further and generating compounds from. Does it provide more low vina, high SA lead compounds for actual test problems?

Reviewer #2 (Remarks to the Author):

Please see attached

Reviewer #2 Attachment on the following page

The authors describe a diffusion model for generation of small-molecule ligands conditioned on a protein binding pocket. They train the model from data of docked protein-ligand complexes. They evaluate their model on a small test set (100 examples) as well as anecdotally with 3 protein test cases. While interesting, the paper does not demonstrate convincingly that the described model is ultimately more useful than current methods. The work would benefit greatly from wet-lab experiments as well as additional computational validation. Please see additional comments below.

It is surprising to see that the authors used such a small test set of 100 protein-ligand pairs. The initial dataset had 22 million pairs; they trained on 100k pairs after filtering for redundancy, and they tested on only 100 pairs. The authors split on sequence similarity, but binding sites of proteins can be highly similar even with only 30% sequence similarity, as binding-site residues are often conserved. It is therefore possible that there is data leakage between train and test sets. The authors might consider evaluating this possibility. Are the ligands in the training set docked ligands that are true binders, are they crystal structures of true binders, or are they just docked ligands of unknown binding affinity? It is not adequately described in the text.

“Furthermore, the generated molecules even touch the disc corner and sphere corner which are not covered by the original test data distribution, indicating that PMDM can not only learn the molecule 3D shape distribution of the dataset but also can explore novel shapes beyond the dataset by importing the random information, which can alleviate the out of distribution (OOD) problems in machine learning “

Are these “novel” molecules physically plausible or synthesizable? The authors do not describe these out of distribution molecules any further.

Do the generated molecules ever clash with the surrounded protein pocket? Do generated ligands dock to the same site (and with the same orientation) via independent docking methods? Do the generated molecules place key functional groups in the same locations as the ligands in the test set? The authors show some coarse-grained analysis for a single protein pocket (Fig 5e), but this analysis would be more informative if done in finer detail for all proteins in the test set. How often do generated ligands recapitulate known protein-ligand interactions from protein-ligand co-crystal structures?

The authors note that their model generated a molecule than might hydrogen bond with different residues in a protein pocket of Mpro. While this is nice to see, it is not evidence that the molecule will actually bind the protein. The authors suggest that binding experiments could be conducted to evaluate affinity, and I strongly recommend that such experiments be carried out if possible. In the absence of experimental validation that the generated molecules are both synthesizable and can bind their target proteins, the authors could choose to dock their compounds using a number of leading docking models, such as DOCK, glide, or diffdock. This might offer some computational consensus that the molecules could plausibly bind their target. Even this however is not true validation that the method can generate a real binder. The ultimate arbiter of truth is a binding experiment in the wet lab.

Reviewer #3 (Remarks to the Author):

The paper proposes a method to generate small ligands binding to some given pocket targets and their corresponding binding conformation. While this is a field with wide interest and high potential impact, I believe the manuscript in its current form is not ready for publication. My main concerns lie with the novelty of the method, ignored related works and experimental evaluation. Moreover several aspects in the paper are unclear and should be clarified. Below, a more detailed description of my concerns.

Novelty of the method and related works: in the past 6 months there has been a number of works using diffusion model for pocket based molecular generation (as an example <https://arxiv.org/abs/2210.13695>). These are very related works with publicly available code the authors should discuss and ideally experimentally compare the very similar approaches to the same task.

Experimental evaluation: the performance of PMDM on the general metrics is strong, especially in comparison to the baselines (although, as mentioned above, stronger baselines could/should be considered). However the reliance on very noisy evaluation criteria such as the Vina Score, calls for further evaluation and the one provided by the authors in the subsequent results sections is not convincing and appears in part flawed.

1. Local geometries: the authors spend significant amount of time discussing the imbalance presence of small rings in other baselines, however from figure 3b it appears that the method widely overestimates the number of 7, 8 and 9 atoms rings. To the best of my knowledge, these rings are also not very stable and uncommon in drug-like molecules, however the authors don't even mention/discuss this aspect.
2. Figure 3a: the types of plots should be changed. Representing a distribution on discrete positive numbers (size of rings) with a continuous plot also going to negative values does not seem appropriate.
3. It is unclear to me how to interpret figures 4abc or how the authors use them to reach the conclusions in the related part of the text (e.g. PMDM correctly models the 2D chemical space). Distributions in Figure 4d (which should be plot as densities not individual points) also do not appear to me to be very similar.
4. Highlighting a lead optimization example is interesting but the authors fall short of truly demonstrating any significant improvement. Being the vina score a noisy and not very reliable metric, molecules with higher score are still unlikely to bind stronger. Affordable and more reliable methods to more reliably test whether any of some subset of these generated hits actually bind are available, for example, free energy perturbation, short MD runs or even synthesizing and testing some of the molecules in a lab.

Method:

1. Is \hat{m}_{ij} a vector? If so why is the norm taken?
2. How is the x (atom types) decoded after the reverse diffusion (e.g. sampling, taking maximum)?
3. what is the form of the noise added to x (e.g. does it preserve it being a distribution over the different atom types or is the one-hot vector just considered as a continuous vector target)?
4. This sentence is unclear "the model performs better than instead with the simplified objective"

Writing:

1. the authors repeatedly refer to their method as one-shot, however given the fact that a diffusion model takes several steps to decode a protein I believe the use of this term when comparing to autoregressive models is misleading (one could argue that autoregressive generation is a particular type of diffusion process where elements are noised one at the time)
2. The abbreviation SBDD is used but never introduced
3. Figure 1 indices G_0 and G_{T-1} seem to be inverted
4. Page 4: ... models on almost [every] metric except ...

Code:

1. The README refers to an environment file "MDM.yml" but I could not find it

Response Letter

Authors' Point-by-Point Responses to the Reviewers' Comments on "A dual diffusion model enables 3D binding bioactive molecule generation and lead optimization given target pockets" by Huang et al. (NCOMMS-23-06531-A) submitted to Nature Communications.

Reviewer #1 (Reviewers' Comments to Author):

The main contribution of this paper is the NN architecture of having separate EGNN parameters for local and distant interactions. The method is interesting and seems to perform better than several recent similar methods. However, the main ideas in the paper are fairly common in the field at this point and the main advance is an incremental gain in metrics. Given this and the lack of wet-lab validation, I think the paper is a better fit for a ML conference than Nature comm. If the lead compounds for SARS-COV2 protease (or some other example application) are synthesized and tested, then I think it is potentially suitable for this journal.

Our response: Thank you very much for your encouraging comments and the constructive advices on improving the manuscript. For clarification of the novelty of this study, we summarize the major technique contributions as follows. First, by introducing three sampling strategies during the inference phase, we have transformed the backbone molecular diffusion model into a versatile and effective framework. This enables a range of tasks, including de novo molecule design, scaffold hopping, and linker generation, whereas previous work focused only on general molecule generation. These tasks have a broad range of real-world applications in drug discovery. Secondly, compared to auto-regressive methods, PMDM could consider the global information to avoid accumulation deviations while it is able to efficiently sample valid molecules with desired lengths. Thirdly, different from the previous diffusion work, the introduction of dual equivariant encoders enables PMDM to construct the global edges and local edges to model different interatomic forces such as van der Waals forces atoms and chemical bonds which can incorporate the molecular topology information. Fourthly, we have considered the protein semantic information to learn high-level protein features. With less parameters, PMDM can sample more chemically valid molecules using less inference time.

The last of which is that, we have conducted wet-lab experiments to validate the generated molecules to follow your suggestion and address your concerns, which we will elaborate on the subsequent responses. Having conducted in vitro experiments, this study is the first to develop deep generative method for structure based drug design verified by wet-lab experiments to the best of our knowledge.

Your comments clearly helped a lot to improve the study. We have summarized the suggested comments (as highlighted) and made point-by-point responses and revisions as follows.

General feedback:

1.1 *The description of the method is lacking some basic details, such as how the atom identities in the small molecule are chosen. Are they output by the model, or are these determined by the user and given as inputs?*

Our response: Thank you for your considerable comments. We utilize the atom identities as inputs and the model also outputs the atom identities as outputs. Specifically, we use one-hot encoding strategy to encode the atom type as discrete inputs X_0^L ; we add noise to the inputs after which we can obtain X_t^L according to the arbitrary t that we sample. During the inference or sampling process, the model would output a continuous vector. The atom identity will be decided by using the argmax function to find the position where the value is the largest. In order to address your concern further, we also revised the Preliminary section, Sampling from scratch section, and Sampling given specific fragments section.

Let $\mathcal{G} = (x, r)$ denote the 3D molecular geometry where $x = (x_1, x_2, \dots, x_n) \in \{0,1\}^{n \times f}$ denotes the **discrete** one-hot encoded atom **types (a.k.a, chemical elements)**, and $r = (r_1, r_2, \dots, r_n) \in \mathbb{R}^{n \times 3}$ denotes the **continuous** atom coordinates as depicted in Figure 1c.

The next less chaotic state \mathcal{G}_{T-1} is generated by $\mathcal{N}(\mathcal{G}_T; \mu\theta, \sigma_T^2 I)$. The final molecule \mathcal{G}_0 is generated by progressively sample $\mathcal{G}_t - 1$ for T times. **Finally, the atom types of the molecule are identified by adopting the argmax function to choose the atom type which has the largest value while we directly adopt r_0^L outputted by the model.** We adopt OpenBabel to build the chemical bonds according to the atom pairwise distances (Figure 1b). For the generic structure-based molecule generation, we adopt this sample strategy.

We drop the denoised fragment data \mathcal{G}^f and only retain the rest of the denoised part $\mathcal{G}^l t - 1$ for the next iteration. **The identification of atom types and coordinates is the same as the sampling process from scratch.** Finally, we combine the fragment data and the denoised part to obtain the complete molecule by OpenBabel. For lead optimization, we adopt this sample strategy.

1.2 *There is no experimental data in this paper and it is unclear based on the metrics shown whether the generated molecules could be synthesized and tested in the lab or whether these molecules would actually have high affinities for the target if tested.*

Our response: Thank you very much for your constructive insights. In order to validate whether the generated molecules can be synthesized and have high affinities with the target pocket, we have designed another case study to generate molecules for CDK2. We selected one of the inhibitors complexed with Cyclin-dependent Kinase 2 (CDK2) (PDB ID: 8H6T) to develop potential inhibitors by performing scaffold hopping using PMDM. The transition from G1 to S phase is driven by CDK2 in complex with its canonical partner cyclin E1 (CCNE1), which is often amplified in various cancers and is associated with worse survival outcomes in patients with breast, ovarian, and other malignancies^{51,52}. Therefore, CDK2 is a potential cancer therapy target with abnormal levels or activity of many tumors. However, there is only a limited number of selective CDK2 inhibitors active in clinical trials. Scaffold hopping which is a topic of high interest in medicinal chemistry refers to the computer-aided design of active compounds with novel core structures. This strategy is crucial for drug development, as it can overcome challenges such as patent issues, toxicity, metabolic instability, or resistance that may affect known active compounds. Moreover, scaffold hopping can enable the exploration of new chemical space and the discovery of new pharmacophores or mechanisms of action. Scaffold hopping methods generally involve modifying the central core structure of known active compounds to generate novel chemotypes.

Specifically, we have conducted two scaffold hopping cases which include the modifications on the reference scaffold and linker generation based on the reference compound and selected two molecules for wet-lab experiments. We have added one section (**PMDM enables scaffold hopping for real synthetic bioactive molecule generation**) to elaborate on the lead optimization cases towards CDK2, which is presented in the following: Scaffold hopping, a topic of high interest in medicinal chemistry, refers to the computer-aided design of active compounds with novel core structures.

“PMDM enables scaffold hopping for real synthetic bioactive molecule generation

Scaffold hopping is very important with appropriate hit compounds in lead optimization since it could not only generate known active scaffolds and improve binding affinity but also identify novel core structures that confer improved properties to overcome challenges in ADME/Tox profiles⁴⁹. The development of advanced methods for making, analyzing, and purifying molecules of drug-like size, has made it possible to synthesize analogs based on a common scaffold, along with the higher and more widespread access to commercial building blocks⁵⁰. In order to validate whether our model could be

applied in scaffold hopping to improve the binding affinities of the given basic bioactive molecule, we select Cyclin-dependent Kinase 2 (CDK2) as the target protein to generate desired molecules with core structures. The transition from G1 to S phase is driven by CDK2 in complex with its canonical partner cyclin E1 (CCNE1), which is often amplified in various cancers and is associated with worse survival outcomes in patients with breast, ovarian, and other malignancies^{51,52}. Therefore, CDK2 is a potential cancer therapy target with abnormal levels or activity of many tumors. However, there is only a limited number of selective CDK2 inhibitors active in clinical trials⁵³.

Following our previous study⁵³, we utilize PMDM to perform scaffold hopping on one of the inhibitors complexed with CDK2 (PDB ID: 8H6T) to develop potential inhibitors. The reference compound is illustrated in Figure 6a. The aminopyrazole moiety of the reference compound forms two hydrogen bonds with residues LEU83 and GLU81 and the carboxyl of the compound forms one hydrogen bond with the residue LYS 33. Besides, the gatekeeper residue's phenyl side chain has van der Waals interactions with the cyclopentyl ring of the compound. The pyridine moiety of the compound is oriented towards the solvent-accessible region and does not exhibit any significant polar or nonpolar contacts with CDK2. We remove the pyridine ring and reserve the remaining fragment as the seed scaffold (Figure 6a). Finally, we leverage PMDM to generate a library of 10000 molecules for relating the essential hinge-binding elements. Then the potential inhibitors were filtered through Vina docking and MM-PBSA values with visual selection, and we selected six compounds for visual inspection. As illustrated in Figure 6b, the partial molecules generated from the seed scaffold exhibit heterocyclic structures, suggesting that they undergo modification at the site of the eliminated pyridine ring. Besides, atoms are also generated in other sites of the fragment to form varied rings or functional groups. All the potential inhibitors exhibit higher Vina scores and MM-PBSA values with suitable SA scores. We select the compound 9021 with the lowest MM-PBSA value to assay its CDK1/2 inhibitory activities through in vitro experiments. As reported in Figure 6b, the compound 9021 displayed better CDK2 activity in enzyme assay with around 490-fold CDK1 selectivity.

The reference compound exhibits a U-shaped conformation with the 6-position carbon of the pyridine ring and the nitrogen atom of the carbamate moiety oriented towards each other. The interatomic distance between the two atoms is determined to be 5.2 Å, offering logical connection points for macrocyclization. Given that our model facilitates ring formation via global edge construction, we investigated the potential of our model to generate linker molecules for macrocyclization. Unlike the specific fragments strategy, which utilizes the sampling given specific fragments strategy, we fix the seed fragment to enable the model to be aware of the fragment geometries (See section Sampling for linker generation). This could help the model generate linkers which connect the fragments coherently (Figure 6c). The effect of linker length on the

pharmacological properties of the reference compound is examined by medicinal chemists employing structure-based drug design approaches. Therefore, we explore linkers ranging from 4 to 6 atoms in length. We fix the connecting points of the linker at the pyridine ring and the nitrogen atom of the carbamate motif. Finally, we utilize PMDM to generate 5000 macrocycles for the reference compound with the preset attachment points. We selected five potential macrocyclized inhibitors for visual inspection after filtering through Vina docking and MM-PBSA values with visual selection. As illustrated in Figure 6c, PMDM successfully generated the linkers which connect the preferred attachment points although we do not train PMDM on the specific linker datasets. The generated linkers improve the Vina score of the reference compound while retaining similar biological activity. We selected one of the potential inhibitors which have better MM-PBSA values to investigate its in vitro results. The ADME profiles which are reported in our previous work⁵³ suggest that this compound 16 shows potent subnanomolar CDK2 activities and two-digit nanomolar activity towards CDK1, indicating high CDK2 selectivity (around 49 folds).

Figure 6. Scaffold hopping case of Cyclin-dependent Kinase 2 (CDK2). a. The complex structure of the inhibitor Compound 13 targeting CDK2 with pharmacochemical properties. b. The generated molecules with desired properties by the scaffold hopping strategy. We selected one molecule for wet-lab experiments and presented the IC50 values for CDK2/E1 and CDK1/A2. c. The generated macrocyclic CDK2 inhibitor by the linker generation method. We also selected one molecule for wet-lab experiments and presented the IC50 values for CDK2/E1 and CDK1/A2.

Reference

1. Wang, L. et al. Accurate modeling of scaffold hopping transformations in drug discovery. *J. chemical theory computation*, 13, 42–54 (2017).
2. Zhao, H. Scaffold selection and scaffold hopping in lead generation: a medicinal chemistry perspective. *Drug discovery today* 12, 149–155 (2007).
3. Tadesse, S., Caldon, E. C., Tilley, W. & Wang, S. Cyclin-dependent kinase 2 inhibitors in cancer therapy: an update. *J. medicinal chemistry* 62, 4233–4251 (2018).
4. Sokolsky, A. et al. Discovery of 5, 7-dihydro-6 h-pyrrolo [2, 3-d] pyrimidin-6-ones as highly selective cdk2 inhibitors. *ACS Medicinal Chem. Lett.* 13, 1797–1804 (2022).
5. Yu, Y. et al. Accelerated discovery of macrocyclic cdk2 inhibitor qr-6401 by generative models and structure-based drug design. *ACS Medicinal Chem. Lett.* 14, 297–304 (2023).

“

1.3 *I know it is common in this subfield to evaluate using Vina energy and train on the CrossDocked set, but this seems like a bad practice to me because the CrossDocked dataset is generated by smina (fork of Vina). The model could just learn how to exploit the Vina force-field and not actually make high affinity compounds. Perhaps the authors could explain the rationale for this choice, or otherwise point this out as a limitation of their (and others') analysis. This makes it all the more important to have wet-lab validation (or an alternative in-silico metric or training set)*

Our response: Thank you for your considerable comments. We agree with you that only training the model on CrossDocked dataset cannot fully learn to generate molecules with high affinities with target protein pockets. Indeed, we have adopted CrossDocked dataset just to follow others for a fair comparison since most current the-state-of-the-art models adopt this dataset for experiments. In other words, it is a wide-used benchmark dataset, which is suitable for comparing the performances of deep generative methods. Besides, the data size of this dataset is sufficient for deep

generative learning training since this dataset is developed from PDB structures. The original PDBbind dataset only have **23,496** biomolecular complexes in PDB.

To address your concern, we pretrained PMDM on CrossDocked dataset and fine tune it on PDBbind dataset and reconstruct our code for building the molecule according to the atom types and atom coordinates. We further utilized well-trained PMDM to generate molecules for SARS-COVID-19 main protein and CDK2 which we validated the molecule by wet-lab experiments.

We plot the distribution of Vina, QED and SA score of the new generated molecules for SARS-COVID-19 main protein and present the figures of previous generated molecules. We can observe that the new generated molecules have higher SA and reasonable QED value.

Furthermore, we have conducted wet-lab experiments for CDK2 and generated molecules which are synthetic and active towards to the protein, indicating that our model can learn the bioactive structures and geometries from CrossDocked dataset.

Specific points:

1.4 *It should be stated in the abstract that all the experiments are done in silico.*

Our response: Thank you for your valuable comment. Most of the experiments were conducted *in silico* except for the scaffold hopping of CDK2. To follow your suggestion, we have revised the manuscript to state that most the experiments are done *in silico*.

We have conducted comprehensive *in silico* experiments to demonstrate that PMDM can generate drug-like, synthesis-accessible, novel, and high-binding affinity molecules targeting specific proteins, outperforming the state-of-the-art (SOTA) models in terms of multiple evaluation metrics. In addition, we perform chemical space analysis for generated molecules and lead compound optimization for SARS-CoV-2 main protease M^{Pro} and Cyclin-dependent Kinase 2 (CDK2) which we have synthesized the selected

molecules and evaluated their in vitro activities towards the targeting proteins. The *in silico* and wet lab experimental results implicate that the structures of generated molecules are rational compared to the reference molecules, and PMDM can generate massive bioactive molecules highly binding to the targeted proteins which are not included in the training set and can be synthesized.

1.5 *In related work, there is another similar 3D equivariant SBDD network recently released (<https://arxiv.org/pdf/2210.13695.pdf>). The authors should mention this work and, ideally, compare to it.*

Our response: Thank you for your constructive suggestions. DiffSBDD is also based on a diffusion model that could generate molecules given a specific project structure. Following your suggestion, we have read DiffSBDD with interests and found it very insightful; therefore, we have revised the Related Work section to incorporate the descriptions of DiffSBDD and compared it to PMDM in Table 1.

Although there is another diffusion model³⁵ developed for structure-based molecule generation, however, it requires training the user-defined parameters, leading to inefficient sampling. Besides, it only utilizes the fully connected adjacent matrix thus can ignore parts of the intrinsic topology in molecular graph.

The results in Table 1 indicate that our model can generate molecules with high binding affinities, QED score and SA score with less sampling time.

Table 1. The comparison of 10000 generated molecules of PMDM and baseline models on the CrossDocked dataset. We generate 100 samples for each target pocket.

Methods	Vina Score (kcal/mol)	High Affinity	QED	SA	Lipinski rules	LogP	Diversity	Time (Seconds)
CVAE	-6.144±1.57	0.238	0.369±0.22	0.590±0.15	4.367±1.14	-0.140±2.73	0.654±0.12	-
AR-SBDD	-6.215±1.54	0.267	0.502±0.17	0.675±0.14	4.787±0.50	0.257±2.01	0.742±0.09	19659±14704
DiffSBDD	-6.584±2.06	-	0.495±0.15	0.366±0.09	4.795±0.49	-	0.730±0.11	1634±769
PMDM	-7.572±2.50	0.628	0.594±0.12	0.611±0.16	4.975±0.16	0.301±1.01	0.709±0.10	906±1097
Test Set	-7.024	-	0.466	0.725	1.413	0.929	-	-

1.6 *In table 1, there is lower synthetic accessibility for the molecules generated by the model compared to other models. How many of the molecules could actually be synthesized in the real world? It might be interesting to plot these values only for samples with some minimum synthetic accessibility score.*

Our response: Thank you for your thoughtful comments. We achieve comparable synthetic accessibility to AR-SBDD and better synthetic accessibility than CVAE and

DiffSBDD. The lengths of molecules generated by auto-regressive models are determined by the terminal node classifier, which predicts whether the current node is the final atom. Thus, auto-regressive models may generate small molecules which have high synthetic accessibilities. Compared to another diffusion model DiffSBDD, our model PMDM achieves much higher performance on SA scores.

We agree with you that it is very important to verify whether the molecule can be synthesized by real wet-lab experiments. Unfortunately, we currently do not have the experimental conditions and sufficient raw materials to synthesize molecules. This paper mainly focuses on *in silico* experiments for virtual screening of the generated molecules. In order to address your concerns, we generate molecules for CDK2 which we will synthesize the molecule by wet-lab experiments since we only have the experimental condition for this target. We have added one section (**PMDM enables scaffold hopping for real synthetic bioactive molecule generation**) to elaborate on the lead optimization cases towards CDK2.

1.7 *Fig. 3b-d has a very difficult-to-see color scheme. Please make the bars different colors, not slightly varying shades of the same color.*

Our response: Thank you for your considerable comments. We have changed the colours for a better presentation. We also have incorporated the results of DiffSBDD in Figure 3b, Figure 3c and Figure 3d.

1.8 Fig. 3 shows that PMDM tends to generate many more 7-9 atom rings than other methods or the test set. Why is this? Do these rings affect the SA scores?

Our response: Thank you for your considerable comment. Actually, PMDM generates as few 8-atom rings and 9-atom rings as other methods. Specifically, PMDM only generates 6.1% of molecules with 8-atom rings while CVAE generates 1.4%, AR-SBDD generates 0.7%, and DiffSBDD generates 5.7% of such molecules. For 9-atom rings, all the models generate 0.6% molecules except DiffSBDD (2.5%) containing such big rings. For 3-atom rings, PMDM only generates 2% while CVAE generates 36.1%, AR-SBDD generates 48.4%, and DiffSBDD generates 44.4% of such molecules. PMDM does generate relatively more 7-atom rings since we construct the global edges which may connect distant atoms, thus, our model also generates more 6-atom rings than other methods but generating the big rings would sacrifice the SA scores. On the other hand, PMDM generates fewer molecules which contain 8-atom and 9-atom rings compared to

DiffSBDD since PMDM constructs the local edges to consider close atomic forces while DiffSBDD only constructs the fully connected edges to incorporate many distant atomic interactions.

1.9 In Figure 3d, it is unclear to me what the difference between the capital C and the lower case c dihedral angles is.

Our response: Thank you for your considerable suggestions. Lower-case c means the atom is in the aromatic ring. And we apologize that there is a typo error in Figure 3d, where 'Ccc' should be 'Cccc'.

1.10 In Figure 5d it almost looks like SA and Vina energy have an inverse correlation. Perhaps plot these 2 variables on their scatterplot so one can see clearly if samples have good vina and SA scores.

Our response: Thank you for your constructive comment. Following your suggestion, we plot the distributions of SA and Vina scores in one figure. The figure demonstrates that SA and Vina energy do not have an inverse correlation. The molecules with good binding affinities may have reasonable SA scores. We have calculated the Pearson's correlation coefficient value (-0.21), which suggests a negligible inverse relationship between Vina and SA scores.

We also plot the distributions of SA and Vina scores in the following figure. It indicates that QED and Vina energy also do not have a strong inverse correlation. Similarly, the

Pearson's correlation coefficient value (-0.13) indicates a negligible inverse relationship between Vina and QED values.

1.11 In Figure 5f, I think it would be helpful to highlight the motif in the new scaffolds that are generated by the model.

Our response: Thank you for your valuable comments. Following your suggestion, we highlight the seed atoms of the new scaffolds in Figure 5f for a clear presentation. The remaining part is the motif that is generated by the model.

1.12 In Figure 5g, the selected samples have very low synthetic accessibility and in the plot in figure 5d, it looks like most of the samples with low Vina energy also had low synthetic accessibility. What do samples with high synthetic accessibility look like?

Our response: Thank you for your valuable comment. The molecules generated that are relatively long with rings, leading to relatively low synthetic accessibility. By finetuning on PDBbind dataset and reconstructing our code for building the molecule, PMDM can generate molecules with higher SA scores. We presented the samples with high synthetic accessibilities to address your concerns.

1.13 *In Table 2 of the Supplement, the model trained without cross attention between the protein and small molecule seems to perform comparably on the Vina metric and generate more synthetically accessible compounds. It seems that this model is actually potentially worth analyzing further and generating compounds from. Does it provide more low Vina, high SA lead compounds for actual test problems?*

Our response: Thank you for your constructive comment. We utilize cross-attention to enable the model to incorporate enriched protein information. Therefore, the model with co-attention module can generate molecules with higher affinities. On the other hand, the model without co-attention mechanism would pay more attention to the molecule itself and can generate molecules with higher synthetic accessibilities. To address your concerns, we utilized PMDM without cross-attention mechanism to generate 40000 molecules for SRAR-Covid-19 main protein. We present the results of PMDM and the PMDM without cross-attention (PMDM w.o. CA) in the following figures.

Figure S1. Results comparison of PMDM and PMDM w.o. CA for generating 40000 molecules for SARS-CoV-2 main protease (M^{pro}).

We can observe that PMDM exceeds PMDM without cross-attention on SA and Vina. Since PMDM without cross-attention pay more attention to the molecule itself, it achieves higher QED than PMDM. Regarding logP value, both PMDM and PMDM w.o. CA could generate most molecules whose logP values are within the reasonable range. Finally, the molecules generated from PMDM can obey more Lipinski rules compared to PMDM w.o. CA. We added this analysis to the supplementary file to address your concern.

“Since PMDM without CA performs better than PMDM on QED and SA, we further investigate whether PMDM without CA can generate molecules binding to the SARS-CoV-2 main protease (M^{pro}) with high SA scores and rational medicinal property values. We also generate 40000 molecules and calculate their Vina, SA, QED, logP and Lipinski rules to compare with PMDM. As demonstrated in Figure 1, we found that PMDM with CA outperformed PMDM without CA on Vina and SA, indicating that it can generate more potent and feasible inhibitors for M^{pro} . On the other hand, PMDM without CA achieved higher QED scores than PMDM with CA, suggesting that it can generate more drug-like molecules. However, both methods generated most molecules with reasonable logP values, and PMDM with CA generated more molecules that satisfied the

Lipinski rules than PMDM without CA. Therefore, we conclude that PMDM with CA is a more effective method for generating molecules that can target M^{pro}.”

Thank you again for your insightful comments and we look forward to hearing from you regarding our revised manuscript.

Review #2

The authors describe a diffusion model for generation of small-molecule ligands conditioned on a protein binding pocket. They train the model from data of docked protein-ligand complexes. They evaluate their model on a small test set (100 examples) as well as anecdotally with 3 protein test cases. While interesting, the paper does not demonstrate convincingly that the described model is ultimately more useful than current methods. The work would benefit greatly from wet-lab experiments as well as additional computational validation. Please see additional comments below.

Our response: Thank you very much for your detailed and insightful comments on our manuscript. Your comments clearly helped a lot to improve this manuscript! To address your concerns, we have conducted additional *in silicon* experiments with more test samples and wet-lab experiments to validate the generated molecules which we will elaborate on the subsequent responses. With your precious comments, this study is the first to develop deep generative method for structure based drug design verified by wet-lab experiments to the best of our knowledge. Compared to current methods, PMDM can efficiently and accurately generate drug-like, synthesis-accessible, novel molecules with high binding affinity against specific proteins using less sampling time. Moreover, PMDM is a versatile framework which can be generalized to lead optimization including scaffold hopping and linker generation. We have summarized your comments and made point-by-point responses and revisions to address your concerns.

2.1 *It is surprising to see that the authors used such a small test set of 100 protein-ligand pairs. The initial dataset had 22 million pairs; they trained on 100k pairs after filtering for redundancy, and they tested on only 100 pairs. The authors split on sequence similarity, but binding sites of proteins can be highly similar even with only 30% sequence similarity, as binding-site residues are often conserved. It is therefore possible that there is data leakage between train and test sets. The authors might consider evaluating this possibility. Are the ligands in the training set docked ligands that are true binders, are the crystal structures of true binders, or have they just docked ligands of unknown binding affinity? It is not adequately described in the text.*

Our response: Thank you for your valuable comments. To follow previous works for a fair comparison setting, we utilize the same test set. We report the performances of all the models from their paper which are evaluated on the same test set.

Indeed, we agree with you that binding-site residues are often conserved, but our model not only focuses on the binding sites but also learns the information from the neighbour residues which may affect the whole pocket shape. Specifically, we leverage our model to generate molecules based on the whole pocket structure with spatial information and semantic information. Thus, in the context of our model in global settings, the similarity of 30% on the protein primary structure can already avoid data

leakage while other competing methods could be given advantages in a generous manner though.

In order to address your concerns further, we retrained the model on 99k samples (8 V100 GPU cards for 3 days) and evaluated the performance on 1k samples with only 20% sequence similarity. We generate 100 molecules for each protein in the test set, which takes around 4 days. The results are presented in the following table. PMDM still generates molecules with high affinities with specific proteins.

Table S3. The results of molecules generated by PMDM on 1K test pairs.

Methods	Vina Score (kcal/mol)	QED	SA	Lipinski rules	LogP
PMDM	-7.473±2.24	0.597±0.13	0.627±0.14	4.977±0.15	0.283±0.99
Test Set	-7.831	0.533	0.747	4.574	1.648

We put this result in the supplementary since we directly report the performances of SOTA models from their papers for a fair comparison.

The CrossDocked dataset was generated from the PDB structures specified by Pocketome v17.12. Pocketome collects conformational ensembles of druggable binding sites that can be identified experimentally from co-crystal structures in the Protein Data Bank. The ligands associated with a given pocket were then docked into each receptor assigned to that pocket by Pocketome using smina. Binding data (pK) for the CrossDocked2020 set was taken from PDBbind v2017 and 41.9% of the complexes have binding affinity data. We thank you again for pointing it out. Following your suggestion, we have added more description about this dataset in section Data.

We conduct experiments to evaluate the generative performance of PMDM on the CrossDocked dataset2020. This dataset contains 22.5 million docked protein-ligand pairs and each pair has different poses to multiple pockets across the Protein Data Bank. The ligands that were associated with a specific pocket were subsequently subjected to docking with each receptor assigned to that particular pocket by utilizing smina through Pocketome. The binding data (pK) for the CrossDocked2020 set was obtained from PDBbind v2017, and it was observed that 41.9% of the complexes have available binding affinity data.

2.2 *“Furthermore, the generated molecules even touch the disc corner and sphere corner, which are not covered by the original test data distribution, indicating that PMDM can not only learn the molecule 3D shape distribution of the dataset but also can explore novel shapes beyond the dataset by importing the random information, which can alleviate the out of distribution (OOD) problems in machine learning .”Are*

these “novel” molecules physically plausible or synthesizable? The authors do not describe these out-of-distribution molecules any further.

Our response: Thank you very much for your constructive comments. We define the generated molecules which touch the disc corner and sphere corner that are not covered by the original test data distribution as out of distribution (OOD) molecules. Here, we point out that our model can alleviate the OOD problems in machine learning since the generated molecules can cover more areas in the 3D shape distribution, which means that our models has the potential to generate molecules binding with the protein which does not obey the distribution of the proteins in the training set. We calculate the mean SA score for the out-of-distribution molecules: 0.628 ± 0.28 , which is a reasonable value. Other chemical property values are listed in the following table. To further address your concern, we added the discussion about the out-of-distribution molecules in the main document.

Table S3. The chemical property values of OOD molecules.

QED	SA	Lipinski rules	LogP
0.448 ± 0.16	0.628 ± 0.19	4.713 ± 0.73	1.065 ± 2.22

Furthermore, the generated molecules even touch the disc corner and sphere corner which are not covered by the original test data distribution, indicating that PMDM can not only learn the molecule 3D shape distribution of the dataset but also can explore novel shapes beyond the dataset by importing the random information, which can alleviate the out of distribution (OOD) problems in machine learning. **In other words, PMDM has the potential to generate more novel generate molecules even facing proteins which do not obey the distributions of the proteins in the training set. We further calculate SA of these out-of-distribution molecules and the mean SA value is 0.628 ± 0.29 , which is higher than the average SA value of the whole generated molecules, indicating that these novel molecules are computationally synthesizable. Besides, the molecules also achieve reasonable values on other chemical properties.**

2.3 *Do the generated molecules ever clash with the surrounding protein pocket? Do generated ligands dock to the same site (and with the same orientation) via independent docking methods? Do the generated molecules place key functional groups in the same locations as the ligands in the test set?*

Our response: Thank you for your valuable comments. We considered the spatial context information to ensure the generated ligand can fit the pocket structure to avoid clash issues. Specifically, we combine the ligand and protein as the complete pocket as the input of the equivariant kernel. Then we construct the local edges and global edges for the ligand and protein themselves in the input pocket. We do not construct the

edges to connect the atoms in the ligand and protein to avoid cross-modal distance inference, in which most edges are noisy.

Since the pocket spatial information is treated as the condition, we keep the protein position fixed during the update of each layer of the equivariant kernel. The fixed protein positions can ensure the relative distance between the atoms of ligand and atoms of proteins, which can avoid the ligand clashes with the protein. We show some examples in the following figures.

The generated molecules shown in the left part of the figures are inclined to clash with the protein surface, although we initialize the ligand position in the center of the protein. When we treat the ligand and the protein as a whole pocket to input to the equivariant graph neural network and keep the protein positions fixed, the model could learn the relative distances between ligands and proteins. The generated molecules do not prone to clash with the protein surfaces, which are shown in the right part of the figure. To further address your concern, we added this discussion in the supplementary.

The generated ligands do not necessarily dock to the same binding site with the same orientations with independent docking methods. Different docking methods have different searching algorithms and scoring functions. We utilize QuickVina2¹ which is based on AutoDock-Vina to dock the generated ligand. This **known-pocket docking method** receives as input the position on the protein where the molecule will bind, which is similar to Glide², MOE DOCK³ and rDock⁴. We define the search space by using the centre of mass of the ligand as the center and choosing 20 Å as the size. Conversely, DiffDock does not assume any prior knowledge about the binding pocket, which could be called **search-based docking method**⁵. Since our model generates molecules which contain the coordinates in the pocket given the pocket structure, we only consider using **known-pocket docking methods** rather than **search-based docking methods** to perform docking. The previous work⁶ proved that Autodock-Vina achieves the best performance among the known-pocket docking methods. QuickVina is a fast and accurate molecular docking tool attained at accurately accelerating AutoDock-Vina. Therefore, comparing the binding affinity by utilizing QuickVina to dock the generated ligands is fair and accurate compared to searching the distant binding sites.

The generated molecules may not place the key functional groups in the same locations as the ligands in the test set. Our model learns the molecular geometries given the protein structures with imported random information which could improve the diversity of the generated samples. The generated molecules can fit the shape of the protein pocket well. However, we do not implement the constraint to enforce the model learns which function group is essential to bind with the binding sites. This constrained generation requires accurate annotations for each pair of crystal complexes, and the model has to be aware of the important functional groups and binding sites. Although our model cannot ensure that it could directly generate the same function group as the test molecule, PMDM could reserve the desired function group in another way. PMDM can fix the desired functional group then generate the remaining fragments of the molecule, which the molecule could be sampled following the algorithm 3 in the supplementary. Give the seed fragment or the key functional group which we would like to reserve, we defuse the seed fragment with the remaining part which is sampled from a standard Gaussian distribution, renoise the noisy remaining part, and concatenate the seed fragment and the denoised remaining part to obtain the final molecule. This final molecule could reserve the fragment or the key functional group which are important to the binding affinities.

Reference

1. Alhossary, A., Handoko, S. D., Mu, Y., & Kwoh, C. K. (2015). Fast, accurate, and reliable molecular docking with QuickVina 2. *Bioinformatics*, 31(13), 2214-2216.
2. Friesner, R. A., Banks, J. L., Murphy, R. B., Halgren, T. A., Klicic, J. J., Mainz, D. T., ... & Shenkin, P. S. (2004). Glide: a new approach for rapid, accurate docking and scoring. 1. Method and assessment of docking accuracy. *Journal of medicinal chemistry*, 47(7), 1739-1749.

3. Ruiz-Carmona, S., Alvarez-Garcia, D., Foloppe, N., Garmendia-Doval, A. B., Juhos, S., Schmidtke, P., ... & Morley, S. D. (2014). rDock: a fast, versatile and open source program for docking ligands to proteins and nucleic acids. *PLoS computational biology*, 10(4), e1003571.
4. Vilar, S., Cozza, G., & Moro, S. (2008). Medicinal chemistry and the molecular operating environment (MOE): application of QSAR and molecular docking to drug discovery. *Current topics in medicinal chemistry*, 8(18), 1555-1572.
5. Corso, G., Stärk, H., Jing, B., Barzilay, R., & Jaakkola, T. (2022). Diffdock: Diffusion steps, twists, and turns for molecular docking. *arXiv preprint arXiv:2210.01776*.
6. Wang, Z., Sun, H., Yao, X., Li, D., Xu, L., Li, Y., ... & Hou, T. (2016). Comprehensive evaluation of ten docking programs on a diverse set of protein–ligand complexes: the prediction accuracy of sampling power and scoring power. *Physical Chemistry Chemical Physics*, 18(18), 12964-12975.

2.4 *The authors show some coarse-grained analysis for a single protein pocket(Fig 5e), but this analysis would be more informative if done in fine detail for all proteins in the test set. How often do generate ligands recapitulate known protein-ligand interactions from protein-ligand co-crystal structures?The authors note that their model generated a molecule than might hydrogen bond with different residues in a protein pocket of Mpro. While this is nice to see, it is not evidence that the molecule will actually bind the protein. The authors suggest that binding experiments could be conducted to evaluate affinity, and I strongly recommend that such experiments be carried out if possible. In the absence of experimental validation that the generated molecules are both synthesizable and can bind their target proteins, the authors could choose to dock their compounds using a number of leading docking models, such as DOCK, glide, or diffdock.This might offer some computational consensus that the molecules could plausibly bind their target. Even this however is not true validation that the method can generate a real binder. The ultimate arbiter of truth is a binding experiment in the wet lab.*

Our response: Thank you for your constructive comments. We perform the experiments of Figure 5 to demonstrate that PMDM could be utilized for lead optimization to generate the drug candidates or hit compounds. We utilize the pharmacophore model to evaluate whether the molecules that conserve the key seed fragment are able to form the hydrophobic groups as the reference molecule. This in silicon experiment is conducted case by case, which requires fine-grained annotation or information to analyse the binding mechanism. Unfortunately, we do not have the binding annotations and quantitative metrics to calculate for all protein pockets in the test set. In order to prove that generated molecules could actually bind to the specific proteins, we conduct scaffold hopping towards CDK2. We selected one of the inhibitors complexed with CDK2 (PDB ID: 8H6T) to develop potential inhibitors by performing scaffold hopping using

PMDM. The transition from G1 to S phase is driven by CDK2 in complex with its canonical partner cyclin E1 (CCNE1), which is often amplified in various cancers and is associated with worse survival outcomes in patients with breast, ovarian, and other malignancies^{51,52}. Therefore, CDK2 is a potential cancer therapy target with abnormal levels or activity of many tumors. However, there is only a limited number of selective CDK2 inhibitors active in clinical trials. Scaffold hopping which is a topic of high interest in medicinal chemistry refers to the computer-aided design of active compounds with novel core structures. This strategy is crucial for drug development, as it can overcome challenges such as patent issues, toxicity, metabolic instability, or resistance that may affect known active compounds. Moreover, scaffold hopping can enable the exploration of new chemical space and the discovery of new pharmacophores or mechanisms of action. Scaffold hopping methods generally involve modifying the central core structure of known active compounds to generate novel chemotypes.

Specifically, we have conducted two scaffold hopping cases which include the modifications on the reference scaffold and linker generation based on the reference compound and selected two molecules for wet-lab experiments. We have added one section (**PMDM enables scaffold hopping for real synthetic bioactive molecule generation**) to elaborate on the lead optimization cases towards CDK2, which is presented in the following:

“PMDM enables scaffold hopping for real synthetic bioactive molecule generation

Scaffold hopping is very important with appropriate hit compounds in lead optimization since it could not only generate known active scaffolds and improve binding affinity but also identify novel core structures that confer improved properties to overcome challenges in ADME/Tox profiles⁴⁹. The development of advanced methods for making, analyzing, and purifying molecules of drug-like size, has made it possible to synthesize analogs based on a common scaffold, along with the higher and more widespread access to commercial building blocks⁵⁰. In order to validate whether our model could be applied in scaffold hopping to improve the binding affinities of the given basic bioactive molecule, we select Cyclin-dependent Kinase 2 (CDK2) as the target protein to generate desired molecules with core structures. The transition from G1 to S phase is driven by CDK2 in complex with its canonical partner cyclin E1 (CCNE1), which is often amplified in various cancers and is associated with worse survival outcomes in patients with breast, ovarian, and other malignancies^{51,52}. Therefore, CDK2 is a potential cancer therapy target with abnormal levels or activity of many tumors. However, there is only a limited number of selective CDK2 inhibitors active in clinical trials⁵³.

Following our previous study⁵³, we utilize PMDM to perform scaffold hopping on one of the inhibitors complexed with CDK2 (PDB ID: 8H6T) to develop potential inhibitors. The reference compound is illustrated in Figure 6a. The aminopyrazole moiety of the reference compound forms two hydrogen bonds with residues LEU83 and GLU81 and the carboxyl of the compound forms one hydrogen bond with the residue LYS 33. Besides, the gatekeeper residue's phenyl side chain has van der Waals interactions with the cyclopentyl ring of the compound. The pyridine moiety of the compound is oriented towards the solvent-accessible region and does not exhibit any significant polar or nonpolar contacts with CDK2. We remove the pyridine ring and reserve the remaining fragment as the seed scaffold (Figure 6a). Finally, we leverage PMDM to generate a library of 10000 molecules for relating the essential hinge-binding elements. Then the potential inhibitors were filtered through Vina docking and MM-PBSA values with visual selection, and we selected six compounds for visual inspection. As illustrated in Figure 6b, the partial molecules generated from the seed scaffold exhibit heterocyclic structures, suggesting that they undergo modification at the site of the eliminated pyridine ring. Besides, atoms are also generated in other sites of the fragment to form varied rings or functional groups. All the potential inhibitors exhibit higher Vina scores and MM-PBSA values with suitable SA scores. We select the compound 9021 with the lowest MM-PBSA value to assay its CDK1/2 inhibitory activities through in vitro experiments. As reported in Figure 6b, the compound 9021 displayed better CDK2 activity in enzyme assay with around 490-fold CDK1 selectivity.

The reference compound exhibits a U-shaped conformation with the 6-position carbon of the pyridine ring and the nitrogen atom of the carbamate moiety oriented towards each other. The interatomic distance between the two atoms is determined to be 5.2 Å, offering logical connection points for macrocyclization. Given that our model facilitates ring formation via global edge construction, we investigated the potential of our model to generate linker molecules for macrocyclization. Unlike the specific fragments strategy, which utilizes the sampling given specific fragments strategy, we fix the seed fragment to enable the model to be aware of the fragment geometries (See section Sampling for linker generation). This could help the model generate linkers which connect the fragments coherently (Figure 6c). The effect of linker length on the pharmacological properties of the reference compound is examined by medicinal chemists employing structure-based drug design approaches. Therefore, we explore linkers ranging from 4 to 6 atoms in length. We fix the connecting points of the linker at the pyridine ring and the nitrogen atom of the carbamate motif. Finally, we utilize PMDM to generate 5000 macrocycles for the reference compound with the preset attachment points. We selected five potential macrocyclized inhibitors for visual inspection after filtering through Vina docking and MM-PBSA values with visual selection. As illustrated in Figure 6c, PMDM successfully generated the linkers which connect the preferred attachment points although we do not train PMDM on the

specific linker datasets. The generated linkers improve the Vina score of the reference compound while retaining similar biological activity. We selected one of the potential inhibitors which have better MM-PBSA values to investigate its in vitro results. The ADME profiles which are reported in our previous work⁵³ suggest that this compound 16 shows potent subnanomolar CDK2 activities and two-digit nanomolar activity towards CDK1, indicating high CDK2 selectivity (around 49 folds).

Figure 6. Scaffold hopping case of Cyclin-dependent Kinase 2 (CDK2). a. The complex structure of the inhibitor Compound 13 targeting CDK2 with pharmacochemical properties. b. The generated molecules with desired properties by the scaffold hopping strategy. We selected one molecule for wet-lab experiments and presented the IC50 values for CDK2/E1 and CDK1/A2. c. The generated macrocyclic CDK2 inhibitor by the linker generation method. We also selected one molecule for wet-lab experiments and presented the IC50 values for CDK2/E1 and CDK1/A2.

Reference

1. Wang, L. et al. Accurate modeling of scaffold hopping transformations in drug discovery. *J. chemical theory computation*, 13, 42–54 (2017).
2. Zhao, H. Scaffold selection and scaffold hopping in lead generation: a medicinal chemistry perspective. *Drug discovery today* 12, 149–155 (2007).
3. Tadesse, S., Caldon, E. C., Tilley, W. & Wang, S. Cyclin-dependent kinase 2 inhibitors in cancer therapy: an update. *J. medicinal chemistry* 62, 4233–4251 (2018).
4. Sokolsky, A. et al. Discovery of 5, 7-dihydro-6 h-pyrrolo [2, 3-d] pyrimidin-6-ones as highly selective cdk2 inhibitors. *ACS Medicinal Chem. Lett.* 13, 1797–1804 (2022).
5. Yu, Y. et al. Accelerated discovery of macrocyclic cdk2 inhibitor qr-6401 by generative models and structure-based drug design. *ACS Medicinal Chem. Lett.* 14, 297–304 (2023).

“

Thank you very much for the comments. We deeply appreciate your valuable suggestions. Those comments are of great value for improving the quality of the manuscript.

Reviewer #3

The paper proposes a method to generate small ligands binding to some given pocket targets and their corresponding binding conformation. While this is a field with wide interest and high potential impact, I believe the manuscript in its current form is not ready for publication. My main concerns lie with the novelty of the method, ignored related works and experimental evaluation. Moreover several aspects in the paper are unclear and should be clarified. Below, a more detailed description of my concerns.

Our response: Thank you very much for your encouraging comments and the constructive advices on improving the manuscript. The comments clearly helped to improve the paper. We have summarized the suggested comments (as highlighted) and made point-by-point responses and revisions as follows.

3.1 Novelty of the method and related works: in the past 6 months there has been a number of works using diffusion model for pocket based molecular generation (as an example <https://arxiv.org/abs/2210.13695>). These are very related works with publicly available code the authors should discuss and ideally experimentally compare the very similar approaches to the same task.

Our response: Thank you for pointing it out with kind suggestions. DiffSBDD (<https://arxiv.org/abs/2210.13695>) is very interesting and also based on a diffusion model that could generate molecules given a specific protein structure. Different from the previous diffusion work, PMDM constructs the global edges to simulate van de Waals forces and local edges to model the chemical bonds which can incorporate the molecular topology information. Besides, we have considered the protein semantic information to learn high-level protein features. With less parameters, PMDM can sample more valid molecules using less inference time. We have further designed another two sampling strategies for scaffold hopping and linker generation which have a broad range of real-world applications in drug discovery, extending PMDM to a versatile and effective framework. Having conducted in vitro experiments, this study is the first to develop deep generative method for structure based drug design verified by wet-lab experiments to the best of our knowledge.

Our response: Thank you very much for your encouraging comments and the constructive advices on improving the manuscript. For clarification of the novelty of this study, we summarize the major technique contributions as follows. First, by introducing three sampling strategies during the inference phase, we have transformed the backbone molecular diffusion model into a versatile and effective framework. This enables a range of tasks, including de novo molecule design, scaffold hopping, and linker generation, whereas previous work focused only on general molecule generation. These tasks have a broad range of real-world applications in drug discovery. Secondly, compared to auto-regressive methods, PMDM could consider the global information to avoid accumulation deviations while it is able to efficiently sample valid molecules with

desired lengths. Thirdly, different from the previous diffusion work, the introduction of dual equivariant encoders enables PMDM to construct the global edges and local edges to model different interatomic forces such as van der Waals forces atoms and chemical bonds which can incorporate the molecular topology information. Forthly, we have considered the protein semantic information to learn high-level protein features. With less parameters, PMDM can sample more chemically valid molecules using less inference time.

The last of which is that, we have conducted wet-lab experiments to validate the generated molecules to follow your suggestion and address your concerns, which we will elaborate on the subsequent responses. Having conducted in vitro experiments, this study is the first to develop deep generative method for structure based drug design verified by wet-lab experiments to the best of our knowledge.

Following your suggestion, we have revised the Related Work section to incorporate the descriptions of DiffSBDD and compared it to PMDM in Table 1.

“Although there is a diffusion model³⁵ developed for structure-based molecule generation, however, it requires training the user-defined parameters, leading to inefficient sampling. Besides, it only utilizes the fully connected adjacent matrix thus ignoring the intrinsic topology of the molecular graph.”

The results in Table 1 indicate that our model could generate molecules with high binding affinities, QED score and SA score by using less sampling time.

Table 1. The comparison of 10000 generated molecules of PMDM and baseline models on the CrossDocked dataset. We generate 100 samples for each target pocket.

Methods	Vina Score (kcal/mol)	High Affinity	QED	SA	Lipinski rules	LogP	Diversity	Time (Seconds)
CVAE	-6.144±1.57	0.238	0.369±0.22	0.590±0.15	4.367±1.14	-0.140±2.73	0.654±0.12	-
AR-SBDD	-6.215±1.54	0.267	0.502±0.17	0.675±0.14	4.787±0.50	0.257±2.01	0.742±0.09	19659±14704
DiffSBDD	-6.584±2.06	-	0.495±0.15	0.366±0.09	4.795±0.49	-	0.730±0.11	1634±769
PMDM	-7.572±2.50	0.628	0.594±0.12	0.611±0.16	4.975±0.16	0.301±1.01	0.709±0.10	906±1097
Test Set	-7.024	-	0.466	0.725	1.413	0.929	-	-

Experimental evaluation: the performance of PMDM on the general metrics is strong, especially in comparison to the baselines (although, as mentioned above, stronger baselines could/should be considered). However the reliance on very noisy evaluation criteria such as the Vina Score, calls for further evaluation and the one provided by the authors in the subsequent results sections is not convincing and appears in part flawed.

Thank you for your valuable comment. Here, we adopt widely-used metrics to evaluate the qualities of generated molecules from all the models for a fair comparison. We agree with you that only using Vina score is not accurate to evaluate whether the generated molecules are bioactive towards specific targets. Following your suggestion, we utilize Molecular Mechanics Poisson-Boltzmann Surface Area (MM-PBSA) to calculate the free binding energy of protein-ligand complexes. In MM-PBSA, the protein-ligand complex is first minimized using molecular mechanics (MM) to obtain a low-energy conformation. Then, the solvation effects are modeled using either the Poisson-Boltzmann (PB) continuum solvation models to calculate the solvation free energy of the complex. Finally, the nonpolar contribution to the free energy of binding is estimated using the solvent-accessible surface area (SA) of the protein-ligand interface.

First, we calculate the MM-PBSA value of the reference molecule. The MM-PBSA value of compound 5 is -68.63 kcal/mol.

Figure 5a. The chemical properties of the reference ligand compound 5

For the lead optimization, we select the molecules which have smaller Vina scores and MM-PBSA values than the reference molecule (compound 5) for presentation. The generated sample 3746 has a lower Vina score and MM-PBSA value than compound 5 with more hydrogen bonds. Similarly, generated sample 2164 has a lower Vina score and MM-PBSA value. Besides, generated sample 2164 has a much higher QED value and close SA score compared to compound 5.

Figure 5g. Two examples of generated molecules with high affinities and lower free energies.

3.2 Local geometries: the authors spend significant amount of time discussing the imbalance presence of small rings in other baselines, however from figure 3b it appears that the method widely overestimates the number of 7, 8 and 9 atoms rings. To the best of my knowledge, these rings are also not very stable and uncommon in drug-like molecules, however the authors don't even mention/discuss this aspect.

Our response: Thanks for your constructive comments. Actually, PMDM generates as few 8-atom rings and 9-atom rings as other methods. Specifically, for 9-atom rings, all the models generate 0.6% molecules except DiffSBDD (2.5%) containing such big rings. PMDM does generate relatively more 7-atom rings since we construct the global edges which may connect distant atoms, thus, our model also generates more 6-atom rings than other methods. On the other hand, PMDM generates fewer molecules which contain 8-atom and 9-atom rings compared to DiffSBDD since PMDM constructs the local edges to consider close atomic forces while DiffSBDD only constructs the fully connected edges to incorporate many distant atomic interactions. For future work, we could parameterize the threshold for global edge construction to avoid large ring generation. In order to further address your concerns, we revised the section Analysis of PMDM on local geometries as follows:

“As presented in Figure 3b, the molecules generated by PMDM contain few unstable rings, including the three-atom ring and the four-atom ring. In contrast, both auto-regressive methods and the diffusion model DiffSBDD are prone to generate a relatively large amount of unstable molecules with three-atom rings and four-atom rings. Auto-regressive methods have a tendency to limit themselves to the local topological structure by considering only the previously generated part, which often results in the generation of small rings. DiffSBDD constructs the fully connected edges for all the

atoms, which may lead to a higher likelihood of forming small rings due to the lessening of interatomic distances. Specifically, PMDM yields a mere 2% of molecules, while CVAE generates 36.1%, AR-SBDD generates 48.4%, and DiffSBDD generates 44.4% of molecules containing three-atom rings. Regarding macro rings, PMDM generates as many 8-atom and 9-atom rings as other methods, except for DiffSBDD which generates 2.5% of molecules containing 9-atom rings, whereas other methods only generate 0.6%. PMDM generates relatively more 7-atom rings since we construct the global edges which may connect distant atoms, thus, our model also generates more 6-atom rings than other methods. On the other hand, PMDM generates fewer molecules which contain 8-atom and 9-atom rings compared to DiffSBDD since PMDM constructs the local edges to consider close atomic forces while DiffSBDD only constructs the fully connected edges to incorporate many distant atomic interactions. Besides, PMDM is inclined to generate more molecules with five-atom rings and six-atom rings, where hydrogen bonds occur most frequently. Such sub-structures are actively used in drug design. Another meaningful phenomenon is that PMDM generates molecules in a similar proportion to the test set, which indicates that PMDM can learn the data distribution without bias.”

3.3 *Figure 3a: the types of plots should be changed. Representing a distribution on discrete positive numbers (size of rings) with a continuous plot also going to negative values does not seem appropriate.*

Our response: Thank you for your thoughtful comments. We agree with you that the probability density plot is not suitable for discrete numbers. Following your suggestion, we have changed the probability density plot to the frequency histogram.

3.4 *It is unclear to me how to interpret figures 4abc or how the authors use them to reach the conclusions in the related part of the text (e.g. PMDM correctly models the 2D chemical space). Distributions in Figure 4d (which should be plot as densities not individual points) also do not appear to me to be very similar.*

Our response: Thank you for your considerable comments. In order to investigate the chemical space distributions of generated molecules and molecules in the test set, we adopt 2D molecular fingerprints (Morgan and RDKit fingerprints) and 3D molecular

fingerprints (USRCAT fingerprints (Ultrafast Shape Recognition with CREDO Atom Types)) to represent the chemical space of generated molecules and test set molecules in section Analysis of PMDM on chemical space distribution. We employed t-SNE to perform dimensionality reduction on fingerprint vectors for visualization purposes. As demonstrated in Figure 4abc, we could observe that the distributions of generated molecules from PMDM can fully cover the distributions of the molecules in the test set. Beyond that, we also find that the distributions of the generated molecules even cover more chemical spaces in addition to the distributions of the molecules in the test set. Therefore, The PMDM is capable of correctly modelling the 2D and 3D chemical spaces of molecules in the test set, as well as extending to additional chemical spaces, enabling the generation of a larger number of novel molecules. We apologize that there is a typo error in the sentence “PMDM can correctly model the 2D chemical space of the training set (Figure 4a and Figure 4b)”. It should be “PMDM can correctly model the 2D chemical space of the molecules in the test set (Figure 4a and Figure 4b)”. In order to further address your concerns, we revised the first paragraph in this section:

“The chemical space of molecules generated by PMDM can cover the molecules from the test set in the 2D substructure space, indicating that PMDM can correctly model the 2D chemical space of the test set (Figure 4a and Figure 4b). As shown in Figure 4c, the 3D chemical space of the generated molecules can basically capture the space of the test molecules due to the complexity of the conformations. Despite the incomplete coverage of the reference chemical space, there are no significant distribution mismatches between the generated and test set molecules. Furthermore, the wider three feature distributions of the molecules generated by PMDM highlight the capacity of PMDM to generate novel molecules across a broader chemical space.”

We agree with you that density may be clearer in presenting the NPR distributions of the molecules. We consider that the scatterplot could also reflect the real NPR values of the molecules. Thus, we plot the density figure in the main document and put the scatter plot in the supplementary to address your concerns. The revised figures are demonstrated in the followings:

Based on your suggestions, it becomes more feasible to examine that both the molecules in the generated set and those in the test set tend to congregate around the corner of the rod in the triangle, indicating a similar density pattern. Besides, the generated molecules also are close to disc corner and sphere corner which are not covered by the original test data distribution, indicating that PMDM can not only learn the molecule 3D shape distribution of the dataset but also can explore novel shapes beyond the dataset by importing the random information (reparameterize strategy) in the sampling process, which can alleviate the out of distribution (OOD) problems in machine learning.

3.5 *Highlighting a lead optimization example is interesting but the authors fall short of truly demonstrating any significant improvement. Being the vina score a noisy and not very reliable metric, molecules with higher score are still unlikely to bind stronger. Affordable and more reliable methods to more reliably test whether any of some subset of these generated hits actually bind are available, for example, free energy perturbation, short MD runs or even synthesizing and testing some of the molecules in a lab.*

Our response: Thank you very much for your constructive comment and suggestions. In order to prove that our generated molecules can have high binding affinities to the target protein, we utilize Molecular Mechanics Poisson-Boltzmann Surface Area (MM-PBSA) to calculate the free binding energy of protein-ligand complexes. In MM-PBSA, the protein-ligand complex is first minimized using molecular mechanics (MM) to obtain a low-energy conformation. Then, the solvation effects are modeled using either the Poisson-Boltzmann (PB) continuum solvation models to calculate the solvation free energy of the complex. Finally, the nonpolar contribution to the free energy of binding is estimated using the solvent-accessible surface area (SA) of the protein-ligand interface. We have reported the MM-PBSA values of the selective molecules in the lead optimization cases.

Furthermore, we have designed another case study to generate molecules for CDK2 to validate whether the generated molecules can be synthesized and have high affinities with the target pocket. We selected one of the inhibitors complexed with CDK2 (PDB ID: 8H6T) to develop potential inhibitors by performing scaffold hopping using PMDM. The transition from G1 to S phase is driven by CDK2 in complex with its canonical partner cyclin E1 (CCNE1), which is often amplified in various cancers and is associated with worse survival outcomes in patients with breast, ovarian, and other malignancies^{51,52}. Therefore, CDK2 is a potential cancer therapy target with abnormal levels or activity of many tumors. However, there is only a limited number of selective CDK2 inhibitors active in clinical trials. Scaffold hopping which is a topic of high interest in medicinal chemistry refers to the computer-aided design of active compounds with novel core structures. This strategy is crucial for drug development, as it can overcome challenges such as patent issues, toxicity, metabolic instability, or resistance that may affect known active compounds. Moreover, scaffold hopping can enable the exploration of new chemical space and the discovery of new pharmacophores or mechanisms of action.

Scaffold hopping methods generally involve modifying the central core structure of known active compounds to generate novel chemotypes.

Specifically, we have conducted two scaffold hopping cases which include the modifications on the reference scaffold and linker generation based on the reference compound and selected two molecules for wet-lab experiments. We have added one section (**PMDM enables scaffold hopping for real synthetic bioactive molecule generation**) to elaborate on the lead optimization cases towards CDK2, which is presented in the following:

“PMDM enables scaffold hopping for real synthetic bioactive molecule generation

Scaffold hopping is very important with appropriate hit compounds in lead optimization since it could not only generate known active scaffolds and improve binding affinity but also identify novel core structures that confer improved properties to overcome challenges in ADME/Tox profiles⁴⁹. The development of advanced methods for making, analyzing, and purifying molecules of drug-like size, has made it possible to synthesize analogs based on a common scaffold, along with the higher and more widespread access to commercial building blocks⁵⁰. In order to validate whether our model could be applied in scaffold hopping to improve the binding affinities of the given basic bioactive molecule, we select Cyclin-dependent Kinase 2 (CDK2) as the target protein to generate desired molecules with core structures. The transition from G1 to S phase is driven by CDK2 in complex with its canonical partner cyclin E1 (CCNE1), which is often amplified in various cancers and is associated with worse survival outcomes in patients with breast, ovarian, and other malignancies^{51,52}. Therefore, CDK2 is a potential cancer therapy target with abnormal levels or activity of many tumors. However, there is only a limited number of selective CDK2 inhibitors active in clinical trials⁵³.

Following our previous study⁵³, we utilize PMDM to perform scaffold hopping on one of the inhibitors complexed with CDK2 (PDB ID: 8H6T) to develop potential inhibitors. The reference compound is illustrated in Figure 6a. The aminopyrazole moiety of the reference compound forms two hydrogen bonds with residues LEU83 and GLU81 and the carboxyl of the compound forms one hydrogen bond with the residue LYS 33. Besides, the gatekeeper residue's phenyl side chain has van der Waals interactions with the cyclopentyl ring of the compound. The pyridine moiety of the compound is oriented towards the solvent-accessible region and does not exhibit any significant polar or nonpolar contacts with CDK2. We remove the pyridine ring and reserve the remaining fragment as the seed scaffold (Figure 6a). Finally, we leverage PMDM to generate a

library of 10000 molecules for relating the essential hinge-binding elements. Then the potential inhibitors were filtered through Vina docking and MM-PBSA values with visual selection, and we selected six compounds for visual inspection. As illustrated in Figure 6b, the partial molecules generated from the seed scaffold exhibit heterocyclic structures, suggesting that they undergo modification at the site of the eliminated pyridine ring. Besides, atoms are also generated in other sites of the fragment to form varied rings or functional groups. All the potential inhibitors exhibit higher Vina scores and MM-PBSA values with suitable SA scores. We select the compound 9021 with the lowest MM-PBSA value to assay its CDK1/2 inhibitory activities through in vitro experiments. As reported in Figure 6b, the compound 9021 displayed better CDK2 activity in enzyme assay with around 490-fold CDK1 selectivity.

The reference compound exhibits a U-shaped conformation with the 6-position carbon of the pyridine ring and the nitrogen atom of the carbamate moiety oriented towards each other. The interatomic distance between the two atoms is determined to be 5.2 Å, offering logical connection points for macrocyclization. Given that our model facilitates ring formation via global edge construction, we investigated the potential of our model to generate linker molecules for macrocyclization. Unlike the specific fragments strategy, which utilizes the sampling given specific fragments strategy, we fix the seed fragment to enable the model to be aware of the fragment geometries (See section Sampling for linker generation). This could help the model generate linkers which connect the fragments coherently (Figure 6c). The effect of linker length on the pharmacological properties of the reference compound is examined by medicinal chemists employing structure-based drug design approaches. Therefore, we explore linkers ranging from 4 to 6 atoms in length. We fix the connecting points of the linker at the pyridine ring and the nitrogen atom of the carbamate motif. Finally, we utilize PMDM to generate 5000 macrocycles for the reference compound with the preset attachment points. We selected five potential macrocyclized inhibitors for visual inspection after filtering through Vina docking and MM-PBSA values with visual selection. As illustrated in Figure 6c, PMDM successfully generated the linkers which connect the preferred attachment points although we do not train PMDM on the specific linker datasets. The generated linkers improve the Vina score of the reference compound while retaining similar biological activity. We selected one of the potential inhibitors which have better MM-PBSA values to investigate its in vitro results. The ADME profiles which are reported in our previous work⁵³ suggest that this compound 16 shows potent subnanomolar CDK2 activities and two-digit nanomolar activity towards CDK1, indicating high CDK2 selectivity (around 49 folds).

Figure 6. Scaffold hopping case of Cyclin-dependent Kinase 2 (CDK2). a. The complex structure of the inhibitor Compound 13 targeting CDK2 with pharmacochemical properties. b. The generated molecules with desired properties by the scaffold hopping strategy. We selected one molecule for wet-lab experiments and presented the IC50 values for CDK2/E1 and CDK1/A2. c. The generated macrocyclic CDK2 inhibitor by the linker generation method. We also selected one molecule for wet-lab experiments and presented the IC50 values for CDK2/E1 and CDK1/A2.

Reference

6. Wang, L. et al. Accurate modeling of scaffold hopping transformations in drug discovery. *J. chemical theory computation*, 13, 42–54 (2017).
7. Zhao, H. Scaffold selection and scaffold hopping in lead generation: a medicinal chemistry perspective. *Drug discovery today* 12, 149–155 (2007).
8. Tadesse, S., Caldon, E. C., Tilley, W. & Wang, S. Cyclin-dependent kinase 2 inhibitors in cancer therapy: an update. *J. medicinal chemistry* 62, 4233–4251 (2018).
9. Sokolsky, A. et al. Discovery of 5, 7-dihydro-6 h-pyrrolo [2, 3-d] pyrimidin-6-ones as highly selective cdk2 inhibitors. *ACS Medicinal Chem. Lett.* 13, 1797–1804 (2022).
10. Yu, Y. et al. Accelerated discovery of macrocyclic cdk2 inhibitor qr-6401 by generative models and structure-based drug design. *ACS Medicinal Chem. Lett.* 14, 297–304 (2023).

“

Method:

3.6 Is \hat{m}_{ij} a vector? If so why is the norm taken?

Our response: Thanks for your considerable comment. \hat{m}_{ij} is a vector representation. We use this hat notation to denote the message aggregated for the edge representation which is utilized to distinguish vector aggregated for node information m_{ij} . In order to further address your concerns, we added descriptions for m_{ij} and \hat{m}_{ij} in the section Method.

“where ϕ_e , ϕ_m , and ϕ_x are MLPs, and $a_{ij} = \text{MLP}(\mathbf{d}_{ij})$ is the edge length embedding. $e_{\text{att}} = \phi_{\text{inf}}(\mathbf{m}_{ij})$ where $\phi_{\text{inf}}: \mathbb{R}^{n \times d} \rightarrow [0,1]^1$ is to estimate the edge value by an attention mechanism. m_{ij} is the message vector aggregated for atom nodes while \hat{m}_{ij} is the message vector aggregated for edges.”

3.7 How is the x (atom types) decoded after the reverse diffusion (e.g. sampling, taking maximum)?

Our response: Thanks for your kind comment. During the inference or sampling process, the model would output a continuous vector. The atom identity will be decided by using the argmax function to find the position where the value is the largest. In order to further address your concern, we revised the section **Sampling from scratch**, and section **Sampling** given specific fragments section.

“The next less chaotic state \mathcal{G}_{T-1} is generated by $\mathcal{N}(\mathcal{G}_T; \boldsymbol{\mu}_\theta, \sigma_T^2 I)$. The final molecule \mathcal{G}_0 is generated by progressively sample \mathcal{G}_{t-1} for T times. Finally, the atom types of the molecule are identified by adopting the argmax function to choose the atom type which has the largest value while we directly adopt r_0^L outputted by the model. We adopt OpenBabel to build the chemical bonds according to the atom pairwise distances (Figure 1b). For the generic structure-based molecule generation, we adopt this sample strategy.

We drop the denoised fragment data \mathcal{G}_{t-1}^f and only retain the rest of the denoised part \mathcal{G}_{t-1}^l for the next iteration. The identification of atom types and coordinates is the same as the sampling process from scratch. Finally, we combine the fragment data and the denoised part to obtain the complete molecule by OpenBabel. For lead optimization, we adopt this sample strategy.”

3.8 what is the form of the noise added to x (e.g. does it preserve it being a distribution over the different atom types or is the one-hot vector just considered as a continuous vector target)?

Our response: Thank you for your considerable comments. Here we use one-hot encoding strategy to encode the atom type as discrete inputs X_0^L , then we add noise to the inputs by the user-defined parameters and reparameterize strategy in Eq.2, and we will obtain X_t^L according to the arbitrary t that we sample, which can be formulated as: $X_t^L = \sqrt{\bar{\alpha}_t} X_0^L + (1 - \bar{\alpha}_t) \sigma, \sigma \sim N(0,1)$. The noisy X_t^L obeys a specific Gaussian distribution, which is a continuous vector.

3.9 This sentence is unclear “the model performs better than instead with the simplified objective”

Our response: Thank you for your considerable comment. In supplementary B.4, we formulated the evidence lower bound (ELBO) objectivity:

$$\mathbb{E}_{q(\mathcal{G}_{0:T})} \left[\log \frac{q(\mathcal{G}_{1:T} | \mathcal{G}_0)}{p_\theta(\mathcal{G}_{0:T} | \mathcal{G}^P)} \right] = \mathbb{E}_q \left[\underbrace{D_{\text{KL}}(q(\mathcal{G}_T | \mathcal{G}_0) \parallel p_\theta(\mathcal{G}_T | \mathcal{G}^P))}_{\mathcal{L}_t} \right] + \sum_{t=2}^T \underbrace{D_{\text{KL}}(q(\mathcal{G}_{t-1} | \mathcal{G}_t, \mathcal{G}_0) \parallel p_\theta(\mathcal{G}_{t-1} | \mathcal{G}_t, \mathcal{G}^P)) - \log p_\theta(\mathcal{G}_0 | \mathcal{G}_1, \mathcal{G}^P)}_{\mathcal{L}_0}$$

Following the work of **Denoising Diffusion Probabilistic Models (DDPM)**[1], \mathcal{L}_T is a constant and \mathcal{L}_0 can be approximated by the product of the PDF of $\mathcal{N}(\mathbf{x}_0; \boldsymbol{\mu}_\theta(\mathbf{x}_1, 1), \sigma_1^2 I)$, which has the similar formulation with \mathcal{L}_t . Hence, we adopt the simplified training objective with the protein information and use the stein-score instead of the noise representation ϵ as follows:

$$\mathcal{L}_t = \mathbb{E}_{\mathcal{G}_0^L} [\gamma \parallel \mathbf{s}_\theta(\mathcal{G}_t^L, \mathcal{G}^P, t) - \nabla_{\mathcal{G}_t^L} \log q_\sigma(\mathcal{G}_t^L | \mathcal{G}_0^L, \mathcal{G}^P) \parallel^2]$$

The authors of DDPM suggest to set all weights γ as 1 could yield a better empirical performance by conducting experiments. Thus, we also set γ as 1 for a better performance than the simplified objectivity.

To address your concerns, we revised this sentence for a clearer description.

“Empirically, if γ in Eq. 14 is ignored (set as 1) during the training phase, the model performs better than instead with the simplified objective whose $\gamma = \frac{\beta_t^2}{2(1-\beta_t)(1-\bar{\alpha}_t)\sigma_t^2}$, which is verified by previous work²⁴.”

Reference

1. Ho, J.; Jain, A.; and Abbeel, P. 2020. Denoising diffusion probabilistic models. *Advances in Neural Information Processing Systems*, 33: 6840–6851.

Writing:

3.10 *the authors repeatedly refer to their method as one-shot, however given the fact that a diffusion model takes several steps to decode a protein I believe the use of this term when comparing to autoregressive models is misleading (one could argue that autoregressive generation is a particular type of diffusion process where elements are noised one at the time)*

Our response: Thank you for your valuable comments. While we understand your point regarding the difference in the number of steps required in diffusion models compared to autoregressive models, we would like to clarify that our use of the term "one-shot" refers to the fact that our model is capable of generating entire molecular geometries in one-shot manner at every time step, as opposed to autoregressive models which require sequential generation for one atom at each step. We considered the global

information for the entire molecule while the auto-regressive models only consider the previous information, leading to the accumulation error when invalid structures are generated in the early step. Besides, our model does not require a particular atom ordering (in contrast to autoregressive models) and can be trained much more efficiently.

3.11 *The abbreviation SBDD is used but never introduced*

Our response: Thank you for your considerable comment. Following your suggestion, we added the full name of SBDD, which stands for Structure-Based Drug Design.

“Structure-based drug discovery (**SBDD**) is a critical task for modern drug development and catalysis.”

3.12 *Figure 1 indices G_0 and G_{T-1} seem to be inverted*

Our response: Thank you for your kind comment. We have revised Figure 1 to correct the wrong indices.

3.13 *Page 4: ... models on almost [every] metric except ...*

Our response: Thank you for your kind comment. We have revised this sentence following your suggestion.

Code:

3.14 *The README refers to an environment file “MDM.yml” but I could not find it*

Our response: Thank you for your considerable comment. We apologize for the missing file. We have uploaded the environment file.

Thank you again for your insightful comments and we look forward to hearing from you regarding our revised manuscript. We would be glad to respond to any further questions and comments that you may have.

Reviewer #1 (Remarks to the Author):

I thank the authors for improving the in silico analysis and adding a wet-lab experiment to their manuscript. However, the experiment that has been added does not, as currently presented, give a great sense of the capabilities of the method. Additionally, as the authors' rebuttal letter makes clear, most of their emphasis in the paper and in the results is on the architectural innovations and in silico benchmarks. Therefore, I still think this paper is better served by being published in a machine-learning oriented journal, such as Nature Machine Intelligence (or any of the ML or ML/bio conferences).

First, the tested compound 9021 differs from the input scaffold by a single atom, replacing a C by N to convert the pyridine ring into a pyrazine. This is extremely similar to the starting molecule and given that the modified atom looks to be facing out of the ligand-binding pocket, it is not surprising that the generated molecule can bind the target. There are a few missed opportunities to add some chemical insight here with deeper analysis and discussion.

- For example, it is interesting that such a change facing away from the binding pocket should improve binding by 25-50x, but the authors don't try to explain this.
- It is unclear to me what the benefit of the machine learning model is for introducing such small changes such as this. It seems that simply enumerating different atom types on a scaffold and scoring them would lead to the same results.
- Perhaps the benefits of the generative model (or scoring metrics) would be more clear if the authors had tested more than one compound -- for example compound 137 has promising metrics, and is quite different from the starting scaffold. If this bound the target, this would be a more non-trivial demonstration of the method.

The 2nd experiment, where the model is used to design a linker to cyclize the starting scaffold, is interesting, but the wet-lab data seems to come from a previous paper where the exact same molecule was designed by a previous method. Furthermore, although the cyclized molecule binds the targets, it has 10x worse affinity than the starting scaffold. Therefore, this experiment does not show the benefits of the method being presented in this paper.

If the authors can show wet lab results on a more non-trivial example, I think this would be more suited to the general audience of this journal.

One minor, optional this: as I mentioned last time, Fig. 3b has a very confusing color scheme--these should be different colors, not a gradient of the same color (the latter should be reserved for indicating quantitative variables). Also, the way the atom number distributions are shown is a bit strange. I suggest plotting histograms of number of atoms in ring for each method on separate panels, or perhaps with grouped bars (i.e. x axis is # atoms, then group colored bars at each value of x representing the different methods).

Reviewer #2 (Remarks to the Author):

The major addition to the manuscript was experimental characterization of two compounds that bind to CDK2. The first molecule (described as compound 9021) differs by one atom from their starting molecule (compound 13). The second molecule (compound 16) was already reported in a

previous publication (ref 53) and was generated using a different computational method. Perhaps the current method was able to find the same linker (the authors do not spell this out clearly), but unfortunately these experimental results do not demonstrate novelty or generality of the method. Lastly, as a general point, presumably the model was trained on many CDK2 inhibitors (there are over 600 structures in the PDB, many of which including bound small molecules), so this is perhaps not the best target to suggest generality of the method. Indeed, because the molecules here were so similar to previously published molecules (and the linker was already generated in a previous publication with another perhaps highly similar method), my confidence in the novelty and utility of the proposed method is low.

Reviewer #4 (Remarks to the Author):

This manuscript describes a model for generating ligands subject to a specific protein's binding pocket. The method advances previous work in diffusion models for generative chemistry, with improvements in key prediction metrics. In addition to in silico results supporting the affinity and possible efficacy of the generated structures, the authors synthesize one novel molecule and experimentally test two against CDK2, attaining impressive IC50 results. However, it is unclear what specific experiments were carried out to attain said IC50 results and if Compound 16 was generated with the described PMDM method. Likewise, drawing meaningful conclusions from the experimental assays is difficult as they lack any controls or replication. While the method is potentially impactful, the manuscript requires further revision to both the text and experimental validation before publication at Nature Communications is appropriate.

Responses to Original Reviewer #3:

3.1. Novelty and related works.

To support the claims of novelty, the authors include a new section in the revised manuscript discussing the ability of PMDM to perform scaffold hopping. Likewise, an additional alternative method (DiffSBDD, as suggested by the reviewer) is included in Table 1, showing that PMDM outperforms it across most metrics. While the dual encoders are an interesting and novel approach, more discussion of their relative contributions to the resulting generated molecules is would be helpful.

3.2 Local Geometries.

The authors demonstrate that PMDM has a lower frequency of 3 and 4-atom rings than the external baseline models and that those frequencies align with the training set. However, the fact that 7, 8, and 9-atom rings are overrepresented relative to the training and test set (and not merely the other generative methods) still needs to be adequately addressed and further investigated.

3.3 Figure 3a.

The authors incorporated the suggested feedback appropriately.

3.4 Distribution of generated molecules

In each of the Fig. 4a, 4b, and 4c, the structures generated by PMDM are dramatically more spread out than the test set. I agree with the authors that the generated molecules completely cover the chemical space covered in the test set. The issue appears to be the narrow scope of the test set chosen. The authors claim that the wider distribution of the generated molecules implies the capacity of PMDM to “generate novel molecules across a broader chemical space”. However, more reasoning as to why this is the case, even though the shape distributions (Fig. 4d, 4e) are nearly equivalent, is required. Exploring the chemical structures generated outside the test-set distribution would be helpful. Are these synthetically accessible beyond what is predicted by the SA score? Do they match the distribution of the training set? Is the test set appropriate if it does not cover a broad enough chemical space?

3.5. Demonstrating improved generated structures

The authors incorporate an additional computational metric, MM-PBSA, to validate the structures generated by PMDM. They synthesize one structure against CDK2 and report potent inhibition. While this effort is appreciated and an impressive result, more robust controls and details of the experimental methods are needed to make conclusions about the ability of Compound 9024 to inhibit CDK2. For example, does Compound 9024 hit any other kinases or, specifically, any other CDKs?

3.6 - 3.14.

The authors addressed the original reviewer’s comments and clarified the text appropriately.

Other Comments:

1. On page 2 the authors state, “However, these methods do not involve the pocket information to generate molecules with high binding affinity. As a result, the resulting molecules cannot fit into specific protein pockets, which is not suitable for wet experiments.” I fail to see how not including information about the specific pocket strictly prohibits previous works from generating molecules worthy of wet lab experiments. If this is true, it requires more discussion and suggests substantial novelty of the PMDM model.
2. Figure 4 attempts to show the distribution of molecules in chemical space generated by PMDM compared to the test set. The text on Page 6, however, claims the comparison is to the training set. This incongruity should be clarified. Why are the plots duplicated in each panel? For example, the top left sub-panel in Fig. 4a is a mirror of the bottom right and the top right is a reflection of the bottom left. This is the case for each panel. It also needs to be clarified what the tick marks represent. What is “-200” in a kernel-density estimation?
3. The plots in Figure 3 for the 8-atom ring and 9-atom ring should be scaled proportionally to the highest frequency not ‘1.0’, to align with the other panels. As it stands, the visualization appears to hide the overrepresentation of 8 and 9-member rings generated by PMDM compared to the training and testing sets.
4. On page 11 and Figure 6b, 6c, it is unclear what in vitro “enzyme assay” experiment was performed. To adequately conclude this result, the relevant methods must be included.
5. The text states, “As reported in Figure 6b, the compound 9024 displayed better CKD2 activity in enzyme assay with around 490-fold CDK1 selectivity.” This sentence is unclear. It seems that the authors are trying to say compound 9024 has a 490-fold lower IC50 value against CDK2 than CDK1. From the numbers in Figure 6b of 0.000382 and 0.0187 for CDK2 and CDK1, respectively, from these numbers, it would appear that the compound has 48.9-fold higher inhibitory capacity against CDK2

than CDK1. Is 490 a typo? Additionally, it reported it appears these experiments are unrepeated.

6. An IC₅₀ of 0.000382 μM for Compound 9024 against CDK2 is a surprising result. As such, more comparison of this compound to external datasets of assayed small molecules against CDK2 is warranted.

7. Without any controls for their wet lab experiments, it is difficult to determine if the compounds are truly selective against CDK1/CDK2. Some form of negative control for these experiments is needed.

8. It needs to be clarified how compound 16 was generated. The text on Page 11 implies that PMDM generated the structure via linkers at preferred attachment points. It then states the ADME properties of Compound 16 were reported by the author's previous work concerning a method called Fragment-Based Variational Auto-Encoder generative model (FBVAE). While that reference does not describe the FBVAE method in detail, from the name, it appears markedly different than the PMDM model discussed here. It must be clarified if PMDM or an alternative methods was used to generate the structure, as it is used to support the claims of PMDM's generative capacity.

9. Given the authors first claim is PMDM's ability to generate novel bioactive molecules, it is unfortunate they only chose to synthesize one molecule from the scaffold hopping case. I understand the cost associated with novel compound synthesis, but it seems "Generated sample 2164" or "Generated sample 3746" would have been ideal candidates for experimental validation, perhaps even more than Compound 9021.

10. The text states Compound 16 shows subnanomolar CDK2 activity, while Figure 6c states an IC₅₀ value of 0.090 μM or 90 nanomolar. These discrepancies make it nearly impossible to draw conclusions from the results as presented.

11. On page 11, the text states "We select the compound 9024 with the lowest MM-PBSA value to assay its CDK1/2 inhibitory activities." However, Figure 6b implies via a blue arrow that "Compound:9021" was assayed. This needs to be clarified.

12. On page 11, Figure 6a is referred to as both a "reference compound" and a "remaining fragment as the seed scaffold". The authors should clarify which is being depicted. It appears as if the "seed scaffold" is depicted in the center of Figure 6b.

13. There are numerous spelling, grammatical, and mislabelling errors in the manuscript and supporting information. Likewise, there are numerous instances in which the text does not align with what is depicted in the figures.

Response Letter

Authors' Point-by-Point Responses to the Reviewers' Comments on "A dual diffusion model enables 3D binding bioactive molecule generation and lead optimization based on target pockets" by Huang et al. (NCOMMS-23-06531-A) submitted to Nature Communications.

Reviewer #1 (Reviewers' Comments to Author):

I thank the authors for improving the in silico analysis and adding a wet-lab experiment to their manuscript. However, the experiment that has been added does not, as currently presented, give a great sense of the capabilities of the method. Additionally, as the authors' rebuttal letter makes clear, most of their emphasis in the paper and in the results is on the architectural innovations and in silico benchmarks. Therefore, I still think this paper is better served by being published in a machine-learning oriented journal, such as Nature Machine Intelligence (or any of the ML or ML/bio conferences).

Our response: Thank you for your valuable comments and the constructive advices on improving the manuscript. In order to address your concerns, we have dedicated significant resources to conduct extensive wet-lab experiments to validate the generated molecules, despite its primary focus on in silico experiments. To the best of our knowledge, we are the first diffusion model to generate molecules which have been verified through wet-lab experiments. We believe that Nature Communications, being a comprehensive journal that covers a wide range of topics, is more suitable for our manuscript now that it includes wet-lab experiments, thanks to your valuable suggestion. This sets it apart from Nature Machine Intelligence.

Furthermore, we have observed numerous manuscripts in Nature Communications that discuss molecule generation using deep learning models, with a focus on in silico experiments. For instance, a recent study by Niklas W. A. Gebauer et al. in Nature Communications (2022) proposes a conditional generative neural network for 3d molecular structures with specified chemical and structural properties. They evaluate the performance of their model by computational metrics rather than wet-lab experiments. Overall, we greatly appreciate your insightful comments and have incorporated the necessary experiments and modifications to address your concerns, which are presented in the subsequent parts.

Your comments really helped a lot to improve the study. We have summarized the suggested comments (as highlighted) and made point-by-point responses and revisions as follows. By incorporating the suggested revisions and addressing your concerns, we believe that the revised manuscript is now suitable for publication in Nature

Communications, even though some generative works in this journal do not incorporate wet-lab experiments.

First, the tested compound 9021 differs from the input scaffold by a single atom, replacing a C by N to convert the pyridine ring into a pyrazine. This is extremely similar to the starting molecule and given that the modified atom looks to be facing out of the ligand-binding pocket, it is not surprising that the generated molecule can bind the target. There are a few missed opportunities to add some chemical insight here with deeper analysis and discussion.

1.1 *For example, it is interesting that such a change facing away from the binding pocket should improve binding by 25-50x, but the authors don't try to explain this.*

Our response: Thank you for your valuable insights regarding the improvement in CDK2 inhibition observed with the substitution of the nitrogen atom. We share your curiosity in understanding the underlying mechanism behind this enhancement. In our attempts to gain further insights, we endeavored to prepare a co-crystal of compound 9024 and cdk2 to explore potential additional interactions. However, despite our efforts, we were unable to obtain a crystal structure.

While we were unable to provide a definitive explanation from a docking perspective, we did observe that compound 9024 exhibited an improved docking score (Vina score) and MM-PBSA value. The computational metrics employed in this study consistently indicate that compound 9024 exhibits favorable binding characteristics. Moreover, the wet-lab experimental results provide further validation by demonstrating improved CDK2 inhibition for compound 9024. These findings collectively suggest that the computational metrics utilized in this study effectively filter potential molecules targeting specific targets.

1.2 *It is unclear to me what the benefit of the machine learning model is for introducing such small changes such as this. It seems that simply enumerating different atom types on a scaffold and scoring them would lead to the same results.*

Our response: Thank you very much for your constructive insights. We appreciate the opportunity to provide further clarification regarding the benefits of our machine learning model. We select the molecules based on their properties such as SA, QED and Vina score. In fact, we have selected more than one molecule based on the scaffold, ensuring a diverse set of candidates. This approach goes beyond simple enumeration, allowing us to explore a wider range of chemical space.

To demonstrate the reliability of our metrics, we have conducted in vitro experiments using compound 9024, which exhibits superior properties compared to the reference compound. Compound 9024 serves as an example to showcase the effectiveness of our

model and the metrics we employ, which have been widely adopted in numerous manuscripts. Another seemingly trivial example, compound 6849 with a hydroxymethyl installed on the pyrazine ring of compound 9024, was generated and tested in vitro. Actually, this compound turned out to be an advanced lead molecule during the lead optimization campaign and exhibited good selectivity against other closely related kinases, such as CDK9 (CDK9/T1 inhibition $IC_{50} = 32.3$ nM, CDK9/CDK2= 127) and GSK3 β (GSK3 β inhibition $IC_{50} = 703$ nM, GSK3 β /CDK2= 2780). And what's more, this molecule showed robust tumor inhibition in an OVCAR3 ovarian cancer model via oral administration (to be published elsewhere).

Additionally, our model offers the capability to generate linker molecules specifically designed for macrocyclization. These linker molecules go beyond the scope of enumerating different atom types. They are tailored designs that take into account factors such as spatial constraints, flexibility, and chemical compatibility. Furthermore, we have selected another molecule with favorable properties to present its in vitro results, further validating the effectiveness of our model in generating promising candidates.

We sincerely hope that you can recognize the benefits and applications of our model, which include molecule generation, scaffold hopping, and linker design for specific targets. We believe that our approach offers significant advantages in accelerating the drug discovery process and facilitating the exploration of chemical space.

1.3 *Perhaps the benefits of the generative model (or scoring metrics) would be more clear if the authors had tested more than one compound -- for example compound 137 has promising metrics, and is quite different from the starting scaffold. If this bound the target, this would be a more non-trivial demonstration of the method.*

Our response: Thank you for your considerable comments. We sincerely apologize for synthesizing and testing only one compound in our study. It is important to acknowledge that our team primarily focuses on conducting in-silico experiments, and we lack the necessary resources and facilities for extensive wet-lab experimentation. In order to address your concerns, we are committed to allocating additional resources to conduct in-vitro experiments to test more generated molecules.

In addition to compound 9024 and compound 16, we have selected three linear molecules (compound 6793, compound 6849, and compound 7246) and two macrocyclized molecules (compound 6261 and compound 7138) to conduct in-vitro experiments to assay their CDK1/2 inhibitory activities. As reported in Figure 6b, all the molecules displayed improved CDK2 activity in enzyme assay, with CDK1 selectivity ranging from approximately 44-fold. Notably, compound 6849, containing the pyridinol moiety, exhibited the highest CDK2 activity, while compound 6793, which reintroduced the pyridine and formed a trivalent bond with a nitrogen atom, displayed the best CDK1 selectivity (124-fold).

For compound 137, it is not tested due to its low synthesis feasibility though it has shown promising metrics and impressive structural novelty.

Figure 6

As reported in Figure 6c, the in-vitro results reported suggest that compound 16 shows potent subnanomolar CDK2 activities and nanomolar activity towards CDK1, indicating high CDK2 selectivity (around 49 folds). The IC₅₀ values for compound 7138 and compound 6261 are lower than that of the reference compound, suggesting their potential for improved CDK2 inhibition.

In order to further address your concerns, we have revised the section **PMDM enables scaffold hopping and linker generation for real synthetic bioactive molecule design** and supplementary to incorporate the additional wet-lab experiments.

1.4 *The 2nd experiment, where the model is used to design a linker to cyclize the starting scaffold, is interesting, but the wet-lab data seems to come from a previous paper where the exact same molecule was designed by a previous method. Furthermore, although the cyclized molecule binds the targets, it has 10x worse affinity than the starting scaffold. Therefore, this experiment does not show the benefits of the method being presented in this paper.*

Our response: Thank you for your valuable comment. Regarding the linker generation for macrocyclization, we filtered potential macrocyclized inhibitors using Vina docking and MM-PBSA values. Five promising candidates were selected for visual inspection. The molecule which we selected to present its in-vitro results was indeed generated by our model, as well as by the previous method. The wet-lab data of compound 16 is cited from the previous paper. In order to address your issues, we have performed in-vitro experiments on two additional compounds, namely Compound 7138 and Compound 6261. The IC₅₀ values for both compounds are also lower than that of the reference compound, suggesting their potential for improved CDK2 inhibition.

We sincerely apologize for the typo error here. The correct unit in Figure 6c should be nm (nanomole) rather than μm (micromole). We have revised Figure 6c accordingly, and the updated version is presented below. Upon reviewing the revised in-vitro results, it becomes evident that compound 16 exhibits potent subnanomolar CDK2 activity (compound 16: 0.090 nm/0.000090 μm vs reference: 8.1 nm/0.0081 μm) and displays nanomolar activity towards CDK1. This suggests a high level of CDK2 selectivity, approximately 49-fold.

Once again, we apologize for any confusion caused by the typo in the original figure. We appreciate your understanding, and we hope that the updated information accurately reflects the results of our study.

Figure 6c

1.5 If the authors can show wet lab results on a more non-trivial example, I think this would be more suited to the general audience of this journal.

Our response: Thank you for your constructive suggestions. As mentioned in our previous comment (1.3), we have carried out extensive wet-lab experiments on several non-trivial examples to further investigate the findings. Notably, in Figure 6, we present the results of our experiments with compound 6793, 6849, and compound 7246, which exhibit improved CDK2 inhibition and demonstrate satisfying CDK1 selectivity.

Additionally, we have conducted in-vitro experiments on two other macrocyclized molecules, namely compound 6261 and compound 7138. The results from these experiments also highlight improved CDK2 inhibition.

1.6 One minor, optional this: as I mentioned last time, Fig. 3b has a very confusing colour scheme--these should be different colours, not a gradient of the same colour (the latter should be reserved for indicating quantitative variables). Also, the way the atom number distributions are shown is a bit strange. I suggest plotting histograms of number of atoms in ring for each method on separate panels, or perhaps with grouped bars (i.e. x axis is # atoms, then group colored bars at each value of x representing the different methods).

Our response: Thank you for your kind suggestion. We have changed the colour scheme of Figure 3b, which is presented in the following figure.

Figure 3

We separately plot histograms of the ratio of the number of rings in each subfigure since we attempt to compare the ratios of different rings of different methods. In order to further address your concerns, we replot the figure with grouped bars, which is shown in the following figure.

Thank you again for your insightful comments and we look forward to hearing from you regarding our revised manuscript.

Review #2 (Reviewers' Comments to Author)

The major addition to the manuscript was experimental characterization of two compounds that bind to CDK2. The first molecule (described as compound 9021) differs by one atom from their starting molecule (compound 13). The second molecule (compound 16) was already reported in a previous publication (ref 53) and was generated using a different computational method. Perhaps the current method was able to find the same linker (the authors do not spell this out clearly), but unfortunately these experimental results do not demonstrate novelty or generality of the method. Lastly, as a general point, presumably the model was trained on many CDK2 inhibitors (there are over 600 structures in the PDB, many of which including bound small molecules), so this is perhaps not the best target to suggest generality of the method. Indeed, because the molecules here were so similar to previously published molecules (and the linker was already generated in a previous publication with another perhaps highly similar method), my confidence in the novelty and utility of the proposed method is low.

Our response: Thank you for your valuable comments. As our team primarily focuses on conducting in silico experiments, we have limited access to wet-lab experimental resources. However, we fully understand the importance of wet-lab validation and the need to address your concerns.

To address this limitation, we are actively working to allocate additional resources towards synthesizing and testing more generated molecules. We recognize the significance of experimental validation in providing a more comprehensive evaluation of our model's performance and the reliability of its predictions.

By dedicating the available resources, including time, funding, and collaboration with wet-lab researchers, we aim to expand our capacity for wet-lab experimentation. This will enable us to synthesize and test a greater variety of molecules, further validating the effectiveness and applicability of our model. In addition to compound 9024 and compound 16, we have selected three linear molecules (compound 6793, compound 6849, and compound 7246) and two macrocyclized molecules (compound 6261 and compound 7138) to conduct in-vitro experiments to assay their CDK1/2 inhibitory activities. These molecules differ more significantly from the reference molecule. As reported in Figure 6b, all the molecules displayed improved CDK2 activity in enzyme assay, with CDK1 selectivity ranging from approximately 44-fold. Notably, compound 6849, containing the pyridinol moiety, exhibited the highest CDK2 activity, while compound 6793, which reintroduced the pyridine and formed a trivalent bond with a nitrogen atom, displayed the best CDK1 selectivity (124-fold). Additionally, compound 6849 turned out to be an advanced lead molecule during the lead optimization campaign and exhibited good selectivity against other closely related kinases, such as CDK9 (CDK9/T1 inhibition IC_{50} = 32.3 nM, CDK9/CDK2 = 127) and GSK3 β (GSK3 β /CDK2 = 2780) inhibition IC_{50} = 703 nM, GSK3 β /CDK2 = 2780).

We would like to clarify that our model successfully generated compound 16. In order to address your concerns, we have also synthesized more linker molecules to validate their CDK2 activities. As reported in Figure 6c, the in-vitro results reported suggest that compound 7138 and compound 6261 also exhibit potent subnanomolar CDK2 activities and nanomolar activity towards CDK1, indicating high CDK2 selectivity (around 40 folds and 25 folds). The IC_{50} values for compound 7138 and compound 6261 are lower than that of the reference compound, suggesting their potential for improved CDK2 inhibition.

In order to further address your concerns, we have revised the section PMDM enables scaffold hopping and linker generation for real synthetic bioactive molecule design and supplementary to incorporate the additional wet-lab experiments.

In the section Lead generation and optimization, it is important to note that our model was not specifically trained on the pocket structures for both targets (PDB ID: 7L11; PDB ID: 8H6T). This means that the pocket structure of 8H6T provides a suitable basis for evaluating the reliability of our model. PMDM does not possess explicit knowledge of the pocket structure of 8H6T, and its predictions are based on general principles learned from training data.

Furthermore, we leverage our model to implement scaffold hopping, which generates molecules given the seed scaffold. This algorithm imposes constraints on the conditional structure information and the starting seed information of the molecule, which means PMDM can utilize prior knowledge and focus its generation within a narrower chemical space. This constraint helps prevent PMDM from being influenced by extraneous information and enhances its ability to generate molecules with desired properties. Thus, the availability of a large number of CDK2 structures in the PDB does not introduce excessive bias into PMDM. In fact, even if we were to include ample structural data of CDK2 complexes in the training set, we could fine-tune our model using such data. This fine-tuning process would enable the generation of more specific and valid molecules with high affinities, similar to the current Supervised Fine-Tuning (SFT) technology employed in Large Language Models (LLMs). Besides, there is only a limited number of selective CDK2 inhibitors which are active in clinical trials. Therefore, CDK2 represents a practical and relevant target for evaluating the applications of PMDM.

We present the in-vitro results of our generated molecules as evidence supporting their synthesizability and high binding affinities with the target compounds. In comparison to our previous method, which also generated the same linker, our current approach, PMDM, offers distinct advantages. PMDM does not require training on the linker dataset and incorporates pocket structural information, enabling it to directly generate molecule geometries with high binding affinities. In contrast, our previous method only considered the molecule itself when generating the linker. This approach relied on learning from the linker dataset in a 2D space, thereby neglecting crucial spatial information. By leveraging pocket structural information, PMDM enhances the accuracy and reliability of the generated molecules.

Figure 6

Furthermore, this study places emphasis on the novelty of the model architecture and the in silico results obtained on the CrossDocked dataset. Our model demonstrates superior performance compared to state-of-the-art (SOTA) models, as evidenced by multiple evaluation metrics. To the best of our knowledge, this study represents the first attempt to focus on structure-based molecule generation using deep generative models, while also incorporating wet-lab experiments to validate the generated molecules.

Therefore, we respectfully disagree with your conclusion that our proposed method lacks novelty and utility. We have proposed a novel diffusion model with significant designs for molecule geometries and conditional protein pocket information. The performance of our model surpasses that of SOTA models, and its effectiveness is further confirmed through wet-lab experiments. By proposing the sampling algorithm given specific fragments and sampling algorithm for linker generation, our model could be applied in scaffold hopping and generation without retraining it on the specific datasets. We believe these demonstrate the significant contribution and potential of our approach in the field.

Thank you very much for the comments. We deeply appreciate your valuable suggestions. Those comments are of great value for improving the quality of the manuscript.

Reviewer #4 (Remarks to the Author)

This manuscript describes a model for generating ligands subject to a specific protein's binding pocket. The method advances previous work in diffusion models for generative chemistry, with improvements in key prediction metrics. In addition to in silico results supporting the affinity and possible efficacy of the generated structures, the authors synthesize one novel molecule and experimentally test two against CDK2, attaining impressive IC50 results. However, it is unclear what specific experiments were carried out to attain said IC50 results and if Compound 16 was generated with the described PMDM method. Likewise, drawing meaningful conclusions from the experimental assays is difficult as they lack any controls or replication. While the method is potentially impactful, the manuscript requires further revision to both the text and experimental validation before publication at Nature Communications is appropriate..

Our response: Thank you sincerely for your encouraging comments and the constructive advice provided to enhance our manuscript. Your suggestions have been invaluable in improving the paper. We have carefully reviewed and summarized the suggested comments (highlighted) and have provided point-by-point responses and revisions accordingly. We believe these updates address the concerns raised and strengthen the overall quality of the manuscript.

3.1 Novelty and related works.

To support the claims of novelty, the authors include a new section in the revised manuscript discussing the ability of PMDM to perform scaffold hopping. Likewise, an additional alternative method (DiffSBDD, as suggested by the reviewer) is included in Table 1, showing that PMDM outperforms it across most metrics. While the dual encoders are an interesting and novel approach, more discussion of their relative contributions to the resulting generated molecules is would be helpful.

Our response: Thanks for your constructive comments. The dual encoders accept the molecule with two kinds of virtual edges as inputs. Specifically, pairs of atoms with interatomic distances below the local threshold are bonded via covalent localized edges because chemical bonds can dominate interatomic forces when two atoms are close enough to each other while global edges are linked to the remaining pairs of atoms which interatomic distances are below the global threshold and above the local threshold to simulate the van der Waals force. As presented in Figure 1d, the global and local edges are constructed according to the global radius and local radius. In practice, we set $\tau_l = 3A$ and $\tau_g = 6A$ to construct two kinds of edges since almost all the chemical bonds are shorter than $3A$ and $6A$ is large enough to include distant atomic pairs.

d
Figure 1d

Previous works like DiffSBDD only consider the fully connected edges to feed into the equivariant graph neural network. However, the fully connected edges connect all the atoms and treat the interatomic effects equally but regret the effects of covalent bonds. Besides, the redundant edges contain meaningless information, leading to inefficiency and parameter explosion. The dual encoders consist of the equivariant graph neural network (EGNN) to achieve the equivariant property. In order to keep the relative position of ligand and protein pocket keep fixed during the denoising process, we only update the coordinates of ligands at each layer of EGNN. We have conducted the ablation study (Supplementary Table 2). If we remove the local kernel from PMDM, the performance of the deformed PMDM will drop on all metrics except the sampling speed since it has fewer parameters. We do not report the results of PMDM without global kernel since we find that PMDM cannot generate valid molecules if we remove this indispensable module. Therefore, the global kernel is necessary to restrict the positions of the atoms while the local kernel further helps to refine their relative positions.

Table 2. Ablation studies.

Methods	Vina Score (kcal/mol)	High Affinity	QED	SA	Lipinski rules	LogP	Diversity	Time (Seconds)
PMDM	-7.572 ± 2.50	0.628	0.594 ± 0.12	0.611 ± 0.16	4.975 ± 0.16	0.301 ± 1.01	0.709 ± 0.10	906 ± 1097
PMDM w.o. CA	-7.515 ± 2.26	0.620	0.606 ± 0.14	0.689 ± 0.14	4.951 ± 0.24	0.514 ± 0.97	0.704 ± 0.09	636 ± 680
PMDM w.o. local	-7.496 ± 2.37	0.598	0.582 ± 0.14	0.598 ± 0.13	4.917 ± 0.33	0.862 ± 1.46	0.720 ± 0.08	585 ± 529
Test Set	-7.024	-	0.466	0.725	1.413	0.929	-	-

3.2 Local geometries

The authors demonstrate that PMDM has a lower frequency of 3 and 4-atom rings than the external baseline models and that those frequencies align with the training set.

However, the fact that 7, 8, and 9-atom rings are overrepresented relative to the training and test set (and not merely the other generative methods) still needs to be adequately addressed and further investigated.

Our response: Thank you for pointing it out with kind suggestions. Actually, PMDM generates as few 8-atom rings and 9-atom rings as other methods. Specifically, for 9-atom rings, all the models generate 0.6% molecules except DiffSBDD (2.5%) containing such big rings. PMDM only generates 6.1% of molecules with 8-atom rings while CVAE generates 1.4%, AR-SBDD generates 0.7%, and DiffSBDD generates 5.7% of such molecules.

We chose the scales of 1.0 and 0.10 for the axes to emphasize the small proportions occupied by the 8-atom and 9-atom rings generated by PMDM. This provides a clear visual representation of the relatively low occurrence of these larger rings in the generated molecules. Although PMDM generates relatively more 7-atom rings, such rings are relatively more stable than 8-atom rings and 9-atom rings. In organic chemistry, 7-membered rings can be found in various compounds and can exhibit stability. For example, cycloheptane is a common organic compound with a 7-membered carbon ring. It is relatively stable due to the sp^3 hybridization of carbon atoms and the presence of single bonds between them.

Correspondingly, PMDM is designed to generate a larger number of 6-atom rings compared to all the baseline models. This is achieved through the local edge construction design incorporated in PMDM, enabling the model to effectively generate molecules with 6-atom rings.

Figure 3b

3.3 Figure 3a: The authors incorporated the suggested feedback appropriately.

Our response: Thank you for your kind comment!

3.4 Distribution of generated molecules

In each of the Fig. 4a, 4b, and 4c, the structures generated by PMDM are dramatically more spread out than the test set. I agree with the authors that the generated molecules completely cover the chemical space covered in the test set. The issue appears to be the narrow scope of the test set chosen. The authors claim that the wider distribution of the generated molecules implies the capacity of PMDM to “generate novel molecules across a broader chemical space”. However, more reasoning as to why this is the case, even though the shape distributions (Fig. 4d, 4e) are nearly equivalent, is required. Exploring the chemical structures generated outside the test-set distribution would be helpful. Are these synthetically accessible beyond what is predicted by the SA score? Do they match the distribution of the training set? Is the test set appropriate if it does not cover a broad enough chemical space?

Our response: Thank you for your insightful comments. As a conditional generative model, our approach generated the molecules given the target pockets of the test set. Consequently, we compare the chemical space of the generated molecules with the molecules in the test set. The shape of the generated molecules is determined by the structures of the pockets, resulting in a distribution of generated molecules that is similar to that of the molecules in the test set.

To enhance the diversity of the generated molecules, PMDM incorporates the reparameterization trick, introducing noise during each sampling process. This noise injection improves the exploration of different chemical structures, leading to a broader range of generated molecules.

Furthermore, the structural information of the protein pocket serves as spatial guidance for the generated molecules. This guidance enables them to explore a limited chemical space within specific conformations that are relevant to the pocket structures in the test set. By focusing on these specific conformations, the generated molecules can capture wider chemical spaces that are specific to the target pockets, rather than attempting to cover the entire chemical space.

By considering the pocket structures in the test set and incorporating noise and spatial guidance, our model generates molecules that are adapted to the specific pocket structures, capturing specific and relevant wider chemical spaces. This targeted exploration of chemical space allows the generated molecules to align with the requirements and characteristics of the test set pockets.

Unfortunately, without access to golden wet-lab metrics for comparing the synthetic accessibility of out-of-distribution (OOD) molecules, we are unable to provide a direct comparison. However, we have calculated the mean SA score for the OOD molecules to address your concerns, which yields a reasonable value of 0.628 ± 0.28 . The SA score is a widely accepted metric for evaluating the synthetic accessibility of molecules, and the OOD molecules in our study adhere to the same SA scoring rules.

It is important to note that our study did not entail any specific design for synthesizing molecules, and thus, we do not anticipate the OOD molecules to be inherently easier to

synthesize than the molecules in the test set. We acknowledge the limitation of not having wet-lab experiments for direct validation, but by considering the SA score and adhering to established rules, we can draw reasonable inferences regarding the synthetic accessibility of the OOD molecules.

Table S3. The chemical property values of OOD molecules.

QED	SA	Lipinski rules	LogP
0.448±0.16	0.628±0.19	4.713±0.73	1.065±2.22

We have visualized the distributions of the molecules in the training set in the following figure. As depicted in the figure, the distribution of the generated molecules is capable of encompassing the distribution of molecules in the training set.

Regarding the choice of the test set, it aligns with our specific objective. Rather than aiming to cover the entirety of chemical space, our focus is on generating molecules that are well-suited for the specific target pockets under consideration. It is important to note that different target pockets may necessitate entirely different molecule structures due to their unique characteristics and requirements.

By concentrating on the suitable compact chemical space that corresponds to the specific target pockets, we can generate molecules that are optimized for those specific contexts. This tailored approach allows us to prioritize the exploration of relevant chemical space, enhancing the likelihood of generating molecules with desirable properties for the given target pockets.

Figure 4d

3.5 Demonstrating improved generated structures

The authors incorporate an additional computational metric, MM-PBSA, to validate the structures generated by PMDM. They synthesize one structure against CDK2 and report potent inhibition. While this effort is appreciated and an impressive result, more robust

controls and details of the experimental methods are needed to make conclusions about the ability of Compound 9024 to inhibit CDK2. For example, does Compound 9024 hit any other kinases or, specifically, any other CDKs?

Our response: Thank you very much for your constructive comment. We have conducted extensional wet-lab experiments to synthesize more molecules which is presented in the revised manuscript. Among them, compound 6849, which exhibits better potency against CDK2 than compound 9024, turns out to be an advanced lead molecule during the lead optimization campaign and exhibited good selectivity against other closely related kinases, including CDK9 (CDK9/T1 inhibition $IC_{50} = 32.3$ nM, CDK9/CDK2= 127) and GSK3 β (GSK3 β inhibition $IC_{50} = 703$ nM, GSK3 β /CDK2= 2780).

In order to further address your concerns, we have revised the section **PMDM enables scaffold hopping and linker generation for real synthetic bioactive molecule design** as follows:

3.6 - 3.14 *The authors addressed the original reviewer's comments and clarified the text appropriately.*

Our response: Thank you very much for your kind comment.

Other comments

4.1 *On page 2 the authors state, "However, these methods do not involve the pocket information to generate molecules with high binding affinity. As a result, the resulting molecules cannot fit into specific protein pockets, which is not suitable for wet experiments." I fail to see how not including information about the specific pocket strictly prohibits previous works from generating molecules worthy of wet lab experiments. If this is true, it requires more discussion and suggests substantial novelty of the PMDM model.*

Our response: Thank you for your considerable comments. The binding of a small molecule to a specific protein pocket is crucial for inhibiting or activating particular biological functions at the molecular level. Neglecting the consideration of target protein pockets in drug design can substantially diminish the success rate of subsequent cell experiments. The previous works^{1,2,3,4} focus on generating molecules from scratch or just conditioning on specific chemical properties such as polarizability in 3D space. They just generate molecules geometries and do not attempt to enable the molecules to bind with target proteins. Consequently, molecules generated through these approaches are unsuitable for cell experiments as they do not prioritize binding with the pockets to modulate biological functions.

It is essential to recognize that the interaction between small molecules and target protein pockets is a fundamental aspect of effective drug design. By taking into account the specific protein pockets, drug designers can increase the likelihood of generating molecules that possess the desired binding affinity and functional modulation capabilities. Ignoring this critical aspect may lead to the production of molecules that lack the necessary characteristics to interact effectively with the target proteins, thereby compromising their potential as successful drug candidates.

Therefore, in our research, we aim to address this limitation by developing a novel approach that incorporates the structural information of the target protein pockets. By leveraging this information, we can generate molecules that are specifically tailored to interact with the desired protein pockets and modulate their biological functions effectively. Our model, PMDM, takes into account the 3D structure of the target pocket as conditional information and captures the interactions between molecules and proteins, enabling us to learn the conditioned density of desired molecular data.

Another category of methods in this field is called ligand-based methods^{5,6,7,8}. These approaches primarily focus on learning the distribution of ligands that exhibit activity towards the target proteins. Ligand-based methods aim to generate molecules that are similar to known active ligands, relying solely on information from the ligand perspective. However, a significant limitation of ligand-based models is their inability to effectively incorporate protein structure information, which is crucial for designing new hits and generating target-specific molecules⁹.

The inherent drawback of ligand-based models is that they heavily rely on the availability of sufficient bioactive molecule data. If the training data for specific proteins is limited or inadequate, these models may struggle to generate valid molecules. Additionally, ligand-based methods often require retraining when dealing with new target proteins, rendering them less versatile and unsuitable for generalization to arbitrary targets.

It is important to acknowledge the challenges and limitations associated with ligand-based approaches. While these methods can generate molecules similar to known ligands, their reliance solely on ligand information restricts their ability to incorporate critical protein structure details and design molecules that are specifically tailored to interact with target protein pockets.

Instead, it is crucial for molecular generative models to go beyond learning solely from the topological data present in compound libraries¹⁰. These models should also extract the underlying interaction principles from geometric data, such as the structure of protein pockets and their corresponding ligands. Our proposed model PMDM, as a structure-based method, is able to perceive the 3D structure of the target pocket as the conditional information and the interaction between molecules and proteins to learn the conditioned density of desired molecular data.

Through extensive experimental analysis, we have demonstrated the capability of our model to generate novel molecules with a remarkable binding affinity against specific proteins. These results serve as compelling evidence of the efficacy of our structure-based approach in generating molecules that exhibit high potential for binding and modulating the biological functions of target proteins.

Reference

1. Gebauer, N., Gastegger, M. & Schütt, K. Symmetry-adapted generation of 3d point sets for the targeted discovery of molecules. *Adv. neural information processing systems* 32 (2019).
2. Huang, Lei, et al. "Mdm: Molecular diffusion model for 3d molecule generation." *Proceedings of the AAAI Conference on Artificial Intelligence*. Vol. 37. No. 4. 2023.
3. Satorras, V. G., Hoogeboom, E., Fuchs, F. B., Posner, I. & Welling, M. E(n) equivariant normalizing flows. In *Beygelzimer, A., Dauphin, Y., Liang, P. & Vaughan, J. W. (eds.) Advances in Neural Information Processing Systems (2021)*.
4. Hoogeboom, E., Satorras, V. G., Vignac, C. & Welling, M. Equivariant diffusion for molecule generation in 3d. In *International Conference on Machine Learning*, 8867–8887 (PMLR, 2022).
5. Zang, Chengxi, and Fei Wang. "Moflow: an invertible flow model for generating molecular graphs." *Proceedings of the 26th ACM SIGKDD international conference on knowledge discovery & data mining*. 2020.
6. Jin, Wengong, Regina Barzilay, and Tommi Jaakkola. "Junction tree variational autoencoder for molecular graph generation." *International conference on machine learning*. PMLR, 2018.
7. Shi, Chence, et al. "Graphaf: a flow-based autoregressive model for molecular graph generation." *arXiv preprint arXiv:2001.09382* (2020).
8. Gao, Kaifu, et al. "Generative network complex for the automated generation of drug-like molecules." *Journal of chemical information and modeling* 60.12 (2020): 5682-5698.
9. Xie, Weixin, et al. "Advances and challenges in de novo drug design using three-dimensional deep generative models." *Journal of Chemical Information and Modeling* 62.10 (2022): 2269-2279.
10. Liu, Tiqing, et al. "BindingDB: a web-accessible database of experimentally determined protein–ligand binding affinities." *Nucleic acids research* 35.suppl_1 (2007): D198-D201.

4.2 Figure 4 attempts to show the distribution of molecules in chemical space generated by PMDM compared to the test set. The text on Page 6, however, claims the comparison is to the training set. This incongruity should be clarified.

Why are the plots duplicated in each panel? For example, the top left sub-panel in Fig. 4a is a mirror of the bottom right and the top right is a reflection of the bottom left. This is the case for each panel. It also needs to be clarified what the tick marks represent. What is “-200” in a kernel-density estimation?

Our response: Thanks for your considerable comments. We apologize that there is a typo error in the sentence “PMDM can correctly model the 2D chemical space of the training set (Figure 4a and Figure 4b)”. We have revised this sentence as the following:

“PMDM can correctly model the 2D chemical space of the test set (Figure 4a and Figure 4b).”

We plot the chemical space distribution using t-SNE by plotting pairwise relationships. A univariate distribution plot is drawn to show the marginal distribution of the data in each column in the diagonal plots. By employing pairwise plots, we aim to provide a clearer illustration of the chemical space distribution of both the generated molecules and the test molecules, specifically highlighting the pairwise relationship between t-SNE1 and t-SNE2.

If you believe that using pairwise plots may not be suitable or have concerns regarding their usage, we are more than willing to address these concerns. Alternatively, we can replot Figure 4a, b, and c to accommodate your feedback and ensure the clarity and accuracy of the visualization.

-200 is the range of the t-SNE values for Morgan fingerprint features of the molecules. It has no specific meaning. In order to further address your concerns, we choose to hide the axis values.

4.3 The plots in Figure 3 for the 8-atom ring and 9-atom ring should be scaled proportionally to the highest frequency not '1.0', to align with the other panels. As it

stands, the visualization appears to hide the overrepresentation of 8 and 9-member rings generated by PMDM compared to the training and testing sets.

Our response: Thank you for your insightful comments. We select the scales of 1.0 and 0.10 for the axes to emphasize the small proportions occupied by the 8-atom and 9-atom rings we generated. In order to address your concerns, we replot the figure with grouped bars, which is shown in the following figure.

We also have reported the figure in another suitable scales for 8-atom rings and 9-atom rings.

As we can observe, PMDM still generates a few 8-atom rings and 9-atom rings compared to other methods.

4.4 On page 11 and Figure 6b, 6c, it is unclear what *in vitro* “enzyme assay” experiment was performed. To adequately conclude this result, the relevant methods must be included.

Our response: Thank you for your considerable comments. In order to address your concerns, we have added the relevant methods in the supplementary file.

“CDK2/Cyclin E1 assay. The homogeneous time-resolved fluorescence (HTRF) assay was performed to detect CDK2/Cyclin E1 catalyzed phosphorylation of peptide substrate in

assay buffer containing 50 mM HEPES, pH=7.5, 10 mM MgCl₂, 1mM EGTA, 2 mM DTT, 0.01% Tween, 0.1% BSA. The enzymatic reaction was carried out in a 10 µL volume containing 0.15 nM CDK2/Cyclin E1 enzyme (Carna, 04-165), 80 µM ATP, 50 nM LANCE Ultra ULight™-eIF4E-binding protein 1 (Thr37/46) Peptide (PerkinElmer, TRF0128-M) and 1 % DMSO (or the test compound at appropriate dilutions in DMSO) in the assay buffer. All the components were added to the 384-well plate (PerkinElmer, 6008280), and incubated at room temperature for 4h. The reaction was terminated by addition of 10 µL detection buffer (PerkinElmer, CR97-100) containing 20 mM EDTA and 4 nM LANCE® Ultra Europium-anti-phospho-eIF4E-bindingprotein 1 (Thr37/46) (PerkinElmer, TRF0216-M) antibody. After 1 h of incubation at room temperature, plate was loaded on Envision Reader (PerkinElmer, EnVision Multilabel Reader) to measure fluorescence intensity which was used to determine IC₅₀ values of the test articles.

CDK1/Cyclin A2 assay. The HTRF assay was performed to detect CDK1/Cyclin A2 catalyzed phosphorylation of peptide substrate in assay buffer containing 50 mM HEPES, pH=7.5, 10 mM MgCl₂, 1mM EGTA, 2 mM DTT, 0.01% Tween, 0.1% BSA. The enzymatic reaction was carried out in a 10 µL volume containing 0.4 nM CDK1/Cyclin A2 enzyme (SignalChem, C22-18G), 10 µM ATP, 20 nM LANCE Ultra ULight™-eIF4E-binding protein 1 (Thr37/46) Peptide (PerkinElmer, TRF0128-M)) and 1 % DMSO (or the test compound at appropriate dilutions in DMSO) in the assay buffer. All the components were added to the 384-well plate (PerkinElmer, 6008280), and incubated at room temperature for 2 h. The reaction was terminated by addition of 10 µL detection buffer (PerkinElmer, CR97-100) containing 20 mM EDTA and 4 nM LANCE® Ultra Europium-anti-phospho-eIF4E-bindingprotein 1 (Thr37/46) (PerkinElmer, TRF0216-M) antibody. After 1 h of incubation at room temperature, plate was loaded on Envision Reader (PerkinElmer, EnVision Multilabel Reader) to measure fluorescence intensity which was used to determine IC₅₀ values of the test articles.”

4.5 *The text states, “As reported in Figure 6b, the compound 9024 displayed better CKD2 activity in enzyme assay with around 490-fold CDK1 selectivity.” This sentence is unclear. It seems that the authors are trying to say compound 9024 has a 490-fold lower IC₅₀ value against CDK2 than CDK1. From the numbers in Figure 6b of 0.000382 and 0.0187 for CDK2 and CDK1, respectively, from these numbers, it would appear that the compound has 48.9-fold higher inhibitory capacity against CDK2 than CDK1. Is 490 a typo? Additionally, it reported it appears these experiments are unreplicated.*

Our response: Thank you very much for your considerable comments. We apologize for the typo. It should be 49 fold. We have unified the units of wet-lab experiments to nm. The wet-lab data of known compound 16 is cited from the previous paper. Since we have conducted wet-lab experiments for more lead optimization compounds, we have revised the corresponding sentences as the following.

“In-vitro experiments were conducted to assay their CDK1/2 inhibitory activities. As reported in Figure 6b, all the molecules displayed improved CDK2 activity in enzyme assay, with CDK1 selectivity ranging from approximately 44-fold. Notably, compound 6849, containing the pyridinol moiety, exhibited the highest CDK2 activity, while compound 6793, which reintroduced the pyridine and formed a trivalent bond with a nitrogen atom, displayed the best CDK1 selectivity (124-fold).”

All the experiments are replicated at least twice.

4.6 An IC_{50} of $0.000382 \mu M$ for Compound 9024 against CDK2 is a surprising result. As such, more comparison of this compound to external datasets of assayed small molecules against CDK2 is warranted.

Our response: Thanks for your considerable comment. The compound 9024 is developed from the reference compound 13.

We listed 12 CDK inhibitors in the following table that have been published during the discovery of CDK2 inhibitors since the 1990s. The CDK2 and CDK1 biochemical activities of the selected CDK2 inhibitors are either tested internally using the CDK2 and CDK1 biochemical assay protocols described in this paper when they are commercially available or cited from the original publications when they are not commercially available. Obviously, compound 9024 achieves comparable or even higher CDK2 inhibition and comparable or better CDK1 selectivity compared with those published molecules. We have added this table into the supplementary file.

Table S5. Representative chemotypes and biochemical activities in the discovery of selective CDK2 inhibitors¹

Entry	Structure of CDK2 inhibitors	CDK2 /E1, IC_{50} (nM)	CDK1/A2, IC_{50} (nM)	CDK1/C DK2	Entry	Structure of CDK2 inhibitors	CDK2 /E1, IC_{50} (nM)	CDK1/A2, IC_{50} (nM)	CDK1/C DK2
1	 BMS-387032		274	13	7	 CYC065	6.4	489	76
2	 AG-024032	0.9	2.1	2.3	8 ³	 4ab	4.0	8.0	2.0

3 ²	 CDKI-277	4.0	8.0	2.0	9	 PF-06873600	0.34	4.5	13
4	 AT7619	92	172	1.9	10	 PF-07104091			38
5	 PHA-793887	5.2	55	11	11	 Compound 13	8.1	550	67
6	 BAY 1000394	0.2	0.5	2.5	12	 Compound 23	0.37	22	59

Reference

(1) The CDK2 and CDK1 biochemical activities of the selected CDK2 inhibitors are either tested internally using the CDK2 and CDK1 biochemical assay protocols described in this paper when they are commercially available or cited from the original publications when they are not commercially available.

(2) Payton, M.; Chung, G.; Yakowec, P.; Wong, A.; Powers, D.; Xiong, L.; Zhang, N.; Leal, J.; Bush, T. L.; Santora, V.; et al. Discovery and Evaluation of Dual CDK1 and CDK2 Inhibitors. *Cancer Res.* **2006**, *66* (8), 4299-4308.

(3) Singh, U.; Chashoo, G.; Khan, S. U.; Mahajan, P.; Nargotra, A.; Mahajan, G.; Singh, A.; Sharma, A.; Mintoo, M. J.; Guru, S. K.; et al. Design of Novel 3-Pyrimidinylazaindole CDK2/9 Inhibitors with Potent In Vitro and In Vivo Antitumor Efficacy in a Triple-Negative Breast Cancer Model. *J. Med. Chem.* **2017**, *60* (23), 9470-9489.

4.7 Without any controls for their wet lab experiments, it is difficult to determine if the compounds are truly selective against CDK1/CDK2. Some form of negative control for these experiments is needed.

Our response: Thank you for your considerable comment. To address your concerns, we have listed the selectivity data of representative CDK2 inhibitors that have been published as the control (see the table mentioned above). As we can observe, all the selected compounds in our manuscript achieve comparable or higher CDK2 inhibition functions, and comparable or better CDK1 selectivity compared to the molecules as the control, indicating that our compounds are truly selective against CDK2/CDK1.

4.8 *It needs to be clarified how compound 16 was generated. The text on Page 11 implies that PMDM generated the structure via linkers at preferred attachment points. It then states the ADME properties of Compound 16 were reported by the author's previous work concerning a method called Fragment-Based Variational Auto-Encoder generative model (FBVAE). While that reference does not describe the FBVAE method in detail, from the name, it appears markedly different than the PMDM model discussed here. It must be clarified if PMDM or an alternative methods was used to generate the structure, as it is used to support the claims of PMDM's generative capacity.*

Our response: Thank you for your valuable comments. We have developed the linker sampling algorithm (See Method) specifically designed to generate linkers for the fragments. This algorithm does not require retraining the model on the linker dataset. In contrast, the FBVAE is a generative model that has been trained solely on the linker dataset and does not incorporate pocket information.

By utilizing our linker sampling algorithm, our model PMDM successfully generated compound 16. Interestingly, we discovered that the previous work (FBVAE) also generated the same linker and conducted in vitro experiments on it. To provide additional support for our findings, we include the in-vitro results from the previous work, which indicates that compound 16 exhibits potent subnanomolar CDK2 activities and displays nanomolar activity towards CDK1. These results suggest a high level of CDK2 selectivity, with approximately 49-fold selectivity over CDK1.

These findings highlight the effectiveness of our approach in generating compounds with desirable properties and demonstrate the consistency between our model and the previous work (FBVAE) regarding the generation of the linker and its subsequent in vitro evaluation.

4.9 *Given the authors first claim is PMDM's ability to generate novel bioactive molecules, it is unfortunate they only chose to synthesize one molecule from the scaffold hopping case. I understand the cost associated with novel compound synthesis, but it seems "Generated sample 2164" or "Generated sample 3746" would have been ideal candidates for experimental validation, perhaps even more than Compound 9021.*

Our response: Thank you for your constructive comments. Since we primarily focus on in-silico experiments, we reported the calculation metrics for generated sample 2164 and generated sample 3746 in the case study of SARS-Cov-2. We appreciate your understanding, we do not have accessible resources for synthesizing the aforementioned lead generation molecules. Instead, we could allocate limited resources for the lead optimization molecules including, scaffold hopping and linker generation of CDK2 inhibitors. In order to further address your concerns, we have conducted wet-lab experiments to validate more molecules rather than compound 9024. Notably, in Figure 6, we present the results of our experiments with compound 6793, 6849, and

compound 7246, which exhibit improved CDK2 inhibition and demonstrate satisfying CDK1 selectivity.

Additionally, we have conducted in-vitro experiments on two other macrocyclized molecules, namely compound 6261 and compound 7138. The results from these experiments also highlight improved CDK2 inhibition.

4.10 The text states Compound 16 shows subnanomolar CDK2 activity, while Figure 6c states an IC₅₀ value of 0.090 μM or 90 nanomolar. These discrepancies make it nearly impossible to draw conclusions from the results as presented.

Our response: Thank you for your considerable comment. We apologize for the typo error. The correct unit in Figure 6c should be nM (nanomole) rather than μM (micromole). We have revised Figure 6c accordingly, and the updated version is presented below. We have unified the units of wet-lab experiments to nM. Upon reviewing the revised in-vitro results, it becomes evident that compound 16 exhibits potent subnanomolar CDK2 activity (compound 16: 0.090 nM/0.000090 μM vs reference: 8.1 nM/0.0081 μM) and displays nanomolar activity towards CDK1. This suggests a high level of CDK2 selectivity, approximately 49-fold.

Figure 6c

4.11 On page 11, the text states “We select the compound 9024 with the lowest MM-PBSA value to assay its CDK1/2 inhibitory activities.” However, Figure 6b implies via a blue arrow that “Compound:9021” was assayed. This needs to be clarified.

Our response: Thank you for your kind comment. We apologize for the typo error. It should be compound 9024. We have revised the Figure 6b, which is presented in the following.

Figure 6b

4.12 On page 11, Figure 6a is referred to as both a “reference compound” and a “remaining fragment as the seed scaffold”. The authors should clarify which is being depicted. It appears as if the “seed scaffold” is depicted in the center of Figure 6b.

Our response: Thank you for your thoughtful comments. In Figure 6a, compound 13 is the reference compound. Here, we attempt to leverage PMDM to perform scaffold hopping based on compound 13. After reviewed by chemical experts, we remove the pyridine ring and reserve the remaining fragment as the seed scaffold. We frame the removed pyridine ring with a dashed line in Figure 6a. And the remaining fragment (seed scaffold) is outlined in the center of Figure 6b. In order to address your concern, we have revised the descriptions of reference compound and seed scaffold in the section PMDM enables scaffold hopping for real synthetic bioactive molecule generation.

“After reviewed by chemical experts, we remove the pyridine ring (dashed box in Figure 6a) and reserve the remaining fragment as the seed scaffold (Figure 6a) which is the key scaffold of the existing CDK2 inhibitors.”

4.13 *There are numerous spelling, grammatical, and mislabelling errors in the manuscript and supporting information. Likewise, there are numerous instances in which the text does not align with what is depicted in the figures.*

Our response: Thank you for your precious comments. Following your comments, we have checked the manuscripts and corrected the errors.

Thank you again for your insightful comments. We greatly appreciate your time and attention in providing us with these valuable comments, as they have undoubtedly contributed to improving the quality and accuracy of our work. We would be glad to respond to any further questions and comments that you may have.

Reviewer #1 (Remarks to the Author):

The new experiments provide much stronger evidence for the usefulness of the reported method for lead generation. Therefore, I think it is now suitable for publication in this journal and have no further comments.

Reviewer #5 (Remarks to the Author):

Review.

The authors proposed a diffusion model to generate 3D molecules that potentially bind to protein pockets. Diffusion models are trendy and have been applied to image generation and protein structure generation and ligand-pocket binding. The novelty in this particular work lies in the proposed neural network architecture with parameter analysis, and the in silico validation by simulation and wet-lab experiments.

I have read previous reviewers' comments and agree with most of them. I think there are sufficient level of novelty, better performance over other methods, and validations that warrant this work to be accepted by Nat Comm. I do have the following critiques that the authors can improve on.

Fig 3.

Am I correct in that Fig 3B is mentioned before Fig 3A in the text ?

I think Fig 3A can be improved. Instead of having three panels of bar charts, it is helpful to show the three plots in one panel. It is also useful to show such distribution for other methods. I know these information can be deduced from Fig 3B but please make it easier for the readers.

Fig 3B : what does the Y-axes indicate ? If they indicate the % of different size of rings among all rings, then for each individual method the % of 3-atom, 4-atom ... 8-atom should add to 1, but they don't. For example, for PMDM, 6-atom ring % = ~ 0.8 , 7-atom ring % = 0.4.

I think it needs to be explained why PMDM generated significantly more 7-atom rings than other methods. The explanation in the text is not satisfactory.

Fig 3C/3D: in addition to just comparing KL divergences in bond pairing and lengths, it is important to visualize the distributions produced by individual prediction methods, and the test set. This needs to be included in the Supplementary Materials, and one representative figure needs to be included in the main text.

Fig 4A, 4B, 4C. Here the authors use t-SNE plots to show that the PMDM predictions encompass the test set. I am not used to look at these types of presentations, some further clarification may be helpful. I am curious why there is only one single data point for the test dataset ?

For every panel in Fig 4, the authors need to provide comparative plots for other methods such as CAVE or SBDD. I am curious whether these characteristics are unique for the diffusion model or are shared by computational methods. These additional plots can be included in the Supplementary Material and described in main text.

English and writing needs further improvement.

It is just my preference, but I think the Results section should start with an overview of the algorithm / approach. After all, this is a methodology paper and the methodology itself is "result". After a brief introduction of the overall pipeline, readers can refer to the Methods section for more details.

Since this is a Methodology paper, I think it is appropriate to have only two test cases with only in vitro validation included in the paper. If this paper is tailored for Nat Chem Biology or Nat Biotech, then more experimental validation is needed.

I don't think I see a link to the codes or the data used in the work, a link to github should be presented in the Abstract or end of Intro.

Methods:

The graph neural network SchNet figures prominently in their pipeline. They provided brief references to this network in the Intro (ref 14, 15), but they also need to cite them in the Methodology part (page 14) and explain it briefly.

Reviewer #6 (Remarks to the Author):

See attached file

Reviewer #6 Attachment on the following page

This work developed a Pocket based Molecular Diffusion Model (PMDM) for 3D molecule generation, which has significant novelty and is valuable for structure-based drug design. Accompanied with the previous revisions according to the reviewer's comments, it has been improved but there are still significant issues with the description of wet experiments. They didn't give any description about the synthesis of compounds including the synthesis procedures and reaction conditions. About the biological activity evaluation, they also didn't give any detailed description, such as the used methods, the original data and figures, etc. If they want to publish this paper, such information is necessary to ensure the reliability of results.

Response Letter

Authors' Point-by-Point Responses to the Reviewers' Comments on "A dual diffusion model enables 3D binding bioactive molecule generation and lead optimization based on target pockets" by Huang et al. (NCOMMS-23-06531-A) submitted to Nature Communications.

Reviewer #1 (Remarks to the Author):

The new experiments provide much stronger evidence for the usefulness of the reported method for lead generation. Therefore, I think it is now suitable for publication in this journal and have no further comments.

Thank you for taking the time to review our revised manuscript. We sincerely appreciate your positive feedback and your recognition of the stronger evidence that we provided to support the usefulness of our proposed method for lead generation. Your assessment that our manuscript is now suitable for publication in this journal is truly encouraging.

We would like to express our sincere gratitude for your thorough review and valuable feedback throughout the entire review process. Your insights and suggestions have been instrumental in helping us improve the quality and clarity of our work. We are grateful for the opportunity to address your comments and concerns.

Reviewer #5 (Remarks to the Author):

The authors proposed a diffusion model to generate 3D molecules that potentially bind to protein pockets. Diffusion models are trendy and have been applied to image generation and protein structure generation and ligand-pocket binding. The novelty in this particular work lies in the proposed neural network architecture with parameter analysis, and the *in silico* validation by simulation and wet-lab experiments.

I have read previous reviewers' comments and agree with most of them. I think there are sufficient level of novelty, better performance over other methods, and validations that warrant this work to be accepted by Nat Comm. I do have the following critiques that the authors can improve on.

Thank you for your valuable comments and the constructive advice on improving the manuscript. Your comments really helped a lot to improve the study. We have summarized the suggested comments (as highlighted) and made point-by-point responses and revisions as follows.

5.1 Fig 3. Am I correct in that Fig 3B is mentioned before Fig 3A in the text?

Our response: Thank you for your kind comments, it is indeed that we describe Figure 3B first. To address your concern and make it clear, we put the descriptions of Figure 3A in front of Figure 3B.

5.2 I think Fig 3A can be improved. Instead of having three panels of bar charts, it is helpful to show the three plots in one panel. It is also useful for show such distribution for other methods. I know these information can be deduced from Fig 3B but please make it easier for the readers.

Our response: We greatly appreciate your constructive suggestions. Based on your comments, we plot the three figures as shown below:

All the three plots are in one panel with a shared y axel and legend.

5.3 Fig 3B : what does the Y-axes indicate ? If they indicate the % of different size of rings among all rings, then for each individual method the % of 3-atom, 4-atom ... 8-

atom should add to 1, but they don't. For example, for PMDM, 6-atom ring % = ~0.8, 7-atom ring % = 0.4.

Our response: Thank you for your considerable comments. The Y-axes indicate the proportion of the molecules containing different rings in the datasets and those generated by PMDM and other methods. A molecule could contain several rings with different sizes, which will be counted for several times. Hence, the sum of proportion of the molecules is not 1. To make it clearer for readers, we revised the descriptions in the corresponding part of the manuscript.

“To further quantify the ring sub-structure of the molecules generated by those methods, we report the proportion of molecules containing rings of different sizes in the training set, the test set, and the generated sets from the methods. In the case of molecules that contain multiple rings, the counting process takes into consideration each individual ring present, resulting in repeated counts proportional to the number of rings.”

5.4 *I think it needs to be explained why PMDM generated significantly more 7-atom rings than other methods. The explanation in the text is not satisfactory.*

Our response: Thank you for your insightful feedback. The reason is twofold. First, in figure 3a, it is evident that both training set and the molecules generated by PMDM exhibit a Gaussian-like distribution in the number of rings. This observation demonstrates PMDM's significant similarity to real data in terms of the number of rings in the generated molecules. From the perspective of the diffusion model, DiffSBDD also generate relatively more molecules with 7-atom rings. This phenomenon is likely due to the fact that diffusion models generate samples using standard Gaussian noise and predefined scheduled noise. Consequently, in the early stages of the reverse process, the number and size of clusters formed by atoms based on density exhibit an inverse relationship. That is why the output of the proposed PMDM also consists more 6-atom and 8-atom rings than the training set.

Second, the implementation of the dual edges design in the PMDM aims to prevent the generation of unstable small rings and macro rings, while simultaneously ensure the production of stable 6-atom rings. However, the current approach solely relies on pairwise distances to construct these edges, lacking explicit constraints to control the size of the rings. As a result, PMDM tends to generate rings of intermediate sizes. It is important to acknowledge that distinguishing between 7-atom rings and 6-atom rings at the geometric level poses is challenging, given their similar structural appearances.

In our future work, we intend to address this limitation by incorporating explicit constraints into the model to avoid the generation of unstable rings. One potential approach is to leverage reinforcement learning during the training process, introducing a reward function that provides immediate rewards when generating stable rings or desired structures.

In order to further address your concern, we have revised the corresponding part.

“

It is evident that both training set and the molecules generated by PMDM exhibit a Gaussian-like distribution in the number of rings. We also notice that PMDM generates relatively more molecules with 7-atom rings. This is due to the fact that distinguishing between 7-atom rings and 6-atom rings at the geometric level is challenging, given their similar structural appearances.

”

5.5 Fig 3C/3D: in addition to just comparing KL divergences in bond pairing and lengths, it is important to visualize the distributions produced by individual prediction methods, and the test set. This needs to be included in the Supplementary Materials, and one representative figure needs to be included in the main text.

Our response: Thank you for your considerable comments. We visualize the bond angle distributions and dihedral angle distributions of the methods and test set by utilizing violin plots. We choose to put these figures in the supplementary files.

Upon observation, it is evident that PMDM demonstrates the ability to capture the local atom angle geometries exhibited in the data. Moreover, the angles of molecules generated by PMDM exhibit greater stability compared to those generated using autoregressive methods.

5.6 Fig 4A, 4B, 4C. Here the authors use *t*-SNE plots to show that the PMDM predictions encompass the test set. I am not used to look at these types of presentations, some further clarification may be helpful. I am curious why there is only one single data point for the test dataset?

Our response: Thank you for your considerable comment. There is not one single data point for the test dataset. It has 100 single data points, but they are very close to each other. Hence, it looks like a single point although it indeed consists of 100 data points.

5.7 For every panel in Fig 4, the authors need to provide comparative plots for other methods such as CAVE or SBDD. I am curious whether these characteristics are unique for the diffusion model or are shared by computational methods. These additional plots can be included in the Supplementary Material and described in main text.

Our response: Thanks for your considerable comments. In Figure 4a, b and c, we adopt both 2D and 3D molecular fingerprints, including Morgan, RDKit, and USRCAT (Ultrafast Shape Recognition with CREDO Atom Types) fingerprints, to visualize the chemical space of the molecules generated by our methods and those in the test set. In response to your suggestion, we have included the three figures for all the baseline models in the supplementary file. Our findings reveal that all the baseline methods successfully cover the chemical spaces represented by RDKit and USRCAT fingerprints of the test molecules. However, both CVAE and DiffSBDD fail to cover the Morgan chemical space of the test molecules. On the other hand, while AR-SBDD shows a significant coverage of the Morgan chemical space, its diversity appears to be limited. Furthermore, we observed that the USRCAT distributions of CVAE and DiffSBDD tend to cluster into a few distinct groups, suggesting a restricted diversity in their 3D structures. To further address your concerns, we added the descriptions about the supplementary figure.

“We have conducted an analysis of the chemical space distribution of molecules generated by other baseline models, as depicted in the Supplementary Figure 2. Our findings indicate that these models are unable to fully encompass the Morgan chemical space of the test set. Additionally, we observed that both CVAE and PMDM exhibited limited diversities in the 3D space.”

In Figure 4d, we depict the Normalized Principal Moment of Inertia ratios (NPR) on a ternary plot. The NPR distributions of other methods are shown in the following figure.

We observed that the distribution of PMDM closely resembles that of the test set, exhibiting a higher degree of similarity compared to other methods. Notably, both the molecules generated by PMDM and those in the test set tend to cluster around the rod corner of the triangular representation. In contrast, the molecules generated by CVAE are predominantly concentrated in the triangle's central region. Similarly, molecules generated by DiffSBDD and AR-SBDD also tend to gather in the rod corner, although DiffSBDD demonstrates lower diversity and the molecules generated by AR-SBDD extend towards the disc corner of the triangle.

To further address your concerns, we have added the descriptions about the NPR plots of other methods.

“In addition, we have generated the NPR (Normalized Property Ratio) distributions of molecules produced by alternative baseline models, as shown in Supplementary Figure 3. Our observations indicate that the molecules generated by PMDM exhibit the closest resemblance to the test set distribution. Conversely, the molecules generated by CVAE tend to cluster in the central region, while those from AR-SBDD extend towards the disc corner. Furthermore, DiffSBDD displays limited diversity in its generated molecules.”

Follow your suggestion, we plotted the PBF (Plane of Best Fit) figures for all the baseline models. Specifically, we truncate the PBF values to the range of 0 to 10 for a fair comparison due to the significantly exceeding maximum value of the CVAE.

Upon analysis, we observe a significant discrepancy in the PBF (Projected Bond Factor) values between the molecules generated by CVAE and those in the test set. This discrepancy implies that the degree of distance from the 2D shape is dissimilar to that of the test set. In contrast, AR-SBDD shows a closer distance between the heavy atoms and the predefined plane. Similarly, DiffSBDD demonstrates a comparable distance from the 2D shape to that of the test set.

We also added the descriptions about the PBF plots of other methods to address your concerns.

“We observe that the PBF values of the generated molecules align well with those of the test set molecules, indicating a similar degree of distance from the 2D shape. In contrast, CVAE exhibits a substantial gap compared to the test set, suggesting that the heavy atoms are significantly distant from the predefined plane (Supplementary Figure 4).”

English and writing needs further improvement.

5.8 *It is just my preference, but I think the Results section should start with an overview of the algorithm / approach. After all, this is a methodology paper and the methodology itself is “result”. After a brief introduction of the overall pipeline, readers can refer to the Methods section for more details.*

Since this is a Methodology paper, I think it is appropriate to have only two test cases with only in vitro validation included in the paper. If this paper is tailored for Nat Chem Biology or Nat Biotech, then more experimental validation is needed.

Our response: Thanks for your considerable comments. We agree with you that it would be appropriate to include an overview of PMDM before the experimental results. In response to your suggestion, we have added a subsection to include the overview of proposed PMDM.

“Overview of the PMDM model

Figure 1 outlines an overview of the conditional generative model PMDM, elucidating its structural components and the processes involved in training and sampling. PMDM gradually introduces Gaussian noise in the forward process while employing a parameterized reverse process to iteratively eliminate the noise (Figure 1a). The model comprises two invariant graph neural networks Schnet³⁶ to obtain the molecule embeddings z_L and pocket embeddings h_P (Figure 1b). To facilitate conditional generation, we have designed two context mechanisms to incorporate both the semantic and geometric information of the protein pocket. Specifically, cross-attention layers are utilized to calculate the attention scores of the molecule and protein, protein pocket. Additionally, a dual diffusion strategy is employed to enable the model to discern atom-wise forces. This strategy involves constructing two types of virtual edges. Firstly, pairs of atoms with interatomic distances below the local threshold τ_l are bonded via covalent localized edges because chemical bonds tend to dominate interatomic forces when atoms are in proximity. Secondly, we build the global edges which are linked to the remaining pairs of atoms to simulate the van der Waals force for the atoms whose distances are greater than the local threshold τ_l but less than the global threshold τ_g (Figure 1d). Furthermore, we have designed an equivariant dynamic kernel that adheres to the translation, rotation, reflection, and permutation equivariance of molecular geometry systems. To ensure that the generated molecule is adapted to the structure pocket, the pocket position remains fixed during the update of the hidden states in the dual equivariant encoders.

In the training stage, both molecules and their corresponding binding protein pockets are regarded as 3D point clouds. In the forward process of PMDM, the molecule input undergoes diffusion, resembling phenomena observed in nonequilibrium thermodynamics, with the sampled time step drawn from the union distribution. Meanwhile, the protein pocket input remains fixed as it serves as the conditional information (Figure 1c). The primary objective of PMDM is to learn how to reverse this process to model a conditioned data distribution. This enables the efficient generation of accurate molecules with high binding affinity when the pocket information is fixed. At each time step, the model outputs the (Stein) score, which represents the logarithmic density of the data point. The ELBO objective is derived from these scores and serves as the loss function (See method).

In the sampling stage, we initialize the data state by sampling from $N(0, I)$ and obtain the transition probability by the dual equivariant encoder of PMDM, given the target pocket protein. The next less chaotic states are iteratively generated by $p_\theta(\mathcal{G}_{t-1}^L | \mathcal{G}^L, \mathcal{G}^P)$. The final molecule \mathcal{G}_0 is generated by progressively sample \mathcal{G}_{t-1} for T times. Finally, the atom types of the molecule are identified by adopting the argmax function to choose the atom type that has the largest value while we directly adopt r_0^L outputted by the model.

”

We incorporate more than two test cases since the previous reviewers think it would be convincing if we include more in-vitro results to validate the accuracy and efficiency of PMDM based on the previous comments. We appreciate your kind comments, but it would be more comprehensible to validate the generated molecules if we include more in-vitro results in this manuscript.

5.9 *I don't think I see a link to the codes or the data used in the work, a link to github should be presented in the Abstract or end of Intro.*

Our response: Thank you for your comment. This is the link for the code: https://drive.google.com/drive/folders/1Tsr7G3OV4VLirPQ31CSj1DKJtZ3hC1bt?usp=drive_link. We will release the GitHub link of the code as soon as the manuscript is accepted.

Methods:

5.10 *The graph neural network SchNet figures prominently in their pipeline. They provided brief references to this network in the Intro (ref 14, 15), but they also need to cite them in the Methodology part (page 14) and explain it briefly.*

Our response: Thank you so much for your considerable comments. We provided the algorithm in the method part. To address your concerns, we have cited the reference of SchNet and added a brief explanation in the method section.

“Here, we adopt an invariant graph neural network SchNet³⁶ to encode the protein semantic information first. SchNet is a graph neural network modeling quantum interaction in molecules in 3D space. It consists of continuous-filter convolutional layers to model atomistic systems and maintain the invariant properties, achieving state-of-the-art performance for benchmarks of equilibrium molecules and molecular dynamics trajectories.”

Thank you very much for the comments. We deeply appreciate your valuable suggestions. Those comments are of great value for improving the quality of the manuscript.

Reviewer #6 (Remarks to the Author):

This work developed a Pocket based Molecular Diffusion Model (PMDM) for 3D molecule generation, which has significant novelty and is valuable for structure-based drug design. Accompanied with the previous revisions according to the reviewer's comments, it has been improved but there are still significant issues with the description of wet experiments. They didn't give any description about the synthesis of compounds including the synthesis procedures and reaction conditions. About the biological activity evaluation, they also didn't give any detailed description, such as the used methods, the original data and figures, etc. If they want to publish this paper, such information is necessary to ensure the reliability of results.

Our response: Thank you so much for your considerable comments and suggestions. We have already put the details about the wet-lab experiments in the supplementary file. Here, we present general experimental procedures, and spectroscopic data of the Compound 9024 as an example.

Experimental procedures and spectroscopic data of Compound 9024

General Procedures. All reactions were carried out under a nitrogen atmosphere with dry solvents under anhydrous conditions, unless otherwise noted. No unexpected or unusually high safety hazards were encountered. Low-resolution mass spectra (LC-MS) was used to monitor progression of reactions and was recorded on Waters ACQUITY UPLC with SQ Detectors using a Waters CORTECS C18 column (2.7 μm , 4.6 \times 30 mm) using a gradient elution method: solvent A: 0.1% formic acid in water; solvent B: 0.1% formic acid in CH_3CN ; 5% solvent B to 95% solvent B in 1.0 min, hold 1.0 min, equilibration to 5% solvent B in 0.5 min; flow rate: 1.8 mL/min; column temperature 40 $^\circ\text{C}$. Purification of final products by Prep-HPLC were carried out on Waters Prep-HPLC with QDA detector, using Xbridge C18 column (5 μm , 150 \times 19 mm) using a gradient elution method. ^1H NMR spectra were recorded on a Bruker Ascend 400 spectrometer. Chemical shifts are expressed in parts per million (ppm, δ units). Coupling constants are in units of hertz (Hz). Splitting patterns describe apparent multiplicities and are designated as s (singlet), d (doublet), t (triplet), q (quartet), quint (quintet), m (multiplet), br (broad). Benzyl (1-(tert-butyl)-3-((1S,3R)-3-hydroxycyclopentyl)-1H-pyrazol-5-yl)carbamate, Benzyl (1-(tert-butyl)-3-((1R,3S)-3-hydroxycyclopentyl)-1H-pyrazol-5-yl)carbamate and (trans, rac)-3-(5-(((benzyloxy)carbonyl)amino)-1-(tert-butyl)-1H-pyrazol-3-yl)cyclopentyl methanesulfonate were purchased from PharmaBlock.

Scheme S1. Preparation of compound 9024: (1R,3S)-3-(3-(pyrazin-2-ylamino)-1H-pyrazol-5-yl)cyclopentyl isopropylcarbamate

Step 1: (1R,3S)-3-(1-(tert-butyl)-5-(pyrazin-2-ylamino)-1H-pyrazol-3-yl)cyclopentan-1-ol

To a solution of (1R,3S)-3-(4-amino-1-tert-butylpyrazol-3-yl)cyclopentanol (94.0 mg, 421 μ mol) in 1,4-dioxane (5.0 mL) were sequentially added 2-bromopyrazine (100 mg, 631 μ mol), Cs₂CO₃ (411 mg, 1.26 mmol), Pd₂(dba)₃ (24.4 mg, 42.1 μ mol), and XantPhos (24.4 mg, 42.1 μ mol) at 25 °C. The reaction was warmed to 90 °C and at that temperature for 4 h. The mixture was diluted with water (20 mL) and extracted with EtOAc (30 mL \times 3). The combined organic layers were washed with brine (20 mL), dried over Na₂SO₄, filtered and concentrated. The residue was purified by silica gel chromatography eluting with EtOAc/PE (with EtOAc from 0 to 50% in 20 min) to afford (1R,3S)-3-(1-(tert-butyl)-5-(pyrazin-2-ylamino)-1H-pyrazol-3-yl)cyclopentan-1-ol (60.0 mg, 47% yield) as a light-yellow solid. LC-MS: m/z [M+H]⁺ calculated for C₁₆H₂₄N₅O⁺ 301.2, found 302.1.

Step 2: (1R,3S)-3-(1-(tert-butyl)-5-(pyrazin-2-ylamino)-1H-pyrazol-3-yl)cyclopentyl isopropylcarbamate

To a solution of (1R,3S)-3-(1-(tert-butyl)-5-(pyrazin-2-ylamino)-1H-pyrazol-3-yl)cyclopentan-1-ol (60.0 mg, 199 μ mol) in MeCN (20.0 mL) were sequentially added NMM (101 mg, 995 μ mol), (4-nitrophenyl) carbonochloridate (120 mg, 597 μ mol) and DMAP (48.6 mg, 398 μ mol) at 25 °C. The reaction mixture was stirred at that temperature for 3 h before isopropylamine (235 mg, 3.98 mmol) was added. The mixture was stirred at 25 °C for 8 h before it was concentrated under vacuum. The residue was purified by silica gel chromatography eluting with EtOAc/PE (with EtOAc from 0 to 50% in 20 min) to afford (1R,3S)-3-(1-(tert-butyl)-5-(pyrazin-2-ylamino)-1H-pyrazol-3-yl)cyclopentyl isopropylcarbamate (30.0 mg, 39% yield) as a yellow solid. LC-MS: m/z [M+H]⁺ calculated for C₂₀H₃₁N₆O₂⁺ 386.2, found 387.2.

Step 3: (1R,3S)-3-(3-(pyrazin-2-ylamino)-1H-pyrazol-5-yl)cyclopentyl isopropylcarbamate

A solution of (1R,3S)-3-(1-(tert-butyl)-5-(pyrazin-2-ylamino)-1H-pyrazol-3-yl)cyclopentyl isopropylcarbamate (30.0 mg, 77.6 μ mol) in formic acid (10.0 mL) was warmed to 50 °C and stirred at that temperature for 12 h. The mixture was cooled to 25 °C and concentrated under vacuum. The residue was purified by silica gel

chromatography eluting with EtOAc/PE (with EtOAc from 0 to 80% in 25 min) to afford a crude product which was further purified by Prep-HPLC (with CH₃CN from 10% to 40% in 8 min) to give (1R,3S)-3-(3-(pyrazin-2-ylamino)-1H-pyrazol-5-yl)cyclopentyl isopropylcarbamate (13.0 mg, 51% yield) as a white solid. LC-MS: m/z [M+H]⁺ calculated for C₁₆H₂₃N₆O₂ 330.2, found 331.2. ¹H NMR (400 MHz, DMSO-d₆): δ = 12.03 (s, 1H), 9.63 (s, 1H), 8.51 (s, 1H), 8.08 (s, 1H), 7.87 (d, J = 2.8 Hz, 1H), 6.97 (d, J = 7.6 Hz, 1H), 6.21 (s, 1H), 5.00 (s, 1H), 3.60 – 3.55 (m, 1H), 3.07 – 3.03 (m, 1H), 2.48 – 2.44 (m, 1H), 2.08 – 1.99 (m, 1H), 1.92 – 1.84 (m, 1H), 1.74 – 1.57 (m, 3H), 1.03 (d, J = 6.4 Hz, 6H).

The experimental settings and steps of other generated compounds are also included in the supplementary file.

Regarding the about the biological activity evaluation, we have also included the protocols of intro assays (CDK2/Cyclin E1 assay, CDK1/Cyclin A2 assay). In order to include more comprehensive biological activity evaluation descriptions, we have supplemented the descriptions about the CDK9/Cyclin T1 assay and GSK3β assay and dose-response curves of the reported compounds to inhibit the kinase activity of CDK2/E1 and CDK1/A2, and compounds 6849 inhibit the kinase activity of CDK9/T1 and GSK3β. To further address your concerns, we have included theses supporting information in the supplementary.

CDK2/Cyclin E1 assay.

LANCE Ultra time-resolved fluorescence energy transfer (TR-FRET) assay was performed to detect CDK2/Cyclin E1 catalyzed phosphorylation of peptide substrate in assay buffer containing 50 mM HEPES, pH=7.5, 10 mM MgCl₂, 1mM EGTA, 2 mM DTT, 0.01% Tween, 0.1% BSA. The enzymatic reaction was carried out in a 10 μL volume containing 0.15 nM CDK2/Cyclin E1 enzyme (Carna, 04-165), 80 μM ATP, 50 nM LANCE Ultra ULight™-eIF4E-binding protein 1 (Thr37/46) Peptide (PerkinElmer, TRF0128-M) and 1 % DMSO (or the test Compound at appropriate dilutions in DMSO) in the assay buffer. All the components were added to the 384-well plate (PerkinElmer, 6008280), and incubated at room temperature for 4 h. The reaction was terminated by addition of 10 μL detection buffer (PerkinElmer, CR97-100) containing 20 mM EDTA and 4 nM LANCE® Ultra Europium-anti-phospho-eIF4E-bindingprotein 1 (Thr37/46) (PerkinElmer, TRF0216-M) antibody. After 1 h of incubation at room temperature, plate was loaded on Envision Reader (PerkinElmer, EnVision Multilabel Reader) to measure fluorescence intensity which was used to determine IC₅₀ values of the test articles.

CDK1/Cyclin A2 assay.

The LANCE Ultra time-resolved fluorescence energy transfer (TR-FRET) assay was performed to detect CDK1/Cyclin A2 catalyzed phosphorylation of peptide substrate in assay buffer containing 50 mM HEPES, pH=7.5, 10 mM MgCl₂, 1mM EGTA, 2 mM

DTT, 0.01% Tween, 0.1% BSA. The enzymatic reaction was carried out in a 10 μ L volume containing 0.4 nM CDK1/Cyclin A2 enzyme (SignalChem, C22-18G), 10 μ M ATP, 20 nM LANCE Ultra ULight™-eIF4E-binding protein 1 (Thr37/46) Peptide (PerkinElmer, TRF0128-M)) and 1 % DMSO (or the test Compound at appropriate dilutions in DMSO) in the assay buffer. All the components were added to the 384-well plate (PerkinElmer, 6008280), and incubated at room temperature for 2 h. The reaction was terminated by addition of 10 μ L detection buffer (PerkinElmer, CR97-100) containing 20 mM EDTA and 4 nM LANCE® Ultra Europium-anti-phospho-eIF4E-bindingprotein 1 (Thr37/46) (PerkinElmer, TRF0216-M) antibody. After 1 h of incubation at room temperature, plate was loaded on Envision Reader (PerkinElmer, EnVision Multilabel Reader) to measure fluorescence intensity which was used to determine IC50 values of the test articles.

CDK9/Cyclin T1 assay. The LANCE Ultra time-resolved fluorescence energy transfer (TR-FRET) assay was performed to monitor CDK9/Cyclin T1-catalyzed phosphorylation of peptide substrate. The enzymatic reaction was carried out in a 10 μ L volume containing 10 nM CDK9/Cyclin T1 enzyme (Carna, 04-110), 10 μ M ATP, 50 nM Ultra ULight™-MBP substrate (PerkinElmer, TRF0109) in the assay buffer (50 mM HEPES, PH=7.5, 10 mM MgCl₂, 1mM EGTA, 2 mM DTT, 0.01% Tween). All the components were added to the 384-well plate and incubated at room temperature for 2 h. The reaction was terminated by addition of 10 μ L detection buffer containing 20 mM EDTA and 4 nM Ultra Europium-anti-phospho-MBP antibody (PerkinElmer, TRF0201). After 1 h of incubation at room temperature, the plate was loaded on Envision Reader to measure fluorescence intensity which was used to determine IC50 values of the test articles.

GSK3 β assay. The LANCE Ultra time-resolved fluorescence energy transfer (TR-FRET) assay was performed to monitor GSK3 β -catalyzed phosphorylation of peptide substrate. The enzymatic reaction was carried out in a 10 μ L volume containing 1 nM GSK3 β enzyme (Carna, 04-141), 10 μ M ATP, 5 nM Ultra ULight™-eIF4E-binding protein 1 (Thr37/46) peptide (PerkinElmer, TRF0128) and 1% DMSO (or the test articles at appropriate dilutions in DMSO) in the assay buffer (50 mM HEPES, PH=7.5, 10 mM MgCl₂, 1mM EGTA, 2 mM DTT, 0.1 mg/mL BSA, 0.01% Tween). All the components were added to the 384-well plate and incubated at room temperature for 1h. The reaction was terminated by addition of 10 μ L detection buffer containing 20 mM EDTA and 4 nM Ultra Europium-anti-phospho-eIF4E-binding protein 1 (Thr37/46) antibody (PerkinElmer, TRF0216). After 1h of incubation at room temperature, the plate was loaded on Envision Reader to measure fluorescence intensity which was used to determine IC50 values of the test articles.

Figure S1 Representative dose-response curves of the reported compounds to inhibit the kinase activity of CDK2/E1

Figure S2 Representative dose-response curves of the reported compounds to inhibit the kinase activity of CDK1/A2

Figure S3 Representative dose-response curves of Compound 6849 to inhibit the kinase activity of CDK9/T1

Figure S4 Representative dose-response curves of Compound 6849 to inhibit the kinase activity of GSK3 β

Thank you again for your insightful comments. We greatly appreciate your time and attention in providing us with these valuable comments, as they have undoubtedly contributed to improving the quality and accuracy of our work.

Reviewer #5 (Remarks to the Author):

I have carefully read author's response and examined the revised manuscript. The authors have satisfactorily addressed the concerns I raised. The figures and the text read better now. I support the publication of the work in the current form.

I suggest the authors to provide more explanation on the software package that they provided on the google drive page. I suppose they will be eventually moved to github, or figshare or similar stable depository ? Instructions should be given clearly on what each sub-package does.

The authors need to think about how these tools can be smoothly run by other researchers.

Reviewer #6 (Remarks to the Author):

The authors have revised their manuscript and added the related description about the wet experiments according to the reviewer's comments. I think it can be accepted for publication now.

Response Letter

Authors' Point-by-Point Responses to the Reviewers' Comments on "A dual diffusion model enables 3D molecule generation and lead optimization based on target pockets" by Huang et al. (NCOMMS-23-06531-A) submitted to Nature Communications.

Reviewer #5 (Remarks to the Author):

I have carefully read author's response and examined the revised manuscript. The authors have satisfactorily addressed the concerns I raised. The figures and the text read better now. I support the publication of the work in the current form.

We highly appreciate your positive and encouraging feedback. Your assessment that our manuscript is now suitable for publication in this journal is truly encouraging.

We would like to express our sincere gratitude for your thorough review and valuable feedback throughout the entire review process. Your insights and suggestions have been instrumental in helping us improve the quality and clarity of our work. We are grateful for the opportunity to address your comments and concerns.

5.1 I suggest the authors to provide more explanation on the software package that they provided on the google drive page. I suppose they will be eventually moved to github, or figshare or similar stable depository? Instructions should be given clearly on what each sub-package does.

The authors need to think about how these tools can be smoothly run by other researchers.

Our response: Thank you so much for your considerable comments. We have provided the source code, introduction and documents of PMDM at GitHub repository: <https://github.com/Layne-Huang/PMDM/tree/main>. These files are also available at Zenodo: <https://zenodo.org/records/10631358>.

Reviewer #6 (Remarks to the Author):

The authors have revised their manuscript and added the related description about the wet experiments according to the reviewer's comments. I think it can be accepted for publication now.

Thank you very much for taking time to review our manuscript. We greatly appreciate your time and attention in providing us with these valuable comments, as they have undoubtedly contributed to improving the quality and accuracy of our work.